# Harnessing CD3 diversity to optimize CAR T cells

Rubí M.-H. Velasco Cárdenas[1,2], Simon M. Brandl [1,2,3], Ana Valeria Meléndez[1,2,3,15], Alexandra Emilia Schlaak[2,4,15], Annabelle Buschky[1,2,3], Timo Peters[5], Fabian Beier[6], Bryan Serrels[7,8], Sanaz Taromi [9,10], Katrin Raute[1,2,3], Simon Hauri [11], Matthias Gstaiger[11], Silke Lassmann[6], Johannes B. Huppa [5], Melanie Boerries [12,13], Geoffroy Andrieux[12], Bertram Bengsch [2,4], Wolfgang W. Schamel [1,2,14,16] & Susana Minguet [1,2,14,16] ✉

Current US Food and Drug Administration-approved chimeric antigen receptor (CAR) T cells harbor the T cell receptor (TCR)-derived ζ chain as an intracellular activation domain in addition to costimulatory domains. The functionality in a CAR format of the other chains of the TCR complex, namely CD3δ, CD3ε and CD3γ, instead of ζ, remains unknown. In the present study, we have systematically engineered new CD3 CARs, each containing only one of the CD3 intracellular domains. We found that CARs containing CD3δ, CD3ε or CD3γ cytoplasmic tails outperformed the conventional ζ CAR T cells in vivo. Transcriptomic and proteomic analysis revealed differences in activation potential, metabolism and stimulation-induced T cell dysfunctionality that mechanistically explain the enhanced anti-tumor performance. Furthermore, dimerization of the CARs improved their overall functionality. Using these CARs as minimalistic and synthetic surrogate TCRs, we have identified the phosphatase SHP-1 as a new interaction partner of CD3δ that binds the CD3δ–ITAM on phosphorylation of its C-terminal tyrosine. SHP-1 attenuates and restrains activation signals and might thus prevent exhaustion and dysfunction. These new insights into T cell activation could promote the rational redesign of synthetic antigen receptors to improve cancer immunotherapy.

CARs combine the specificity of monoclonal antibodies with the signaling machinery of the T cell antigen receptor (TCR–CD3) and are used for cancer immunotherapy[1–3]. CARs consist of an antigen-binding domain linked to hinge, transmembrane and intracellular signaling domains. US Food and Drug Administration (FDA)-approved CARs contain a costimulatory molecule (4-1BB or CD28) and the TCR ζ chain as the intracellular signaling domain[4]. The mechanisms underlying how these CARs mediate T cell activation are not fully understood. The ζ-based CARs might activate pathways that differ from those engaged by the TCR–CD3 because CARs lack the complexity that has coevolved with the TCR–CD3 (refs. 5,6).

The TCR–CD3 is a multiprotein complex conformed by the TCR-αβ heterodimer, which lacks signaling domains, together with the ζ homodimer and the CD3δ, CD3ε and CD3γ subunits, which constitute the signaling apparatus. CD3δ/ε/γ each contains an extracellular immunoglobulin (Ig) domain, a transmembrane and a cytoplasmic intracellular domain (ICD), which contains a single immunoreceptor tyrosine-based activation motif (ITAM)[7]. In contrast, ζ has a short extracellular stalk and a longer ICD with three ITAMs. In fact, very early studies set the path for the engineering of CARs by demonstrating that ζ ICD was sufficient to activate T cells[1]. Even though the ITAMs share a conserved YxxL/I-x$_{6–8}$-YxxL/I sequence, the amino acids of each ITAM

are distinct, displaying different binding affinities to signaling molecules[8]. The TCR–CD3 has, altogether, ten ITAMs. The multitude of ITAMs probably contribute to signal amplification, because reducing the number of ITAMs leads to impaired TCR–CD3 function in murine models[9]. Furthermore, ITAM diversity is important for signal transduction and T cell development even when the number of ITAMs is conserved[10]. Despite the ITAMs, the ICDs of each CD3 subunit differ in their molecular interactions. ζ and CD3ε contain basic rich stretches (BRSs), which mediate ionic interactions with the inner leaflet of the plasma membrane[11,12]. CD3ε interacts with the kinase Lck either through ionic interactions between the BRS and the acidic residues in the Lck unique domain or in a noncanonical fashion between the receptor kinase (RK) motif and the Lck SH₃ domain[13,14]. CD3ε also contains a proline-rich sequence (PRS), which recruits, among others, the adaptor protein Nck (ref. 15). CD3γ holds a membrane proximal di-leucine motif involved in TCR downregulation[16]. In CD3δ, no motifs beside the ITAM have been identified so far. The different motifs present in each subunit might play a relevant role in the CAR context. However, to date, studies evaluating the outcome of exchanging ζ with CD3δ/ε/γ ICDs in CARs are missing.

## Results

### The ICDs from CD3δ/ε/γ generate functional CAR T cells
We used an anti-CD19, second-generation CAR containing 4-1BB and engineered the CD3δ, CD3ε or CD3γ ICDs into the carboxy terminus (henceforth termed BBδ, BBε and BBγ, respectively). The BBζ CAR was used as a reference (Fig. 1a and Extended Data Fig. 1a). Human T cells were isolated from healthy donors and transduced with lentiviruses encoding the different CAR-T2A–green fluorescent protein (GFP) constructs. GFP⁺ cells were sorted (Fig. 1b). All new CARs were detected on the surface of T cells (Fig. 1c,d). All BBδ/ε/γ CAR T cells showed similar killing of CD19⁺ pre-B cell leukemic Nalm6 cells in vitro compared with BBζ cells (Fig. 1e and Extended Data Fig. 1b). Even at low effector-to-target (E:T) ratios, all presented a similar killing efficacy (Fig. 1f). To test the therapeutic potential of BBδ/ε/γ CARs, we employed a stress test in Nalm6-bearing mice[14], in which CAR T cell doses are purposefully lowered to levels where CAR T cell therapy begins to fail (Extended Data Fig. 1c). Remarkably, mice bearing T cells expressing BBδ/ε/γ CARs survived significantly longer than those treated with BBζ CAR T cells. Among the newly designed CARs, BBδ showed the best anti-tumor function (Fig. 1g–i). Hence, one ITAM is sufficient to generate functional CAR T cells.

### CAR T cell responses are defined by the TCR ICDs
Next, we characterized the BBδ/ε/γ CAR T cells in vitro. Tonic CAR signaling resulted in the expression of activation markers (CD25, 4-1BB and CD69) in unstimulated conditions, but no differences among the CARs were observed (Extended Data Fig. 2a). We did not detect any changes in the expression of endogenous TCR–CD3, T cell maturation stages or the CD4-to-CD8 ratio (Extended Data Fig. 2b–d). These results indicate that all TCR–CD3 ICDs equally induced tonic signaling in a CAR context. Next, CAR T cells were cocultured with Nalm6 cells for 24 h. BBε and BBζ CAR T cells presented a higher percentage of activated cells than BBγ and BBδ (Fig. 2a). BBζ CAR T cells differentiated more after contact with target cells, exhibiting a lower proportion of naive T cells (Fig. 2b). The presence of less differentiated populations in the BBδ/ε/γ CAR

T cells compared with BBζ might be associated with their improved in vivo performance, because CAR T cell populations with higher proportions of naive T cells displayed greater anti-tumor responses and persistence in vivo[17,18]. BBζ CAR T cells produced the largest amounts of the cytokines tested, whereas BBγ and BBδ CAR T cells secreted cytokines in similar fashion to control cells (Mock) (Fig. 2c). One of the most common side effects of CAR T cell immunotherapy is the cytokine release syndrome (CRS), which can be prevented in mice by blocking the granulocyte–macrophage colony-stimulating factor (GM-CSF)[19]. It is interesting that only BBζ CAR T cells secreted significant amounts of GM-CSF (Fig. 2c).

Continuous antigenic stimulation for 18–24 h commits T cells to replenish and upregulate TCR–CD3 expression[20]. On CAR stimulation, all cells expressing a CAR displayed higher levels of TCR–CD3 on the cell surface (Fig. 2d). Our data thus suggest efficient T cell activation and crossregulation between CARs and TCRs. Next, we analyzed the calcium signaling capacity of our new constructs using a planar glass-supported lipid bilayer (SLB)[21] (Fig. 2e). We found that BBζ was the most sensitive CAR. BBζ T cells responded at the lowest antigen density despite exhibiting the highest levels of tonic calcium signaling. BBε T cells fluxed calcium only at higher antigen densities, whereas those expressing BBγ and BBδ failed to flux calcium regardless of the antigen densities (Fig. 2f). Consistent with this, BBδ and BBγ CAR T cells also failed to degranulate on coincubation with CD19⁺ Nalm6 cells (Fig. 2g,h). Killing by all CAR T cells involved Fas–FasL, but not TRAILR1/2–DR5, interactions (Fig. 2i). Blocking the Fas–FasL axis reduced the killing by 18% for BBζ, 48% for BBε, 39% for BBδ and 47% for BBγ (Fig. 2i). Altogether, BBζ and BBε CARs exhibited stronger signaling capacity linked to more potent degranulation than BBγ and BBδ CARs. These data unraveled different modes of unleashing cytotoxic tumor cell eradication by CARs containing different ICDs.

### TCR ICDs imprint distinct transcriptional signatures
GFP⁺ CAR T cells were sorted and messenger RNA isolated to analyze gene expression. Principal component analysis (PCA) demonstrated that the constructs clustered by donor but not by ICD. This suggests that the expression of a given ICD by itself does not globally change the transcriptome signature (Extended Data Fig. 3a). Differentially expressed gene (DEG) analysis revealed differential tonic signaling of the CARs. BBε delivered the least tonic transcriptional activity, whereas BBδ caused marked changes in individual gene expression (Extended Data Fig. 3b,c). Among the DEGs, a pathway analysis using a CAR T cell gene expression panel (NanoString) showed that all constructs activate the expression of costimulatory molecules, genes related to T cell exhaustion/dysfunction and chemokine signaling (Extended Data Fig. 3d). BBε was the only CAR notably enhancing the expression of genes related to glycolysis, suggesting a previously unknown link between the CD3ε ICD and metabolism. Furthermore, generally applicable gene-set enrichment (GAGE; Extended Data Fig. 3e) analysis indicated that all CARs exhibited a stimulation-independent signature associated with signaling by costimulatory molecules of the CD28 family, programmed cell death protein 1 (PD-1) and second messengers. BBγ CAR T cells were significantly linked to cell division. BBζ CAR T cells exhibited a significant transcriptional signature associated with interferon (IFN)-γ responses and cytokine signaling. All CARs downregulated genes related to metabolism of lipids. We next

---

**Fig. 1 | CARs containing CD3δ/ε/γ ICD outperformed the CAR containing ζ. a**, Schematic representation of the CARs used. **b–d**, Percentage of positive CAR T cells (**b**) and surface CAR expression of GFP⁺ sorted cells from one representative donor (**c**) or from several donors pooled (**d**) (*n* = 6). **e**, Specific killing of CD19⁺ Nalm6 cells by primary human CAR T cells (1:1 ratio) for 6–8 h (*n* = 8, except for BBδ: *n* = 7). **f**, Specific killing at different cell-to-cell ratios (one representative donor from two is shown; *n* = 3 independent cocultures). **g**, The log(rank) Mantel–Cox survival test of Nalm6-bearing mice treated with

CAR T cells sorted for GFP expression. **h,i**, Leukemia progression (average radiance) from weeks 2 and 3 post-CAR T cell injection (**h**) and average radiance analyzed through a 60-d period (**i**) (*n* = 11–24 mice pooled from 3 independently performed experiments). Each dot represents an independent donor (**d** and **e**) or mouse (**h**). Data are represented as mean ± s.d. One-way ANOVA followed by Dunnett's multiple-comparison test (**d**, **e** and **h**) was used. APC, allophycocyanin. UTD, untransduced cells.

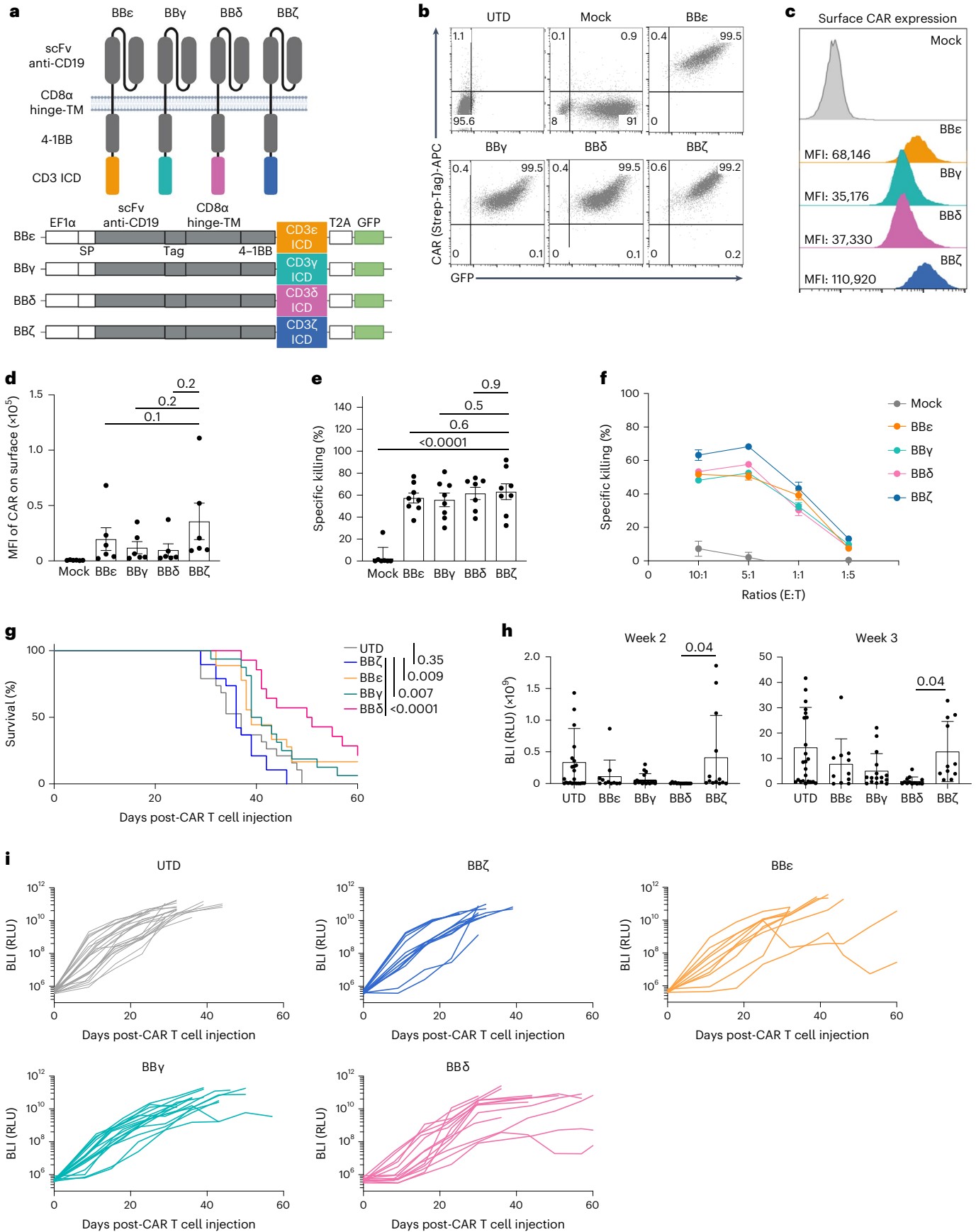

investigated the transcripts that were solely regulated by a given ICD or commonly regulated. CD3δ and CD3ε activated the most unique transcriptional signature (Extended Data Fig. 3f,g). Unexpectedly, very little overlap was found among the TCR–CD3 ICDs, and only two transcripts, inducible T cell costimulator ligand (ICOSLG) and CCL22, were commonly upregulated by all CARs. ICOS signaling was reported as part of the stimulation-independent signature of BBζ CARs[22] and CCL22 is expressed by T cells on activation and is involved in trafficking[23]. Altogether, our experiments suggest that minor tonic signaling overlap exists among the TCR–CD3 ICDs.

To test the transcriptional programming of CAR T cells on antigen stimulation, cells were cocultured with Nalm6 cells for 24 h, sorted and analyzed. As a stimulation-dependent transcriptional signature imprinted by the costimulatory domain 4-1BB has been described[24], we included T cells expressing a CAR containing only 4-1BB (indicated as BB). Cells exposed to antigen activation clustered according to the CAR construct (Fig. 3a). On Nalm6 stimulation, >400 transcripts were differentially regulated in BB CAR T cells when compared with cells transduced with the empty vector (Mock) (Fig. 3a). Pathway analysis of BB CAR T cells compared with Mock revealed significant induction of transcriptional signatures associated with cytokine secretion, chemokine signaling, nuclear factor κ-light-chain-enhancer of activated B cell (NF-κB) activation, helper T cells ($T_H17$ cells) differentiation and expression of costimulatory molecules (Fig. 3b).

We then used BB CAR T cells as a reference to analyze transcriptional changes induced specifically by each TCR–CD3 ICD. BBζ had the greatest impact on the transcriptome (481 DEGs), BBε differentially regulated 366 genes, whereas BBγ and BBδ had a more moderate impact (238 and 236 DEGs, respectively) (Fig. 3c). All TCR–CD3 ICDs reduced 4-1BB-associated signatures (Fig. 3d and Extended Data Fig. 4a). Pathway analysis showed that all constructs activated signatures linked to T cell activation and TCR signaling, lipid metabolism and cell cycle (Extended Data Fig. 4b). BBε and BBζ significantly activated transcriptional signatures associated with metabolically active T cells (oxidative phosphorylation (OXPHOS) and glycolysis; Fig. 3d). Although BBζ stimulated signatures linked to T cell exhaustion/dysfunction and $T_H2$ cell differentiation, BBε significantly impacted the upregulation of activation markers (Fig. 3d). In contrast, BBγ and BBδ CARs significantly reduced cytokine secretion and phosphoinositide-3-kinase (PI3K)/mitogen-activated protein kinase (MAPK) activation (Fig. 3d). BBζ and BBε CAR T cells showed the strongest overlap (40% and 30%, respectively) with a previously defined TCR–CD3 activation signature using the same CAR T cell gene expression panel[24]. Next, we compared the transcripts that were exclusively regulated by each ICD (Fig. 3e and Extended Data Fig. 4c,d). The ζ ICD uniquely regulated 49 transcripts. Pathway analysis revealed a transcriptional signature indicating cytokine production (interleukin (IL)-2, IL-4, IL-5, IL-9 and IFN-γ), $T_H1$ and $T_H2$ cell differentiation and TCR activation. The CD3ε ICD uniquely regulated 16 transcripts; this limited number of genes revealed a tendency to repress $T_H17$ cell differentiation and T cell exhaustion/dysfunction, and to promote cell–cell interactions and OXPHOS metabolism. The CD3δ

and CD3γ ICDs uniquely regulated eight and three genes, respectively. They shared only three regulated transcripts, which was unexpected considering the high degree of sequence similarity between them (Fig. 3e and Extended Data Fig. 4c). We directly compared the CD3δ and CD3γ ICDs (Fig. 3f,g). Pathway analysis revealed only minor differences (Fig. 3g and Extended Data Fig. 5a). Among the statistically significant regulated pathways, BBγ promoted stronger transcriptome signatures for glycolysis, IL-6 production and its signaling, and extracellular signal-regulated kinase (ERK) activation. In contrast, BBδ induced stronger transcriptome signatures for OXPHOS, amino acid metabolism and memory T cell differentiation, which might explain the outperformance of BBδ in vivo. It is interesting that BBγ induced a stronger CD8 T cell activation signature, whereas BBδ induced CD4 T cell activation and helper T cell differentiation (Extended Data Fig. 5a). When comparing BBζ with BBε (Fig. 3f,g), we observed that the ζ ICD had a greater impact on the T cell transcriptome on stimulation. BBζ enhanced T cell activation and helper T cell differentiation more efficiently than BBε. Specifically, BBζ better promoted TCR and PI3K signaling, expression of costimulatory molecules and secretion of cytokines (Fig. 3g). BBζ showed significantly greater upregulation of apoptosis genes compared with BBε. Conversely, BBε showed a significant increase in expression of genes associated with persistence, cell–cell interactions and expression of the integrins, Lck and Lyn (Fig. 3g and Extended Data Fig. 5b), providing a possible explanation for the better in vivo performance when compared with BBζ (Fig. 1).

## CD3γ and CD3δ ICDs protect CAR T cells from dysfunction

We next designed a protocol in which CAR T cells were challenged repeatedly, mimicking persistent target cell encounters in vivo (Extended Data Fig. 6a). At the end of the culture time, we stimulated both nonchallenged (cells that have never encountered tumor cells) and repeatedly challenged (indicated as rechallenged) CAR T cells with phorbol myristate acetate (PMA)/ionomycin to evaluate their potential functionality on a single-cell level using cytometry by time of flight (CyTOF) (Extended Data Fig. 6a). A uniform manifold approximation and projection (UMAP) approach was used to visualize the high-dimensional exhaustion landscape of the 24 samples (the 4 CAR constructs BBδ, BBγ, BBε and BBζ expressed in 3 independent donors under nonchallenged and repeatedly challenged conditions) that informed about large differences in the CAR T cell landscape depending on the history of antigen encounter (Fig. 4a). In line with our transcriptome analysis, the global UMAP distribution from all the nonchallenged cells did not show major differences (Fig. 4a). In contrast, large differences in the cellular proteome were observed in repetitively challenged cells depending on the construct expressed (Fig. 4a). Analysis of the percentage of CAR T cells expressing a given marker, irrespective of the CAR construct, revealed that repeatedly challenged cells significantly upregulated proteins related to T cell activation and T cell exhaustion/dysfunction (CD38, CD71, PD-1, Tim-3, TOX (thymocyte selection-associated high mobility group box factor)), whereas they downregulated TCF-1 (T cell factor-1), which is expressed in a subset of cells with cell stem-like properties or precursors of exhausted

**Fig. 2 | BBζ and BBε CARs induce stronger T cell activation on antigen encounter. a**, Expression of activation markers by CAR T cells after 24 h of stimulation with Nalm6 (1:1 ratio) (*n* = 4). **b**, Phenotype of CAR T cells after 48 h of stimulation with Nalm6 (1:1 ratio) (*n* = 6). TCM, T central memory cells; TEM, T effector memory cells; TEMRA, T effector memory cells re-expressing CD45RA. **c**, Cytokines secreted by CAR T cells after 24 h of stimulation with Nalm6 (1:1 ratio) evaluated by ELISA (*n* = 4). **d**, Endogenous TCR–CD3 surface expression after 24 h of stimulation with Nalm6 (1:1 ratio) (*n* = 4). **e**, Schematic representation of an SLB featuring fluorescently labeled CD19 antigen, the adhesion molecule ICAM-1 (intercellular adhesion molecule 1) and the costimulatory molecule B7.1 for recognition by CAR T cells. LFA-1, Lymphocyte function-associated antigen 1. **f**, Percentage of cells fluxing calcium on

encountering CD19 at the indicated densities. Data represent three independent experiments done with three independent donors. **g,h**, Flow cytometry analysis of degranulation assayed by upregulation of CD107a in response to CD19⁺ Nalm6 contact for 4 h (**g**) and 8 h (**h**). Shown are representative dot plots (4 h) and statistical analysis (*n* = 4). BFP, blue fluorescent protein. **i**, Specific killing of CD19⁺ Nalm6 cells by primary human CAR T cells (1:1 ratio) for 12 h in the presence of the indicated blocking antibodies (*n* = 4). Specific killing was normalized to isotype control (100%). Two independent anti-Fas antibodies were used and the results pooled (*n* = 4 healthy donors, 6 independent cocultures). Each dot represents an independent donor (*n*). Data are represented as mean ± s.d. One-way (**a**, **c** and **d**) or two-way (**b** and **g**–**i**) ANOVA followed by Dunnett's (**b** and **i**) or Sidak's (**g** and **h**) multiple-comparison test was used.

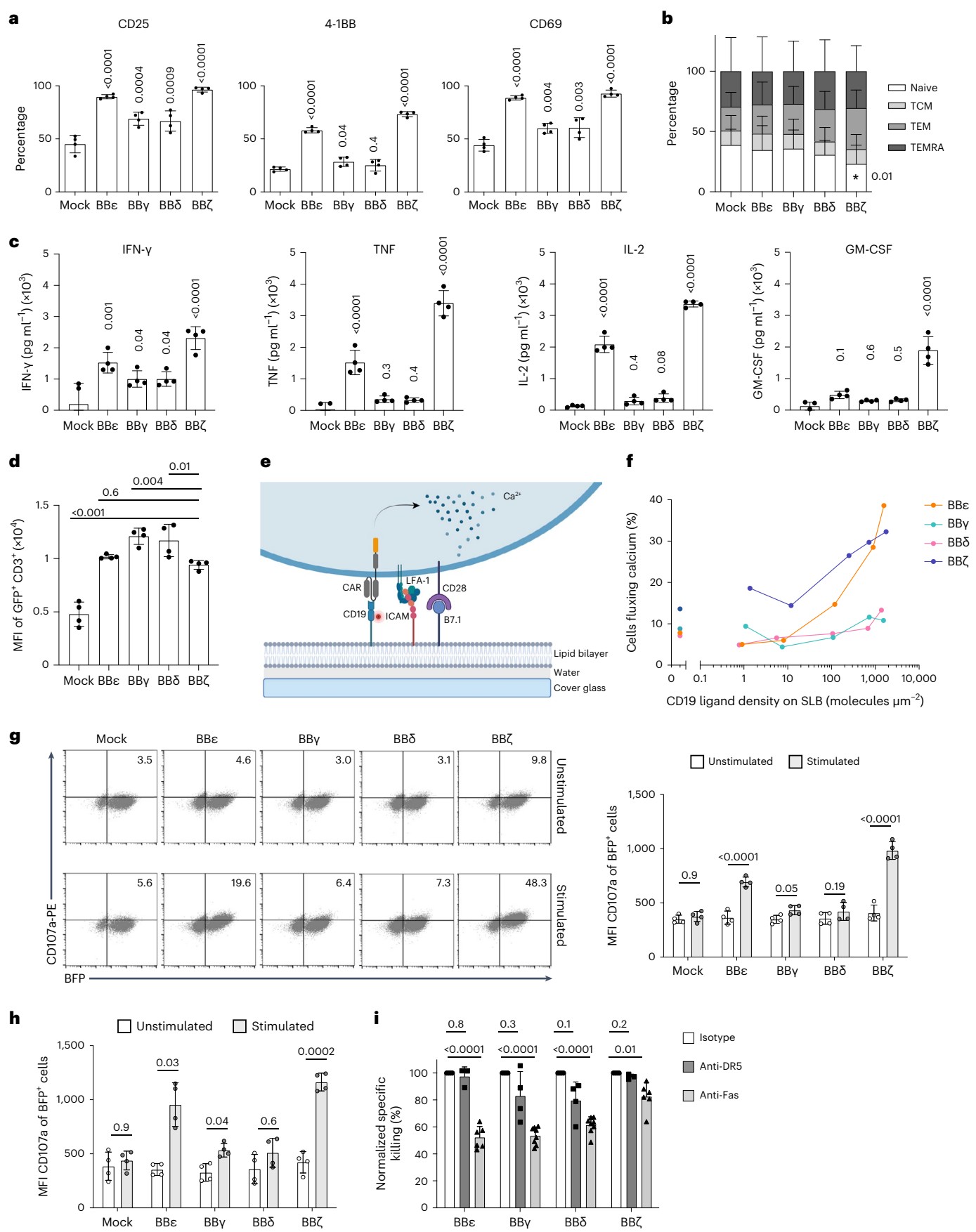

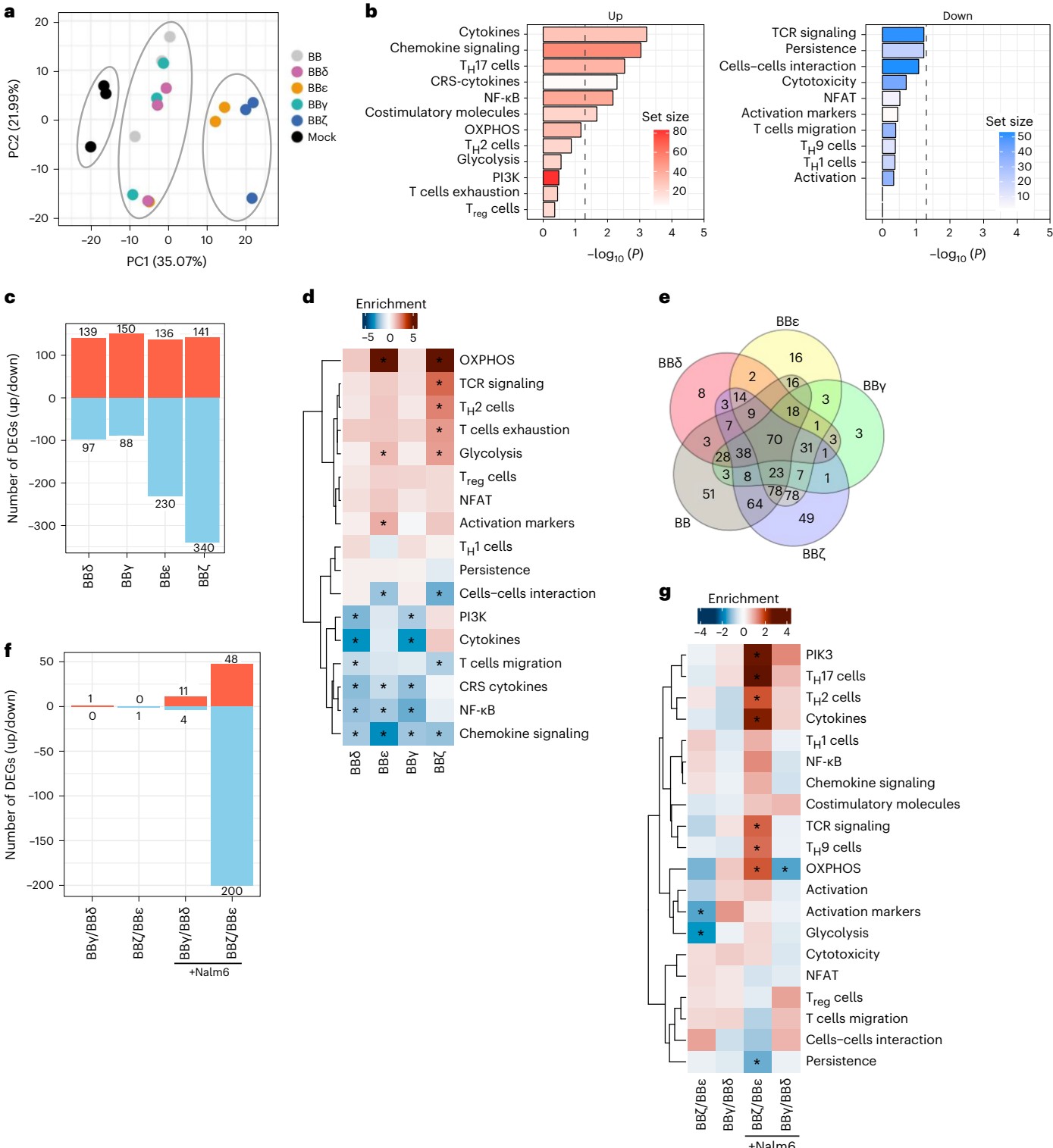

**Fig. 3 | Exclusive and shared gene expression profiles on antigen encounter.**
**a**, PCA. **b,c**, Pathway analysis of primary T cells expressing BB CAR versus Mock (**b**) and number of DEGs significantly up- and downregulated (**c**) comparing cells expressing every CAR stimulated versus BB 24 h after stimulation with CD19⁺ Nalm6 cells. **d**, Heatmap of the pathway analysis performed for each CAR construct compared with BB after 24 h of stimulation. **e**, Venn diagrams indicating the number of DEGs changed for each construct or shared among them. Every CAR is compared with the BB CAR 24 h after stimulation. **f,g**, Number of DEGs significantly up- or downregulated (**f**) and heatmap of the pathway analysis (**g**) performed for BBζ versus BBε and BBγ versus BBδ. Stimulation conditions are indicated. Three independent donors were analyzed in all panels. For **d** and **g**, a two-sample Student's *t*-test was used. *Indicates statistical significance; *P* < 0.05. Exact *P* values are provided in Source data.

T cells with memory and self-renewing properties[25,26] (Fig. 4b). The functional exhaustion score (FES) illustrates the dysfunctional cytokine production pattern by exhausted cells[27]. This FES was

significantly upregulated in repetitively challenged cells (Fig. 4b). Repetitively challenged CAR T cells produced fewer proinflammatory cytokines (IFN-γ, tumor necrosis factor (TNF) and IL-2) and more of the

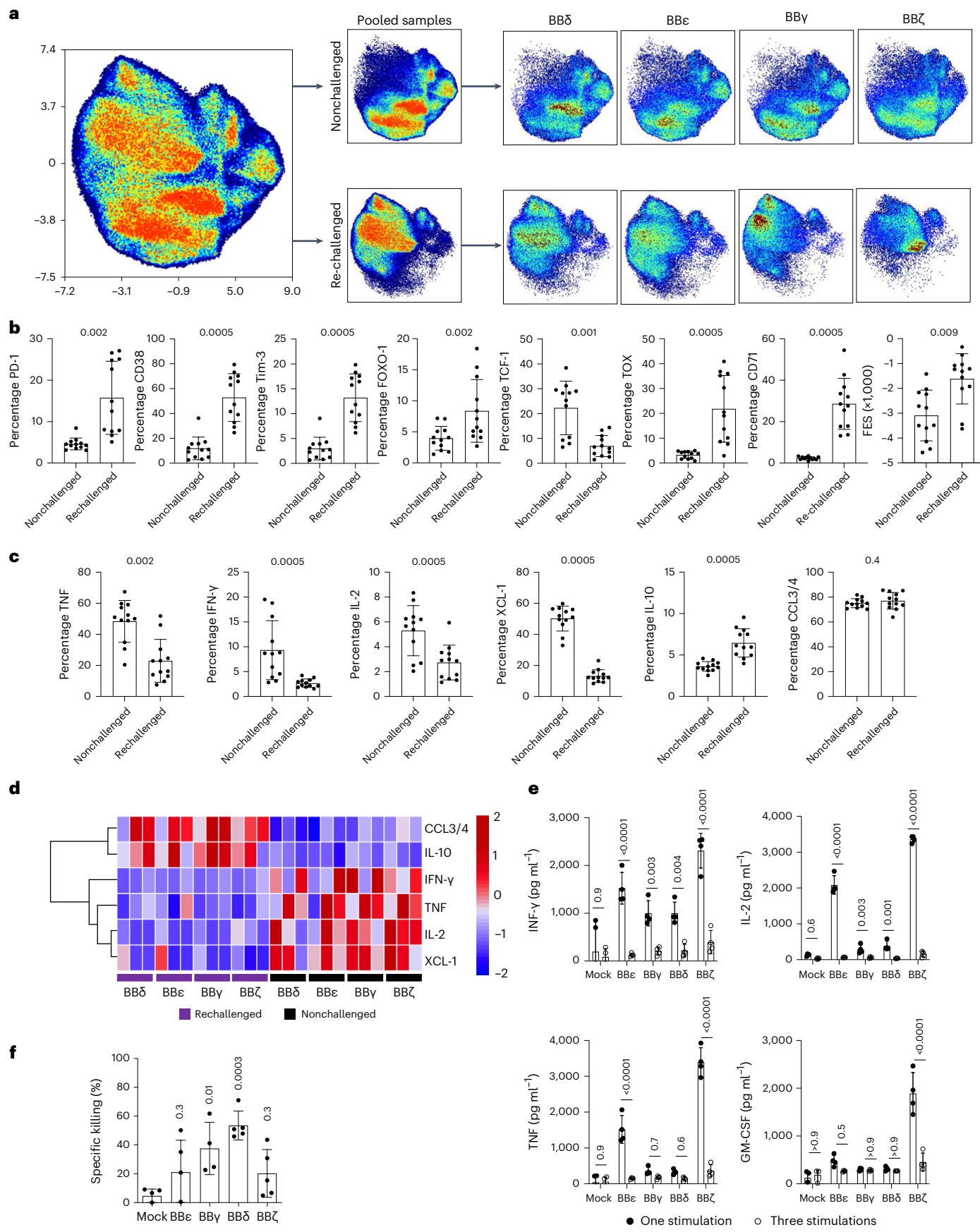

**Fig. 4 | CD3γ and CD3δ ICDs protect CAR T cells from dysregulation on serial antigen encounters. a**, UMAP plot from CyTOF data showing all the samples together (left), pooled nonchallenged or rechallenged cells (middle) and cells sorted by CAR (right). **b,c**, Percentage of cells expressing T cell exhaustion/dysregulation markers (**b**) and cytokine production (**c**) comparing all nonchallenged samples pooled with all rechallenged samples pooled. **d**, Heatmap showing the cytokine production in nonchallenged and rechallenged conditions. **e**, Cytokine secretion assayed by ELISA of CAR T cells on incubation with Nalm6 cells (1:1 ratio) for 24 h after one (black) or three (white) challenges. Each dot represents an independent donor ($n = 4$). **f**, Specific killing of CD19$^+$ Nalm6 cells by CAR T cells submitted to repetitive stimulations (E:T ratio = 1:1 for 6–8 h). Each dot represents an independent donor ($n = 3$ for **a**–**d**, $n = 4$ for **e**, $n = 5$ for **f**). Data are represented as mean ± s.d. Wilcoxon's two-tailed test (**b** and **c**), two-way ANOVA followed by Sidak's multiple-comparison test (**e**) or one-way ANOVA followed by Dunnett's multiple-comparison test (**f**) is used.

anti-inflammatory cytokine IL-10 than nonchallenged cells despite the CAR construct expressed (Fig. 4c,d). We validated cytokine production on one additional coincubation with Nalm6 cells using ELISA. Even though all CAR T cells showed a reduction in cytokine secretion after sequential antigen encounter, BBζ displayed the most prominent reduction (Fig. 4e). Repetitively challenged BBγ and BBδ CAR T cells expressed remarkably lower levels of the exhaustion markers PD-1, Tim-3, LAG-3 and CTLA4 than the BBζ and BBε T cells (Extended Data Fig. 6b,c). This suggests that CARs that provided a more potent activation signal on antigen encounter (Fig. 2) might be more prone to T cell dysfunction after serial tumor cell encounters. These differences remarkably impacted the killing of Nalm6 cells by the CAR T cells: BBδ CAR T cells maintained the best ability to kill tumor cells, followed by BBγ, whereas BBζ and BBε T cells retained reduced cytotoxicity (Fig. 4f). Killing by repetitively challenged CAR T cells correlates best with the median survival obtained in vivo (Fig. 1g; Pearson's correlation = 0.04) in comparison to killing by freshly generated CAR T cells (Fig. 1e,g; Pearson's correlation = 0.28).

### CD3δ ICD maintains precursor CAR T cells expressing TCF-1

Maintenance of precursor populations expressing high levels of the transcription factor TCF-1 correlates with lasting immunotherapies. The proportion of cells expressing TCF-1 was highest in BBδ and lowest in BBζ CAR T cells using both CyTOF and intracellular flow cytometric analyses (Fig. 5a–c). We thus interrogated the impact on CAR T cell differentiation programs. FlowSOM clustering revealed nine clusters on the pooled, repeatedly challenged, CAR T cells (Fig. 5d,e). BBζ-expressing cells had the most distinct cluster distribution with the lowest percentage of cluster 1 and the highest percentage of clusters 2 and 3 (Fig. 5f). Cluster 1 is characterized by high expression of Eomes, CD45Ra, TCF-1, CD27 and FOXO1 (forkhead box protein O1) and, therefore, it is enriched in T cells with self-renewal and memory properties (Fig. 5g). Cluster 1 is indeed most enriched in BBδ cells in line with lasting immunotherapy (Figs. 5g and 1). Clusters 2 and 3 are characterized by Ki-67 expression, indicating cell cycle activity consistent with enrichment of effector T cells. Cluster 2 is enriched in cells expressing IRF4, T-bet and Eomes that are CCR7$^-$ in line with activated effector T cells, whereas cluster 3 upregulates CCR7, suggesting differentiation toward memory (Fig. 5g). We next assayed the capacity to produce effector cytokines (Fig. 5h). Cluster 1 CAR T cells secreted IL-2, chemokine ligand XCL-1 and TNF consistent with a stem cell memory phenotype. Cells in cluster 2 highly produced the effector cytokines IL-2, CCL3/4, IL-10, IFN-γ and TNF according to an effector phenotype. Clusters 1 and 2 have a low score on the FES, whereas clusters 3, 6 and 9 had the highest FES scores, suggesting exhaustion. These results indicate that BBζ CAR T cells more efficiently differentiate to activated effector and exhaustion programs at the expense of TCF-1 precursor populations. To further test this notion, we performed Wanderlust analysis to identify possible differentiation trajectories[28]. Trajectory inference indicated progressive differentiation from the TCF-1$^{high}$ cluster 1 cells toward activated and exhausted T cells, which acquired the highest expression of PD-1 while lacking TCF-1 expression (Fig. 5i,j). Thus, cluster 1 represented the lowest differentiated CAR T cell cluster in this analysis, followed by clusters 5 and 8 (Fig. 5k). These clusters were notably reduced in BBζ (clustering 20% of all cells) but enriched in BBδ CAR T cells (clustering 46% of all cells), providing a probable explanation for the differential long-term efficacy in vivo.

### Dimeric CARs improve overall functionality

The ζζ homodimer is a natural module of the TCR–CD3. In contrast, the CD3 chains are not naturally found as homodimers, but form CD3ε–CD3δ and CD3ε–CD3γ heterodimers. Our CAR constructs contain the extracellular hinge region of human CD8α (Extended Data Fig. 1a), holding two cysteines that drive the formation of homodimers. To evaluate the impact of homodimerization, we mutated both cysteines to serines (Fig. 6a and Extended Data Fig. 7a). Monomeric and dimeric BBζ CARs were equally well expressed, whereas surface expression of monomeric BBγ and BBδ CARs was significantly increased compared with the dimeric versions (Fig. 6b,c). Monomeric BBζ CARs maintained functionality in regard to specific killing and degranulation, as well as CD25, CD69 and 4-1BB upregulation on coincubation with tumor cells (Fig. 6d–f). However, cytokine secretion was significantly decreased in monomeric BBζ CARs (Fig. 6g,h). Dimeric BBγ and BBδ CARs are more efficient at transmitting signals than their monomeric counterparts, because they promote a similar level of activation despite lower expression (Fig. 6d–h).

The CD3γ ICD has a membrane proximal di-leucine motif that is required, along with the phosphorylation of a closely located serine, for TCR internalization and recycling on antigen-induced TCR signaling[16]. The CD3δ ICD also contains a membrane proximal di-leucine motif; however, it lacks the preceding serine (Extended Data Fig. 1a and Fig. 6a). Di-leucine mutants, named ΔLL, resulted in significantly increased CAR expression, overriding the differences between monomeric and dimeric BBγ and BBδ CARs (Fig. 6b,c). Dimeric and monomeric ΔLL mutants displayed similar tumor cell killing, but slightly reduced degranulation and CD25 upregulation (Fig. 6d–f). Conversely, dimeric ΔLL mutants were more efficient in upregulating CD69 and 4-1BB, and showed a tendency toward enhanced cytokine secretion on activation (Fig. 6f–h).

**Fig. 5 | BBδ CAR T cells maintain precursors expressing TCF-1. a**, UMAP visualization from CyTOF data of CAR$^+$ T cells on serial antigen encounters. Color indicates TCF-1 expression ($n = 3$). **b,c**, Frequency of TCF-1-expressing CAR T cells assayed by CyTOF (**b**) ($n = 3$) and by intracellular flow cytometric analysis (**c**) ($n = 3$). **d**, FlowSOM clustering of CAR T cells. Clusters are visualized by the indicated colors on the UMAP plot. **e**, Stack bar graph displaying the median frequency of each cluster. **f**, Cluster frequency of selected clusters ($n = 3$). **g**, Hierarchically clustered heatmap indicating the percentage of expression of activation and exhaustion markers per cluster. **h**, Heatmap showing the percentages of cytokine-expressing CAR T cells per cluster ordered by increasing FES. **i**, Wanderlust trajectory analysis, with progressing trajectory pseudotime color visualized on the UMAP. **j**, PD-1 (top) and TCF-1 (bottom) expression according to the Wanderlust trajectory. **k**, Stacked bar histogram indicating the position of the indicated cluster cells according to the trajectory. Each dot represents an independent donor (**b**, **c** and **f**). Data are represented as mean ± s.d. One-way ANOVA followed by Dunnett's multiple-comparison test was used.

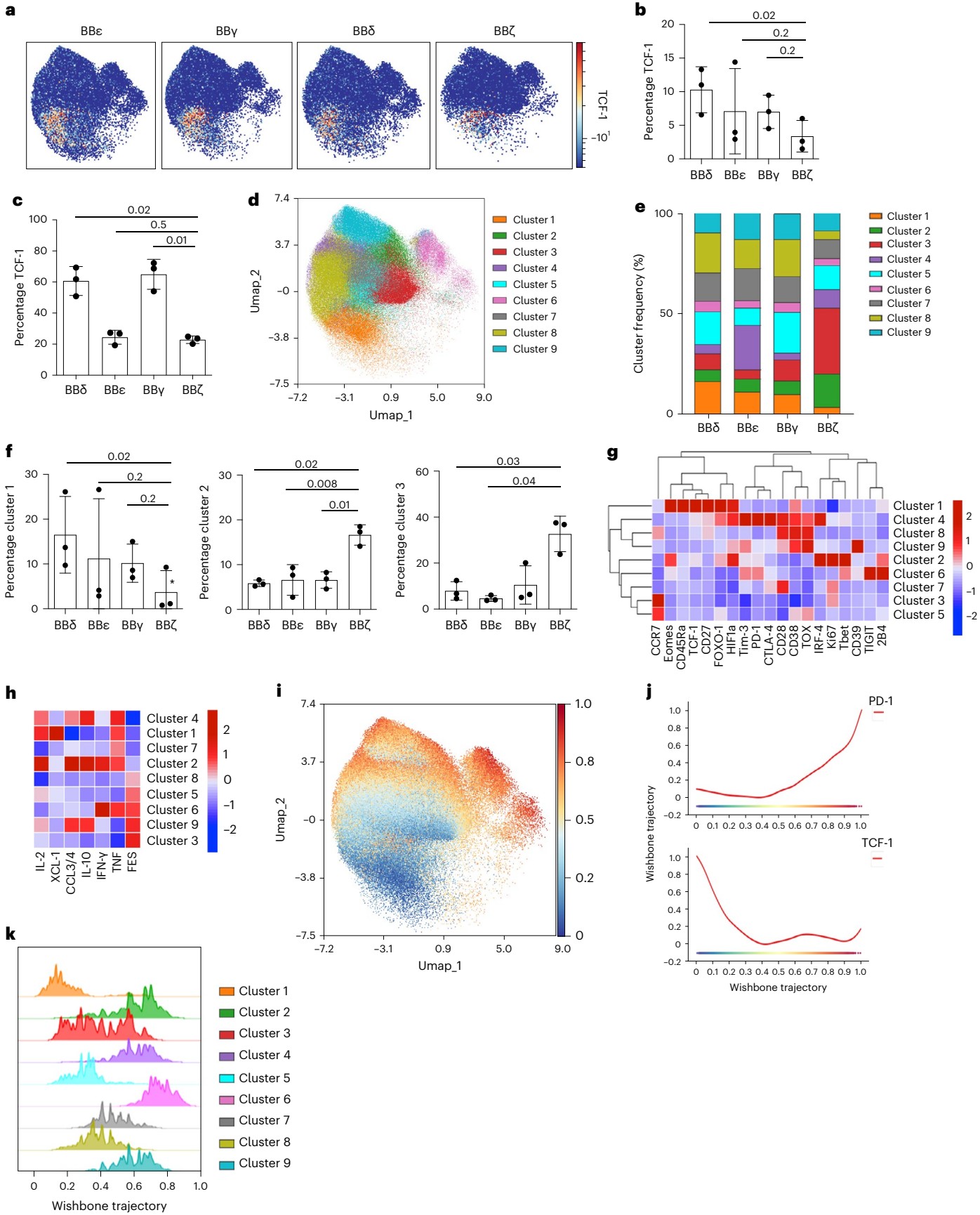

In sharp contrast, monomeric BBε CARs failed to be expressed and, consequently, to mediate tumor cell killing or cytokine secretion (Extended Data Fig. 7a–e). The CD3ε-ICD contains an endoplasmic reticulum retention (ERR) motif that includes the last five amino acids of CD3ε (NQRRI) and controls TCR surface expression[29,30] (Extended Data Fig. 7a). Deletion of the ERR motif failed to recover surface expression of monomeric BBε (Extended Data Fig. 7g,h), suggesting the existence of additional retention signals in CD3ε that are overridden only after the formation of dimers. Altogether, dimeric CARs improve the functionality for all CARs, best in the context of ΔLL mutants for BBδ and BBγ, by mechanisms that are ICD specific, highlighting the uniqueness of each TCR–CD3 chain.

## Motifs recruiting Lck enhances BBε CAR functionality
We previously reported that combining CD3ε and ζ ICD into a BB-based CAR improved tumor therapy[14]. However, the CD3ε ICD itself is sufficient to generate a functional CAR that outperformed the FDA-approved BBζ CAR in vivo (Fig. 1). The CD3ε ICD possesses the highest number of known signaling motifs. We generated CARs with individual mutations in the BRS, PRS, RK and ITAM, or with double mutations in PRS/RK and BRS/RK, to study their role without possible compensation effects within the TCR–CD3 (Fig. 7a). These mutations did not affect the percentage of CAR⁺ T cells (Fig. 7b) or CAR expression (Fig. 7c). T cells expressing BBεΔRK, BBεΔITAM and BBεΔBRS presented reduced killing compared with BBε cells, suggesting that these motifs are important to activate cytotoxicity. In contrast, cells expressing BBεΔPRS did not show any diminished killing capacity. These data suggest that the PRS–Nck axis is not relevant for the cytotoxic response, at least in the CAR context, because we have mutated the two prolines responsible for Nck recruitment. It is interesting that BBεΔBRSΔRK has an even lower specific killing, indicating that both the BRS and the RK motifs are required to optimally activate T cells, without having redundant functions. In contrast, the killing potential of CAR T cells expressing the double mutant BBεΔPRSΔRK was similar to that of cells expressing BBεΔRK, supporting the idea that the PRS is not involved in cytotoxicity (Fig. 7d). Both the RK and the BRS motifs are important for CAR T cell activation and sensitivity as shown by evaluating the proportion of CAR T cells fluxing calcium (Fig. 7e). Likewise, T cells expressing BBεΔRK, BBεΔITAM, BBεΔBRS or BBεΔBRSΔRK mutants were less activated on coincubation with tumor cells (Fig. 7f). Hence, in addition to the ITAM, the CD3ε motifs responsible for Lck recruitment (BRS and RK) are crucial to ensure T cell activation and cytotoxicity in the context of CARs.

## Monophosphorylated BBδ recruits SHP-1
CD3γ and CD3δ diverged only recently in evolution, have a similar sequence and length and harbor a single ITAM[31]. Thus, it was surprising that BBδ outperformed BBγ in vivo (Fig. 1g). Aside from the di-leucine motif in CD3γ, neither specific motifs nor unique interaction partners have been reported for these two chains. Therefore, we aimed to identify new interaction partners of CD3δ and CD3γ. As surrogates of these ICDs, we used biotinylated peptides that were either non-phosphorylated or phosphorylated on both ITAM tyrosines (Fig. 8a). T cells were subjected to stable isotope labeling in cell culture (SILAC) and the lysates were individually incubated with the CD3δ- or CD3γ–ICD-derived peptides. Peptide-bound proteins were isolated, heavy and

light samples were pooled and interacting proteins were identified by tandem mass spectrometry (MS–MS). ZAP70 serves as a control. We found that ZAP70 bound equally well to both phosphorylated peptides and did not bind to unphosphorylated ones (SILAC ratio 1.06; Fig. 8b). A previous report has identified the phosphatase Src homology 2-containing protein tyrosine phosphatase-1 (SHP-1) among a list of different SH₂-containing proteins that coimmunoprecipitated with the TCR[32]. However, SHP-1 was not found in our initial analysis. Using monophosphorylated peptides in which only either the C-terminal or the amino-terminal tyrosine was phosphorylated, we found that SHP-1 bound better to the phosphorylated C-terminal tyrosine of the CD3δ ITAM compared with the N-terminal tyrosine, and did not bind to monophosphorylated CD3γ peptides (Fig. 8c). Immunoblot analysis confirmed these results (Fig. 8d). To rule out confounding effects due to binding competition with ZAP70, peptides and beads were used in vast molecular excess.

We next generated new CAR constructs containing a mutation of the N-terminal tyrosine in the ITAM to phenylalanine in the ICD of both CD3γ and CD3δ, so that these CARs can be phosphorylated only at the C-terminal tyrosine (BBγFY and BBδFY; Fig. 8e). To ensure optimal phosphorylation, we stimulated Jurkat (Fig. 8f) and primary human T cells (Fig. 8g) expressing these CARs with the phosphatase inhibitor pervanadate and performed a CAR immunoprecipitation (IP) to evaluate SHP-1 binding. SHP-1 preferentially bound to the CARs containing monophosphorylated BBδ independently of the dimeric or monomeric format (Fig. 8f,g). These results identify SHP-1 as a new binding partner of CD3δ. The recruitment of SHP-1 to CD3δ might fine-tune and balance T cell activation, preventing exhaustion and dysfunction.

## Discussion
A detailed mechanistic understanding of CAR T cell activation is essential to improve CAR immunotherapy. Current CAR designs incorporate the ζ ICD[4]. CD3δ/ε/γ makes a important contribution to T cell signaling within the TCR–CD3; however, its potential to improve CAR performance has not been thoroughly explored. It is challenging to study the individual roles of each of the ICDs of TCR–CD3 by genetic means owing to the risk of alterations in the assembly and expression of TCR–CD3 and/or compensation mechanisms. CARs can be seen as minimalistic TCRs that allow us to study the signaling properties of each TCR–CD3 chain. Using a 4-1BB-based CAR, we have systematically exchanged the ζ ICD to each of the CD3 chains to gain insights into their role on T cell activation and therapeutic potential. All CD3 ICDs generated functional CAR T cells that outperformed the ζ-based CAR in vivo. An obvious difference between ζ and CD3δ/ε/γ is that the latter only has one ITAM, whereas ζ has three, suggesting that one ITAM is sufficient and even beneficial for the functionality of CAR T cells in vivo. Consistent with our results, mutations in the ζ ITAMs resulting in single ITAM-containing ζ CARs improved performance in vivo[33]. Furthermore, introduction of the CD3ε ICD into ζ CARs is beneficial for preclinical CAR T cell therapy[6,14,32]. In contrast, we have studied the contribution of each individual CD3 ICD independently of the ζ ICD and demonstrated that the presence of the ζ ICD is not essential to generate functional CAR T cells. In the TCR–CD3, only the ζ chains form homodimers. Our CARs contain an extracellular portion of human CD8α, which holds two cysteines that drive the formation

**Fig. 6 | Dimeric CARs displayed improved functionality. a**, Scheme of monomeric and dimeric CARs and the mutations introduced. **b,c**, Flow cytometric analysis depicting the percentage of positive CAR T cells (**b**) and quantification of surface CAR expression for GFP⁺ or BFP⁺ cells (**c**) (*n* = 8, except for monomer ΔLL: *n* = 5). **d**, Specific killing of CD19⁺ Nalm6 cells by primary human CAR T cells (1:1 ratio) for 12 h (*n* = 8, except for BBζ and monomer ΔLL: *n* = 5). **e**, Flow cytometry-based analysis of degranulation. Statistical analysis of the mean fluorescence intensity of CD107a is shown (*n* = 4). **f**, Activation markers

on CAR T cells 24 h after stimulation with Nalm6 (1:1 ratio) (*n* = 4). **g,h**, Cytokine secretion, IFN-γ (**g**) and TNF (**h**), assayed by ELISA of CAR T cells on incubation with Nalm6 cells at a 1:1 ratio for 24 h (*n* = 5, except for monomer ΔLL: *n* = 2). Data are represented as mean ± s.d. Each dot represents an independent donor. In **c**, **d**, **g** and **h**, two-tailed, paired Student's *t*-test for BBζ and paired one-way ANOVA followed by Holm–Sidak multiple-comparison test for BBγ and BBδ were used. In **e** and **f**, paired two-way ANOVA followed by the Holm–Sidak multiple-comparison test was used.

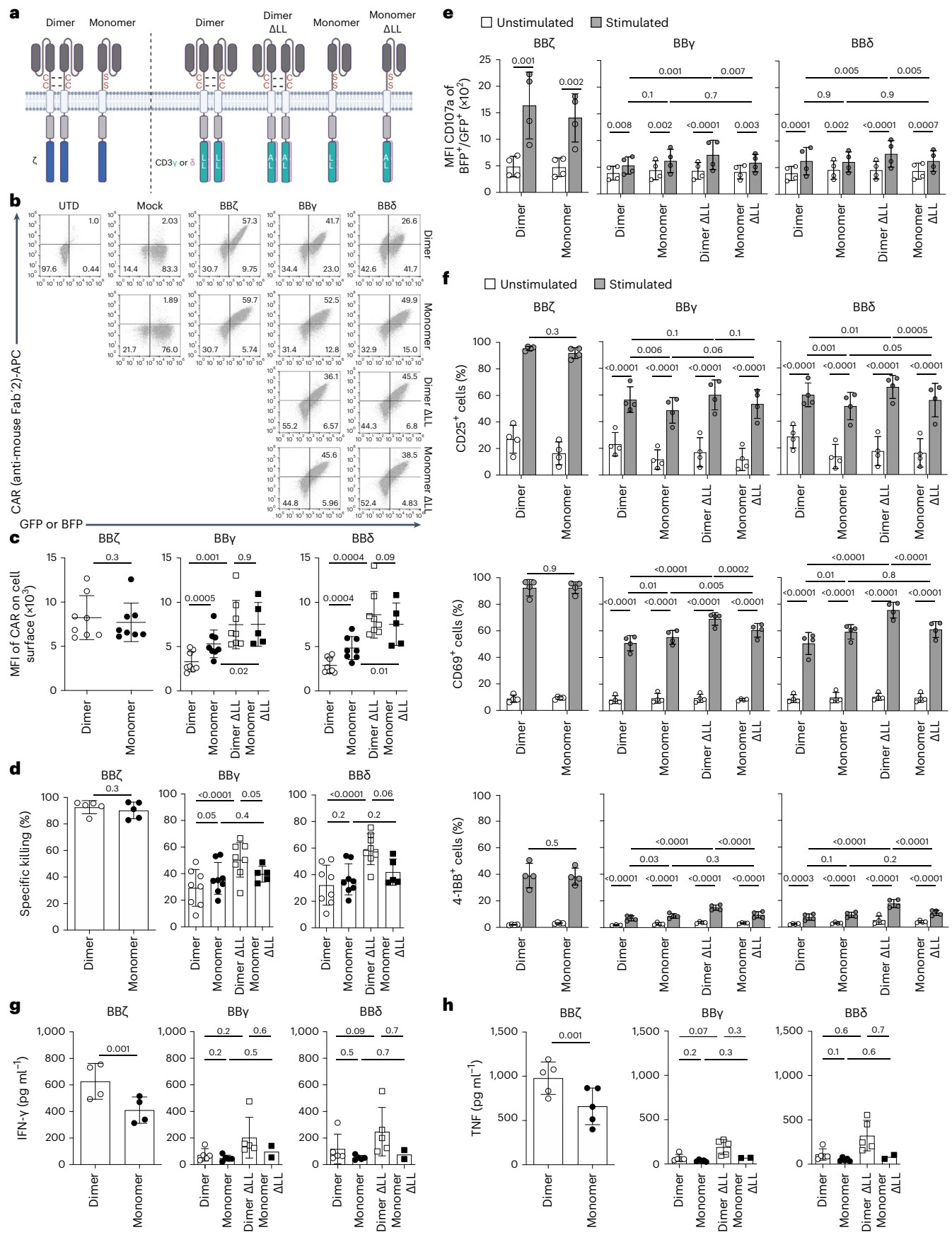

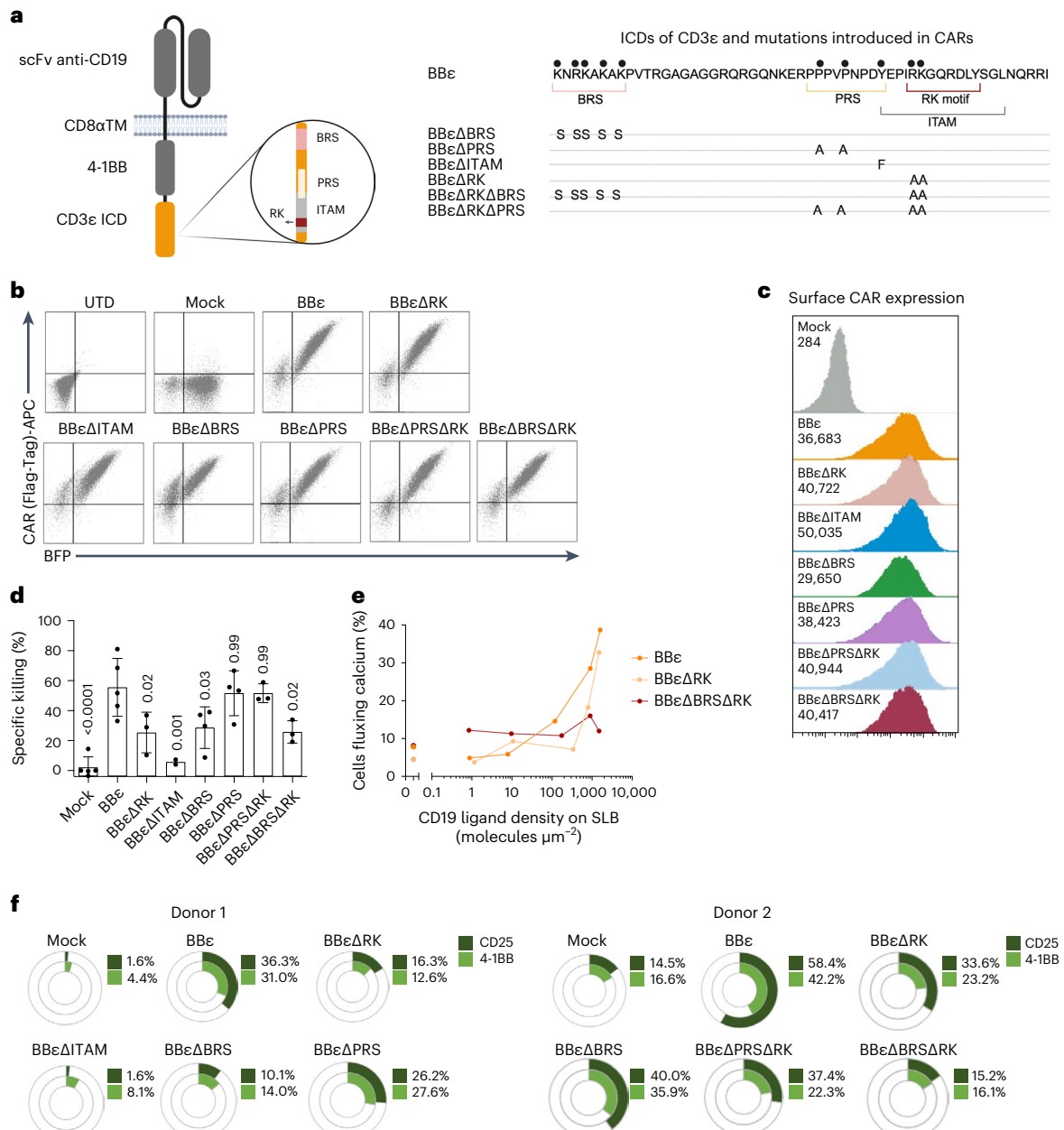

**Fig. 7 | The enhanced functionality of the BBε CAR relies on motifs recruiting Lck. a**, Schematic representation of the mutations introduced in BBε. **b,c**, Flow cytometric analysis (**b**) and surface CAR expression (**c**) gated on BFP⁺ cells (*n* = 5). **d**, Specific killing of CD19⁺ Nalm6 cells by CAR T cells (1:1 ratio for 6–8 h). Each dot represents an independent donor (*n* = 5 for BBε, *n* = 4 for BBεΔBRS and BBεΔPRS, *n* = 3 for BBεΔRK, BBεΔPRSΔRK and BBεΔBRSΔRK, and *n* = 2 for BBεΔITAM). Data are represented as mean ± s.d. One-way ANOVA followed by Dunnett's multiple-

comparison test was used. **e**, Percentage of CAR T cells fluxing calcium after contacting SLBs functionalized with adhesion and costimulatory molecules as well as CD19 molecules at indicated densities. Data represent one out of three independent experiments done with three independent donors. **f**, Percentage of CAR T cells positive for activation markers after 48 h of stimulation with CD19⁺ Nalm6 cells (1:1 ratio). Two representative donors are shown.

of homodimers. Homodimerization of ζ, but also of the other CD3 chains, enhanced CAR functionality by mechanisms that are ICD specific. Indeed, we gained insights into nonredundant functions of each TCR–CD3 ICD beyond the number of ITAMs. We show that the BRS and the RK motifs, which mediate Lck recruitment to the CD3ε ICD, are not redundant and both are necessary for optimal CAR efficacy and receptor sensitivity. In both the TCR and the CAR context, Lck initiates signaling by phosphorylating the ITAMs[34,35]. However, the recruitment of Lck to these receptors must be well balanced. For example, differences in signal intensity and therapeutic efficacy between 28ζ and BBζ CARs are related to higher Lck association via the CD28 ICD[36].

Despite the differences between the ICDs of CD3ε and ζ, BBε and BBζ CARs revealed very similar responses and gene profiling after stimulation with tumor cells. However, BBζ secreted the highest levels of cytokines, many of which have been associated with CRS, a major complication of CAR T cell therapy that inversely correlates with clinical outcome[37–40]. Our results suggest that designs incorporating the ICD of CD3δ/ε/γ might reduce the adverse effects associated with existing ζ-based therapies. BBζ CAR T cells displayed higher expression of genes associated with apoptosis. In contrast, BBε CAR T cells showed a gene signature associated with T cell persistence. This might be related to the specific recruitment of Csk to CD3ε that self-restrains signaling,

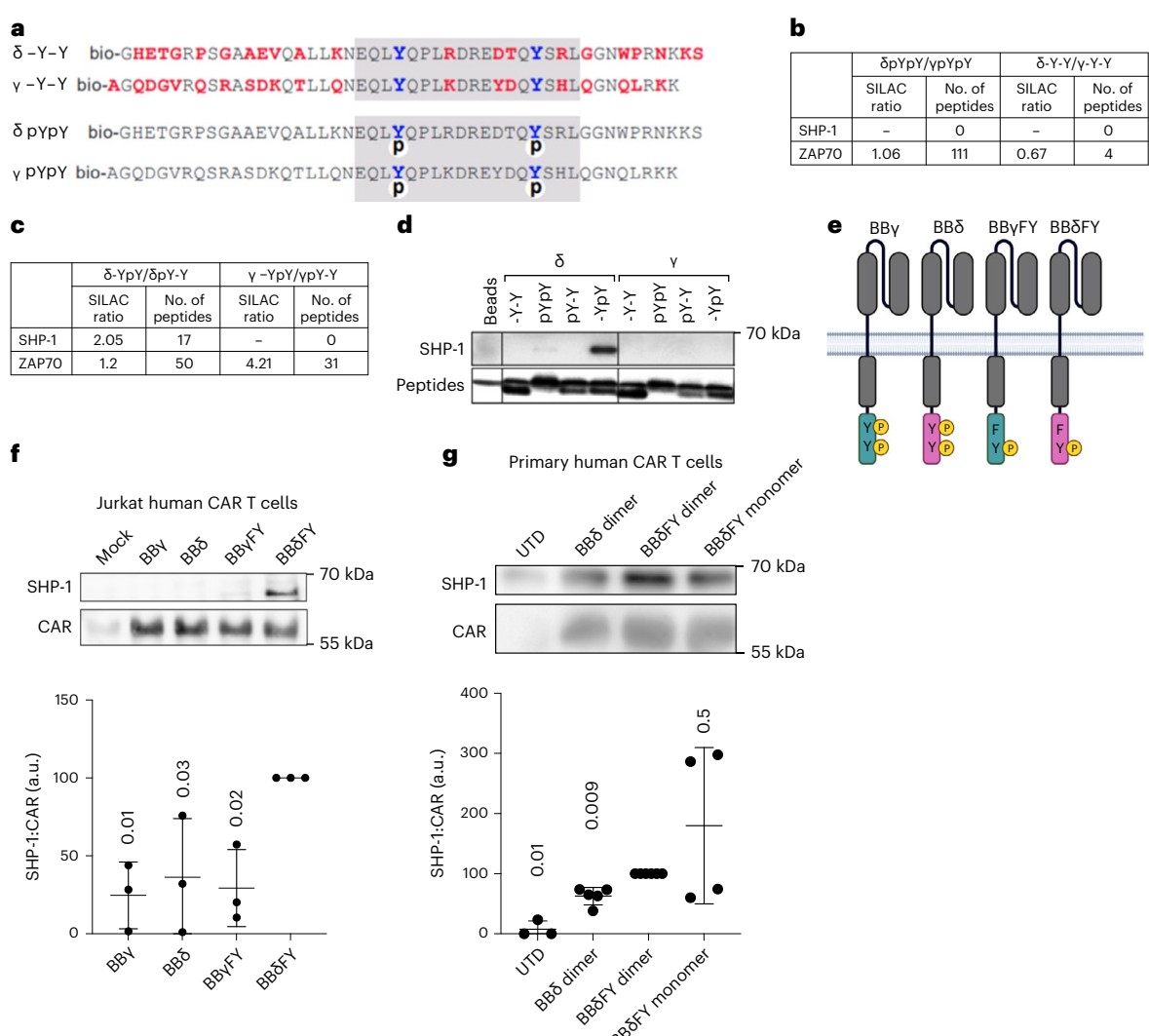

**Fig. 8 | Monophosphorylated CD3δ recruits SHP-1. a**, Peptide sequences corresponding to the cytoplasmic tails of mouse CD3δ and CD3γ. Red amino acids are not conserved between CD3δ and CD3γ; ITAMs are marked in gray, phosphates are indicated with a circled 'p' and N-terminal biotin with a 'bio'. **b,c**, SILAC ratio and the number of identified peptides for SHP-1 (**b**) and ZAP70 (**c**) on quantification by MS–MS. T cell lysates were incubated with doubly phosphorylated, unphosphorylated or singly phosphorylated peptides. The SILAC ratio indicates the relative amounts of a protein bound to one peptide in comparison to the amounts of the same protein bound to the other peptide. **d**, The experiments in **b** and **c** were repeated, purified proteins were separated

by SDS–PAGE and visualized using immunoblotting (*n* = 2). **e**, Schematic representation of the CARs with a mutation in the N-terminal tyrosine leaving functional just the C-terminal tyrosine. **f,g**, Jurkat (*n* = 3) (**f**) and primary human T cells (*n* = 3 for UTD, *n* = 5 for BBδ dimer, *n* = 6 for BBδFY dimer and *n* = 4 for BBδFY monomer) (**g**). Each dot represents a healthy donor transduced with the indicated CARs. Cells were stimulated with pervanadate for 5 min to achieve maximum phosphorylation and the CARs were immunoprecipitated. Purified proteins were separated by SDS–PAGE and visualized using immunoblotting. The ratio of SHP-1 and CAR was calculated. Data are represented as mean ± s.d. One-way ANOVA followed by Dunnett's multiple-comparison test was used.

favoring T cell persistence and preventing exhaustion[32]. In line with the idea of the beneficial effect of restraining CAR signaling, recruitment of THEMIS–SHP-1 to 4-1BB sequences attenuates CAR T cell exhaustion[41]. Indeed, BBζ CAR T cells are characterized by lower expression of exhaustion markers, more central memory T cell polarization and slower, but more persistent, tumor eradication than 28ζ CAR T cells[42,43]. The phosphatase SHP-1 reduces activation of Src-family kinases, such as Lck and Fyn[44], and dephosphorylates ZAP70 (ref. [45]). In the present study, we demonstrated that SHP-1 is also recruited to the CD3δ ICD when the latter is monophosphorylated in the second tyrosine. It is of interest that SHP-1 recruitment to the doubly phosphorylated CD3δ ICD was less efficient, suggesting structural changes in the ITAM and/or long-range effects of the first phosphate group. BBδ CAR T cells secreted fewer cytokines and showed the lowest expression of activation and exhaustion/dysfunction markers on stimulation. Altogether,

these results call for a paradigm shift in the design of next-generation CARs such that they deliver more balanced signals instead of just increasing the signaling potency. Our study contributes to mounting evidence suggesting that TCR–CD3 is a self-restrained piece of signaling machinery containing both activating and inhibitory motifs that fine-tune T cell activation[5,32,46].

An additional limitation to the design of CARs for immunotherapy is the lack of high-throughput, cost-effective and reproducible ex vivo approaches to predict CAR efficacy in vivo. In the present study, we have established a serial challenging protocol that recapitulates the in vivo results and allows examination of T cell dysfunction. Using our protocol, CAR T cells, regardless of which ICD was included, that were repeatedly stimulated with tumor cells secreted significantly lower levels of proinflammatory cytokines and upregulated the expression of IL-10 and the exhaustion-related transcription factor TOX.

When each ICD was analyzed separately, BBδ CAR T cells retained the best killing efficacy, showed the lowest PD-1 expression and maintained the highest population of cells expressing TCF-1, which is a key transcription factor for the formation of memory T cells and associated with self-renewing stem cell-like properties[25,26]. In fact, cells with stem cell memory or central memory phenotype are beneficial and exert a better anti-tumor activity in adoptive T cell therapies[47–49]. In support of this, BBδ-expressing T cells showed expression signatures consistent with reduced glycolysis and increased mitochondrial metabolism, thus differentiating less to effector cells and instead keeping memory-like properties. We propose that CARs containing the CD3δ ICD transmit self-restrained signals that favor self-renewing properties while preventing dysfunction, thus explaining their superior anti-tumor efficacy in vivo.

Our data demonstrate the promise of harnessing the signaling diversity of the CD3 chains, rather than simply their signaling strength, to improve CAR T cell therapy. Our study also contributes to a greater understanding of CAR and TCR–CD3 signaling, including how unique signaling motifs influence expression profiles, metabolism, phenotype and functionality.

## Online content

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

[1]Faculty of Biology, University of Freiburg, Freiburg, Germany. [2]Signalling Research Centres BIOSS and CIBSS, University of Freiburg, Freiburg, Germany. [3]Spemann Graduate School of Biology and Medicine, University of Freiburg, Freiburg, Germany. [4]Clinic for Internal Medicine II, Medical Center, Faculty of Medicine, University of Freiburg, Freiburg, Germany. [5]Center for Pathophysiology, Infectiology and Immunology, Institute for Hygiene and Applied Immunology, Medical University of Vienna, Vienna, Austria. [6]Institute for Surgical Pathology, Medical Center, Freiburg, Germany. [7]Wolfson Wohl Cancer Research Centre, Institute of Cancer Sciences, College of Medical, Veterinary and Life Sciences, University of Glasgow, Glasgow, UK. [8]NanoString Technologies, Inc., Seattle, WA, USA. [9]Department of Medicine I, Medical Center, Faculty of Medicine, University of Freiburg, Freiburg, Germany. [10]Faculty of Medical and Life Sciences, University of Furtwangen, Freiburg, Germany. [11]Department of Biology, Institute of Molecular Systems Biology, ETH Zürich, Zürich, Switzerland. [12]Institute of Medical Bioinformatics and Systems Medicine, Medical Center, Faculty of Medicine, University of Freiburg, Freiburg, Germany. [13]German Cancer Consortium and German Cancer Research Center, Freiburg, Germany. [14]Center of Chronic Immunodeficiency, University Clinics and Medical Faculty, Freiburg, Germany. [15]These authors contributed equally: Ana Valeria Meléndez, Alexandra Emilia Schlaak. [16]These authors jointly supervised this work: Wolfgang W. Schamel, Susana Minguet. ✉e-mail: Susana.minguet@biologie.uni-freiburg.de

## Methods

### Generation of CAR constructs

The lentiviral vector pCDH-EF1-19BBζ-T2A-copGFP (coding for BBζ CAR) was a gift from TCR[2] Therapeutics. The ζ chain was replaced for the ICDs of CD3δ, CD3ε or CD3γ. In addition, a Strep-Tag II or a Flag-Tag was placed between the scFv and the CD8α hinge region for flow cytometric detection. All the CARs contained an anti-human CD19 scFv from murine monoclonal antibody origin (FMC63), a tag (Strep-Tag II or Flag-Tag), an extracellular hinge and a transmembrane domain from human CD8α (amino acids 138–206), a costimulatory 4-1BB ICD and the respective CD3 tail (wild-type or with the indicated mutation) followed by a T2A peptide and copGFP or mTAGBFP2 to serve as a fluorescent marker to measure transduction efficiency. Monomeric and L153A (BBγ) and L152A (BBδ) CAR mutants were generated by introducing specific point mutations via PCR. Deletion of the ERR motif was generated by specific primers omitting the last five amino acids (NQRRI) of the CD3ε ICD. Final plasmids were generated by either Gibson Cloning[50] or restriction enzyme digestion and subsequent ligation. The integrity of the plasmids was verified by test digest and sequences were verified by Sanger sequencing (Eurofins).

### Lentiviral production

Polyethylenimine transfection of HEK293T cells (American Type Culture Collection (ATCC), catalog no. CRL-1573) was performed as previously described[14]. On harvesting the supernatant from transfected HEK293T cells, virus was concentrated with a solution of 10% sucrose for 4 h at 10,000$g$. Virus preparations were titrated to obtain the transducing units (TU ml$^{-1}$).

### Primary human T cell activation, transduction and expansion

Buffy coats from healthy donors (ethics approval no. 22-1275-S1) were used to isolate peripheral blood mononuclear cells (PBMCs) by density centrifugation (Pancoll). The cells were activated with anti-CD3 and anti-CD28 antibodies (1 μg ml$^{-1}$) and 500 U ml$^{-1}$ of human IL-2 for 48–72 h. T cell purity and activation were verified by flow cytometry. Primary human T cells were lentivirally transduced with 5 μg ml$^{-1}$ of proteamine sulfate in the presence of 500 U ml$^{-1}$ of human IL-2 using a multiplicity of infection of 4 by spin infection (652$g$, 30 °C, 90 min). CAR T cells were maintained with RPMI-1640 medium plus 5% fetal calf serum (FCS), Hepes, pH 7, penicillin–streptomycin (Pen–Strep), 2-mercaptoethanol and 100 U ml$^{-1}$ of human IL-2. CAR expression was verified after 5–7 d with an anti-Strep-Tag II, anti-Flag-Tag II or anti-mouse Fab'2 antibody and experiments started at the indicated time point.

### Cell lines and isotope labelling

CD19-expressing Nalm6 tumor cells expressing firefly luciferase (a gift from TCR[2] Therapeutics) were used in the present study. Human Jurkat T cells (ATCC, catalog no. TIB-152) were lentivirally transduced with the CARs to perform IP and sodium dodecylsulfate–polyacrylamide gel electrophoresis (SDS–PAGE)/immunoblotting. The murine T cell hybridoma 2B4 was used to characterize SHP-1. All these cells were maintained with RPMI-1640 medium plus 5% FCS, Hepes, pH 7, Pen–Strep and 2-mercaptoethanol. For the stable isotope labeling of cells with heavy amino acids (SILAC), 2B4 T cells were grown in RPMI-1640 medium without arginine and lysine (Perbio Science Deutschland, catalog no. 89984) supplemented with 5% dialyzed FCS, 50 mg l$^{-1}$ of heavy ($^{13}$C) or light ($^{12}$C) arginine, 50 mg l$^{-1}$ of heavy ($^{15}$N) or light ($^{14}$N) lysine, and Pen–Strep was used.

### In vivo animal studies

Rag2$^{-/-}$γc$^{-/-}$ (Rag2tm1.1Flv IL-2rgtm1.1Flv) mice were originally purchased from Jackson Laboratory and bred at the Center for Experimental Models and Transgenic Service, Freiburg, under specific pathogen-free conditions. Both adult (>8 weeks old) females and males were used. The sample size was calculated using the following parameters: 1.06 effect size, 5% significance level, 80% power and 1.06 s.d. Then 500,000 Nalm6 cells were injected intravenously (i.v.) 3 d before CAR T cell injection. Mice were randomly distributed to the experimental conditions. CAR T cells, 1,500,000, were injected i.v. per mouse after expansion using an adapted rapid expansion protocol (REP)[51]. All animal protocols (G18/03) were performed according to the German animal protection law, with permission from the Veterinär und Lebensmittelüberwachungsbehörde Freiburg. The maximal tumor burden was not exceeded.

### Cytotoxicity assay

To measure specific killing, a bioluminescence (BLI)-based cytotoxicity assay was performed. Nalm6 cells (CD19$^+$), 3 × 10$^4$, expressing firefly luciferase were plated on white, 96-well, flat-bottomed plates supplemented with 75 μg ml$^{-1}$ of D-luciferin firefly substrate (Biosynth) in complete RPMI medium. The BLI baseline was measured in a luminometer (Tecan infinity M200 Pro or BioTek Synergy 4). CAR T cells were plated at the indicated ratio and cells were incubated at 37 °C. The BLI signal was measured at the indicated time points in relative light units (RLU). A spontaneous death control was used including Nalm6 cells alone. The maximal death control contained 1% Triton X-100 and Nalm6 cells. Specific lysis was calculated using: percentage specific lysis = 100 × (average UTD death RLU − test RLU)/(average UTD death RLU − average maximal death RLU); UTD, untransduced cells. For blocking experiments, Nalm6 tumor cells were preincubated with 5 μg ml$^{-1}$ of the indicated blocking antibodies anti-CD95 (BioLegend, catalog no. A16086F or eBioscience, catalog no. SM1/23), anti-DR5 (BioLegend, catalog no. DJR2-4(7-8)) or isotype (BioLegend, catalog no. MG2b-57) at 37 °C for 30 min. Afterwards, CAR T cells were cocultured with the preblocked Nalm6 tumor cells in the presence of 2.5 μg ml$^{-1}$ of blocking antibodies. Specific tumor cell lysis was measured as described above. To evaluate the impact of the cytotoxic pathway inhibition on the ability to lyse Nalm6 tumor cells, samples were normalized to the isotype control (set to 100%).

### Degranulation assay

Degranulation of primary human CAR T cells was measured by the upregulation of CD107a. CAR T cells, 1 × 10$^5$, were cocultured with 1 × 10$^5$ Nalm6–CD19$^+$ tumor cells in the presence of 1 μl of anti-CD107a-PE antibody (BioLegend). Cells were harvested at the indicated time points and analyzed by flow cytometry.

### NanoString analysis (RNA)

Primary human T cells were transduced with the different CAR constructs. After 6 d of transduction, 5–6 × 10$^6$ CAR T cells were stimulated 24 h with Nalm6 cells in a 1:1 ratio or maintained without stimulation. Then, 500,000 CAR T cells were sorted for GFP$^+$ and RNA was isolated using the QIAGEN RNeasy kit. The quality of the RNA was verified. Code-set probes were hybridized with RNA for 19 h at 65 °C, subsequently loaded into nCounter MAX cartridges and run on the nCounter MAX/FLEX according to NanoString protocols. Then, nCounter gene expression assays (NanoString Technologies) were performed using NanoString XT CAR-T Panel Standard. The resulting data were analyzed using nSolver 4.0 software. Downstream bioinformatic analysis was performed with R (4.2.1). Briefly, the differentially regulated genes were identified using the limma R package[52], with a paired design (donor based). Adjusted $P$ value (Benjamini–Hochberg) <0.05 was considered to be significant. The GAGE[53] R package was used to identify regulated gene sets among the whole MSigDB[54] repository and the NanoString panel gene sets. $P$ < 0.05 was considered to be significant.

### ELISA

Supernatant from CAR T cells cocultured for 24 h with Nalm6 cells was collected. IFN-γ, TNF, GM-CSF and IL-2 human uncoated ELISA Kit (Invitrogen) was used.

## Protein production and labeling and SLB preparation

The extracellular portion of CD19, ICAM-1 and B7-1, all harboring 12 consecutive histidine residues, was produced as described[21]. The poly-histidine tag interacts with 18:1 DGS-NTA(Ni) present in the SLB. CD19 was fluorescently labeled using the *N*-hydroxysuccinimide ester derivative of Alexa Fluor-647 (Thermo Fisher Scientific) as described[21]. SLBs were prepared as described[55]. Microscopy was conducted using two inverted setups. One setup (Eclipse Ti-E, Nikon Instruments) allowed for TIR-based imaging and was equipped with a chromatically corrected ×100 TIR objective (CFI SR Apo TIR ×100 oil, numerical aperture: 1.49; Nikon Instruments), a 647-nm diode laser (OBIS) for excitation light and a customized Notch filter (Chroma Technology) to block reflected stray light of 647 nm from reaching the camera. Furthermore, this microscope featured an ET700/75 emission bandpass filter (Leica) present in the emission pathway. An iXon Ultra 897 EMCCD camera (Oxford Instruments) was used for data recording. An eight-channel DAQ-board PCI-DDA08/16 (National Instruments), in combination with the microscopy automation and image analysis software MetaMorph v.7.8.13.0 (Molecular Devices), was used to program and apply timing protocols and control all hardware components of the microscope components. A second inverted microscope (Leica Microsystems, catalog no. DMI4000) was equipped with a ×20 objective (HC PL FLUO-TAR ×20/0.50 PH2∞/0.17/D; Leica Microsystems) and a mercury lamp (Leica Microsystems, catalog no. EL6000) for Fura-2-based calcium recordings. This microscope was equipped with a fast filter wheel containing 340/26 and 387/11 excitation bandpass filters (both Leica Microsystems). Data were recorded using a sCMOS Andor Prime95b (Photometrix). Open-source software Micromanager was used to program and control all hardware components.

## Measurements of antigen densities on SLBs

SLB antigen density was determined by counting the number of diffraction-limited fluorescent events within a region of interest (ROI) or by dividing the fluorescence intensity value within an ROI by the single-molecule fluorescent intensity value. For bilayers where fluorescent signals were clearly distinguishable, 30 images were recorded within an ROI of 100 × 100 pixels. The total number of molecules within the ROI of each image was determined using Fiji Thunderstorm plugin (ImageJ/Fiji) and corrected for pixel size and number of images to determine the antigen density (1 pixel $\triangleq$ 0.0256 μm$^2$, 100 × 100 pixels = 10,000 pixels $\triangleq$ 256 μm$^2$). For determining the antigen density of SLBs with crowded antigen densities, the average intensity value of at least 300 single-molecule fluorescence events within the ROI was determined using the Fiji Thunderstorm plugin as described above. The average integrated intensity value of ROIs of ten images was determined and divided by the average single-molecule intensity value to arrive at the number of molecules in the chosen ROI. This value was corrected for pixel size to determine the antigen density.

## Calcium imaging

Intracellular changes in Ca$^{2+}$ levels were measured using Fura-2-AM. A total of $9 \times 10^5$ cells was incubated in 0.5 ml of imaging buffer (Hanks' balanced salt solution, supplemented with 2 mM CaCl$_2$, 2 mM MgCl$_2$ and 2% FCS) supplemented with 5 μM Fura-2-AM for 15 min at 37 °C, washed twice with 10 ml of imaging buffer and resuspended in 135 μl of imaging buffer. Cells were kept at 20 °C for a maximum of 30 min before starting the experiment. Right before imaging, the SLB buffer was exchanged for an imaging buffer prewarmed to 37 °C. T cells were pipetted into the imaging buffer and allowed to sink for 30 s, after which 510/80-nm emission was recorded with alternating 340-nm and 387-nm excitation every 15 s for 20 min. An inhouse customized Matlab software was used to track cells in each frame, using a particle-tracking algorithm published by Gao and Kilfoi[56]. Tracking parameters were chosen so only single cells in contact with the SLB were included. We used the Matlab software to create ratio images for each frame. Methods for automated

and accurate analysis of cell signals (MAACS) were used for population analysis as described[57]. For each trajectory within a population, the ratio was normalized frame-wise to that of the population median of T cells in contact with antigen-free SLBs. Cells that were above the threshold for at least 80% of their trajectory were counted as activated and plotted in a dose–response curve. The calcium histograms were compiled from the measured population values of the median Fura-2-AM ratio, corresponding to the first ten frames after the peak Fura-2-AM ratio value within the trajectory. The latter was normalized frame-wise to the population median of the negative control, that is, cells confronted with antigen-free SLBs.

## Rechallenge protocol

Nalm6 cells were irradiated with 40 Gy and incubated for 24 h at 37 °C (day 0). Then, CAR T cells (5–7 d after lentiviral transduction) were added in a 1:1 ratio (day 1). After 48 h, new Nalm6 cells were irradiated with 40 Gy and incubated for 24 h at 37 °C (day 3). Then, the same CAR T cells were counted and adjusted in a 1:1 ratio with the newly irradiated Nalm6 cells (day 4). The same process was repeated until CAR T cells had three contacts with the Nalm6 cells, always having 3 d between each of them. Experiments were performed 24 h after the last contact.

## CyTOF

Cells pretreated as indicated were stimulated with PMA/ionomycin for 5 h in the presence of Brefeldin A and monensin (BD Biosciences). A β$_2$-microglobulin-based barcoding approach was used to minimize batch effects. Briefly, single-cell suspensions were pelleted, incubated with 20 μM lanthanum-139 (Trace Sciences)-loaded maleimido-monoamine-DOTA (MM-DOTA; Macrocyclics) in phosphate-buffered saline (PBS) for 5 min at 20 °C for live/dead discrimination. Cells were washed and each sample was then incubated with a distinct mix of β$_2$-microglobulin-based barcodes for 30 min at 4 °C and washed twice before pooling. Cells were resuspended in surface antibody cocktail, incubated for 30 min at 4 °C, washed twice in staining buffer, prefixed with paraformaldehyde (PFA) 1.6%, washed, and then fixed and permeabilized using FoxP3 staining buffer set (eBioscience) and stained intracellularly for 60 min at room temperature. Cells were further washed twice before fixation in 4% PFA (Electron Microscopy Sciences) solution containing 125 nM iridium overnight at 4 °C. After acquisition, all CyTOF files were normalized together using the bead-based Nolan Lab normalizer (available from https://github.com/nolanlab/bead-normalization/releases). Afterwards, clean-up of the FCS files was performed with FlowJo v.10 using Gaussian-derived parameters 'Residual', 'Center', 'Offset', 'Width', 'Event Length', MM-Dota and iridium to eliminate dead cells, debris and normalization beads before de-barcoding and further data analysis.

## Flow cytometry

For extracellular staining, cells were collected, washed once with FACS buffer (PBS and 2% FBS) and stained for 15 min at 4 °C in the dark. Intracellular staining was performed using eBioscience FOXP3/Transcription Kit (Invitrogen). Acquisition was performed in the Gallios (Beckman Coulter) or in the Attune NxT Acoustic Focusing Cytometer (Invitrogen). Analysis was done using the FlowJo Software v.10. For FACS, cells were resuspended in MACS buffer (PBS, 0.5% bovine serum albumin and 2 mM EDTA, pH 8) sorted in a MoFlo Astrios EQ (Beckman Coulter). Cell collection was done in RPMI medium + 20% FCS and cells were directly plated or frozen after the procedure. Gating strategies are found in Extended Data Fig. 8.

## Peptide pull-down and MS

Peptides were purchased from Eurogentec. Dephosphorylation of the peptides was done by incubation with calf intestine phosphatase (New England Biolabs) for 1 h at 37 °C, followed by inactivation with 4 mM Na$_3$VO$_4$ and 20 mM NaF. Cells, $20 \times 10^6$, were lysed in lysis buffer (0.35%

Brij96V, 20 mM Tris, pH 7.4, 137 mM NaCl, 10% glycerol, 2 mM EDTA, 200 mM phenylmethylsulfonyl fluoride (PMSF), 5 mM iodoacetamide, 4 mM $Na_3VO_4$ and 20 mM NAF) for 1 h on ice. Then 2 µg of biotinylated peptide was added to each postnuclear supernatant and incubated for 1 h at 4 °C. For bead control, no peptides were added. Subsequently, 10 µl of StrepT-actine-coupled Sepharose beads (IBA) was added and incubated for 1 h under agitation. Beads were washed 3× with lysis buffer and proteins analyzed by SDS–PAGE and immunoblotting. For MS analysis, $200 \times 10^6$ heavy and light isotope-labeled 2B4 T cells were used for the appropriately upscaled pull-down procedures as above. To compare samples by MS, heavy and light isotope-labeled pull-down samples were mixed in the third wash of the beads. Elution from the beads was done using 0.2 M glycine, pH 2.5 followed by a direct neutralization with 133 mM $(NH_4)_2CO_3$, pH 8.8.

### IP, SDS–PAGE and immunoblotting

Jurkat T cells, $30 \times 10^6$, or primary human T cell-expressing CARs, $10 \times 10^6$, were treated with 50 mM of the phosphatase inhibitor pervanadate (100 µl of 50 µM $Na_3VO_4$ and 0.5 µl of 10 M $H_2O_2$) at 37 °C for 5 min. Cells were lysed with 1 ml of lysis buffer containing 20 mM Tris-HCl, pH 8, 137 mM NaCl, 2 mM EDTA, 10% glycerol, protease inhibitor cocktail (Sigma-Aldrich), 1 mM PMSF, 5 mM iodoacetamide, 0.5 mM sodium orthovanadate, 1 mM NaF and 0.5% Brij96V for 30 min on ice, followed by a 15-min centrifugation to pellet the nuclei and insoluble materials. The supernatant was collected and 10 µl of protein G beads and 1 µg of anti-Strep-Tag II or 5 µg of anti-mouse $F(ab')_2$ antibody was added and incubated at 4 °C overnight. Subsequently, an incubation with 50 µg of purified SHP-1 was performed for 2 h at 4 °C. Beads were centrifuged at 4 °C, maximum speed, for 1 min. Beads were washed 3× with 50% EMBO lysis buffer, 10% Brij96V and 40% $H_2O$. Beads were boiled for 10 min at 95 °C with reducing sample buffer. Samples were separated using SDS–PAGE and a wet transfer to poly(vinylidene fluoride) membranes was performed. Membranes were developed for SHP-1 and CAR (anti-Strep-Tag II). Quantification was done using Image Lab Software from BioRad after chemiluminescence detection in an Image Quant LAS 4000 Mini (GE Healthcare).

### Chemicals, antibodies and peptides

For PBMC isolation, Pancoll (Pan Biotech) was used. For T cell activation, anti-human CD3 (UCHT-1, from J. Bluestone, University of California, San Francisco, USA) and anti-human CD28 (BioLegend, catalog no. CD28.2) were used. Primary human T cells were grown with human IL-2 (Perprotech). For transduction, proteamine sulfate salt from herring (Sigma-Aldrich, Life Sciences) was used. For IP the THE NWSHPQFEK tag antibody (Genscript, catalog no. 5A9F9) or the biotin-coupled anti-mouse IgG ($F(ab')_2$) biotin (Thermo Fisher Scientific, catalog no. AB_228311) and the anti-human SH-PTP1 (D11 or C19, Santa Cruz Biotechnology) were used. Precision count beads (BioLegend) were employed to count the cells in the rechallenge and CyTOF experiments. For the REP, anti-human CD3ε (OKT-3) was utilized.

The antibodies used for mass cytometry and flow cytometry are listed in the Supplementary Tables 1 and 2, respectively.

The following peptides were used in the present study:
CD3γ-Y-Y-biotin–AGQDGVRQSRASDKQTLLQNEQLYQPLKDREY-DQYSHLQGNQLRKK–COOH;
CD3γ-pYpY-biotin–AGQDGVRQSRASDKQTLLQNEQLY(PO3H2)QPLKDREYDQY(PO3H2)SHLQGNQLRKK–COOH;
CD3γ-pY-Y-biotin–AGQDGVRQSRASDKQTLLQNEQLY(PO3H2)QPLKDREYDQYSHLQGNQLRKK–COOH;
CD3γ-YpY-biotin–AGQDGVRQSRASDKQTLLQNEQLYQPLKDREYDQY(PO3H2)SHLQGNQLRKK–COOH;
CD3δ-Y-Y-biotin–GHETGRPSGAAEVQALLKNEQLYQPL-RDREDTQYSRLGGNWP;
RNKKS–COOH;

CD3δ-pYpY-biotin–GHETGRPSGAAEVQALLKNEQLY(PO3H2)QPLRDREDTQY(PO3H2)SRLGGNWPRNKKS–COOH;
CD3δ-pY-Y-biotin–GHETGRPSGAAEVQALLKNEQLY(PO3H2)QPLRDREDTQYSRLGGNWPRNKKS–COOH;
CD3δ-YpY-biotin–GHETGRPSGAAEVQALLKNEQLYQPLRDREDTQY(PO3H2)SRLGGNWPRNKKS–COOH.

### Data collection and statistical analysis

Data collection and analysis were not performed blind to the conditions of the experiments. Apart from in vivo experiments, no statistical methods were used to predetermine sample sizes, but our sample sizes are similar to those reported in previous publications or based on our own experience. Data were tested for normality applying the Shapiro–Wilk test. Statistical analysis was performed using GraphPad Prism (v.9). Applied analyses and statistical significances are indicated in the corresponding figures and figure legends. Differences with $P \le 0.05$ were considered statistically significant. NS, nonsignificant, $^*P \le 0.05$, $^{**}P \le 0.01$, $^{***}P \le 0.001$, $^{****}P \le 0.0001$. No datapoints were excluded from analysis.

### Reporting summary

Further information on research design is available in the Nature Portfolio Reporting Summary linked to this article.

### Data availability

Transcriptome data have been deposited in the Gene Expression Omnibus under accession no. GSE243226. Source data are provided with this paper. All other data that support the findings of the present study are present in the article or are available from the corresponding author upon request.

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

### Acknowledgements

We thank M. Duchniewicz for her contribution to the experiments in Fig. 8 and to S. L. von Löwesprung and K. Fehrenbach for technical assistance. We thank M. Kutuzov and O. Dushek (University of Oxford) for the synthesis of SHP-1, A. Rensing-Ehl (Universtiy of Freiburg) for the blocking antibodies, NanoString for trusting us to receive the CAR T cell NanoString grant in Freiburg in 2021

and the CIBSS Impulse Funds for the support to perform the CyTOF analyses (EXC 2189, to S.M. and B.B.). R.M.V.C. and T.P. were supported by the European Union's Horizon 2020 research and innovation program under the Marie Skłodowska-Curie grant agreement no. 721358 and R.M.H.V.C. by the FAZIT-Stiftung. A.B. was supported by the Excellence Initiative of the German Research Foundation (GSC-4, Spemann Graduate School). S.M. and W.S. were supported by the German Research Foundation (DFG) through BIOSS—EXC294 and CIBSS—EXC 2189. The DFG supports the following projects: SFB1479 (project no. 441891347: P15 to S.M., S1 to M.B. and P16 to B.B.), SFB1160 (project no. 256073931: B01 to S.M. and Z02 to M.B.), SFB1381 (A09 to W.S.), project 256073931 to B.B., MI 1942/4-1 (project no. 501418856 to S.M.) and MI 1942/5-1 (project no. 501436442 to S.M.). The German Federal Ministry of Education and Research supported the Medical Informatics Funding Scheme (MIRACUM, FKZ 01ZZ1801B to M.B., PM$^4$Onco FKZ 01ZZ2322A to M.B. and EkoEstMed FKZ 01ZZ2015 to G.A.). W.S., S.H. and M.G. were supported by the European Union 7th Framework project, SYBILLA (Systems Biology of T-cell activation) and M.G. by EU/EFPIA/OICR/McGill/KTH/Diamond Innovative Medicines Initiative 2 Joint Undertaking (EUbOPEN, grant no. 875510). We thank CIBSS Scientific Editing Service, C. Gross, for helpful comments on the manuscript.

## Author contributions

R.M.H.V.C. performed most experiments. S.M.B. and A.V.M. performed the revision experiments. R.M.H.V.C. and A.E.S. completed the mass cytometry (CyTOF) experiments and analysis. A.B. molecularly characterized BBε. T.P. performed the calcium experiments. R.M.V.C., S.M.B., K.R. and S.T. performed in vivo experiments. R.M.V.C. and F.B. performed the RNA experiments. S.M., B.S., M.B. and G.A. performed the respective analyses. S.H. and M.G. performed MS. B.B. advised on the CyTOF analysis. J.H. enabled the SLB experiments. S.L. facilitated the RNA experiments. S.M. and W.W.S. conceived the project and designed the study. S.M. and R.M.H.V.C. wrote the manuscript with the input of all the authors.

## Competing interests

S.M. and W.W.S. are patent holders on 'Lck-binding motif in CD3e', US patent application no. 20230070126. B.B. is a patent holder on 'Methods and compositions for treating diseases associated with exhausted T cells', US patent application no. 20210033595. The remaining authors declare no competing interests.

## Additional information

**Extended data** is available for this paper at https://doi.org/10.1038/s41590-023-01658-z.

**Correspondence and requests for materials** should be addressed to Susana Minguet.

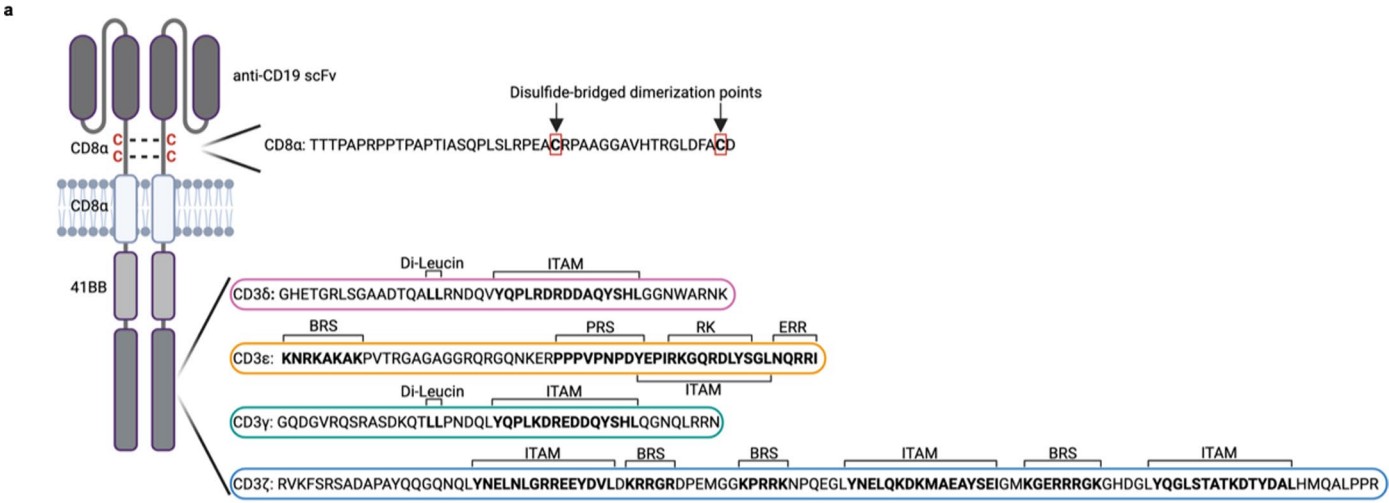

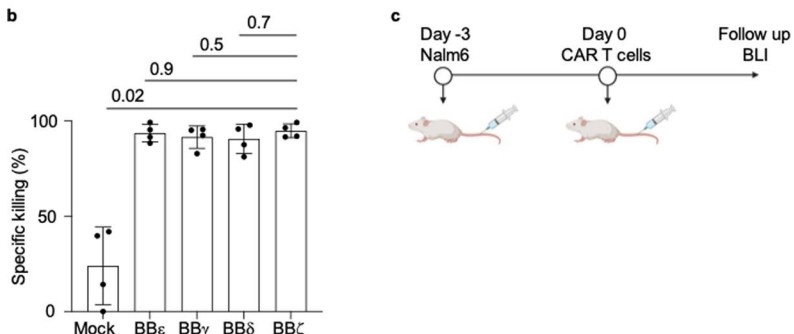

**Extended Data Fig. 1 | CARs containing the CD3δ/ε/γ ICD are functional.**
**a**, Scheme depicting the ICD of all CD3 chains used in this study and the localization of the cysteines driving CAR dimerization. **b**, Specific killing of CD19⁺ Nalm6 cells by primary human T cells lentivirally transduced with the indicated CAR after 24 h of co-incubation (1:1 ratio). Each dot represents an independent donor (*n* = 4). **c**, Schematic representation of the *in vivo* model. Data are represented as mean ± s.d. and analyzed by paired one-way ANOVA and Dunnett's multiple comparisons test. ITAM, Immunoreceptor Tyrosine-based Activation Motif; BRS, Basic Rich Stretch; PRS, Proline Rich Sequence; RK, Receptor Kinase; ERR, endoplasmic reticulum retention.

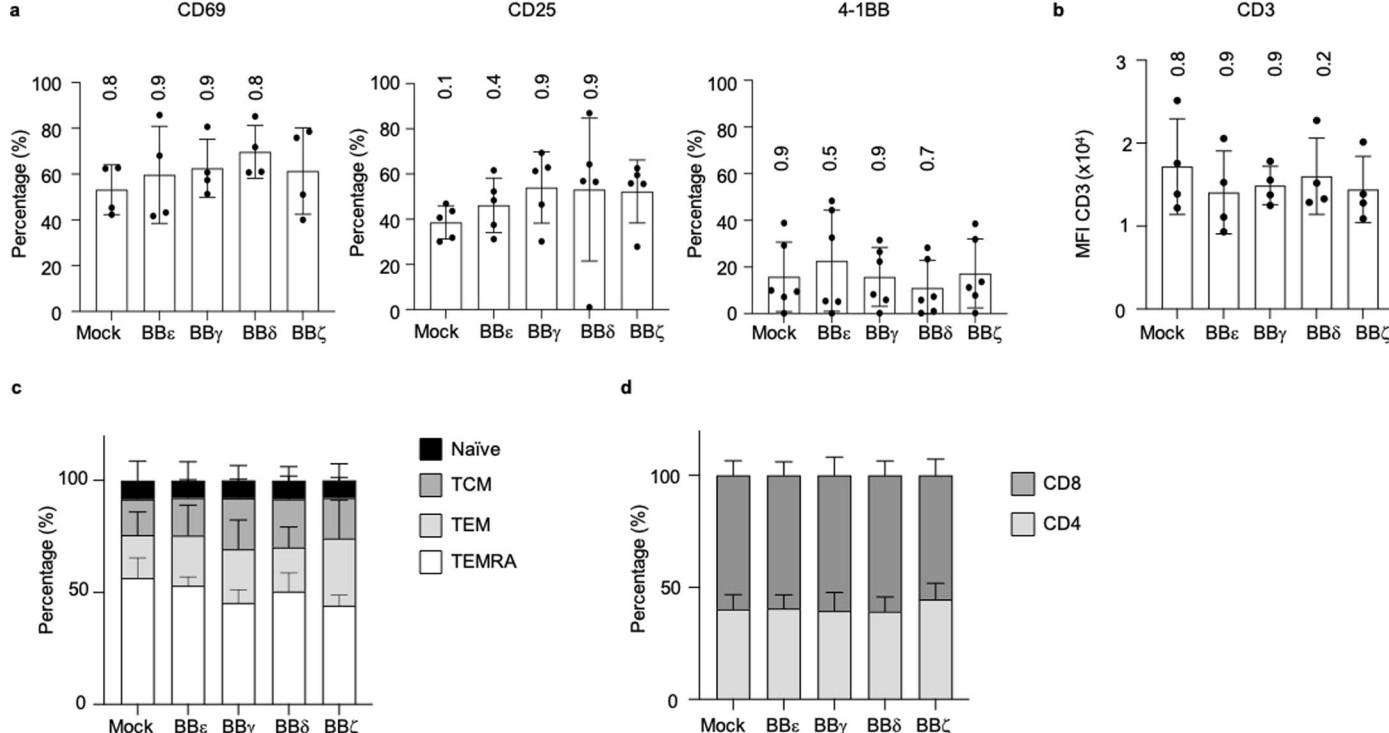

**Extended Data Fig. 2 | All CARs containing the TCR-CD3 ICD deliver tonic signals. a**, Activation markers (*n* = 4 for CD69, *n* = 5 for CD25, *n* = 6 for 4-1BB), **b**, endogenous TCR-CD3 expression (*n* = 4), **c**, differentiation (*n* = 4) and **d**, CD4/CD8 ratios (*n* = 3) of CAR T cells 6 days after transduction measured by flow cytometry. Each dot represents an independent donor. TCM, central memory; TEM, effector memory; and TEMRA, effector memory RA. Data are represented as mean ± s.d. One-way ANOVA (**a, b**) or Two-way ANOVA (**c, d**), followed by Dunnett's multiple comparisons test. All comparisons to BBζ were non-significant.

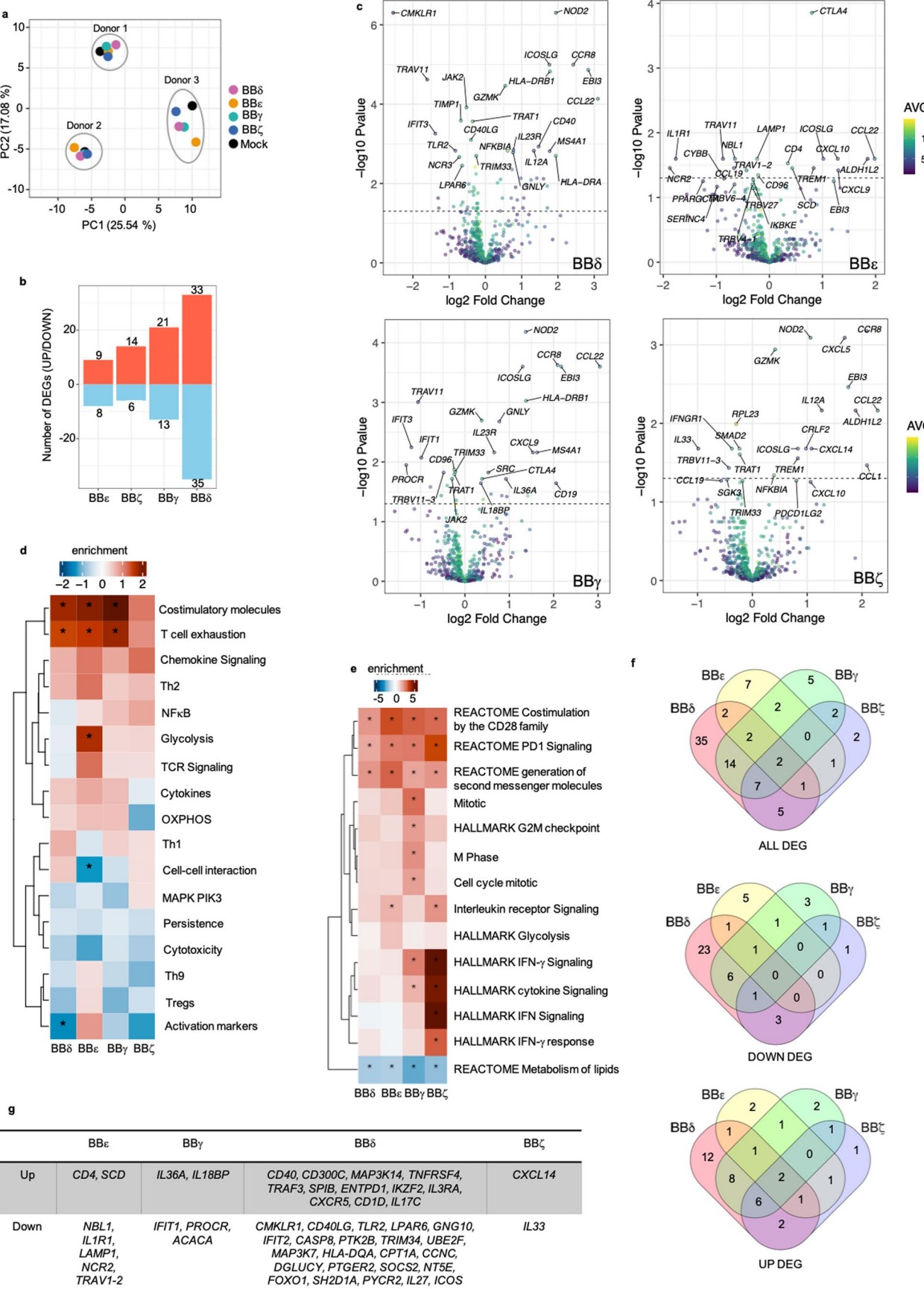

**Extended Data Fig. 3 | See next page for caption.**

**Extended Data Fig. 3 | Gene expression profile induced by CAR-derived tonic signals. a**, Principal component analysis (PCA), **b**, number of Differentially Expressed Genes (DEGs) significantly up- and down-regulated by each CAR versus Mock and **c**, volcano plots showing the differentially expressed genes between CAR and Mock cells from three different donors 6 days after CAR transduction. **d**, Heatmap of pathway analysis from each CAR construct compared to Mock. **e**, Heatmaps of the pathway analysis (Reactome, Consensus, Hallmark and GO) regulated by each single CAR construct compared to BB. **f**, Venn diagrams indicate the number of DEGs changed for each construct or shared among them. Every CAR is compared to Mock. **g**, Genes exclusively regulated by a single CAR construct. (**d, e**) Two-sample t-test, * indicates statistical significance, p < 0.05. Exact p values are provided in the source data file. *n* = 3 independent donors.

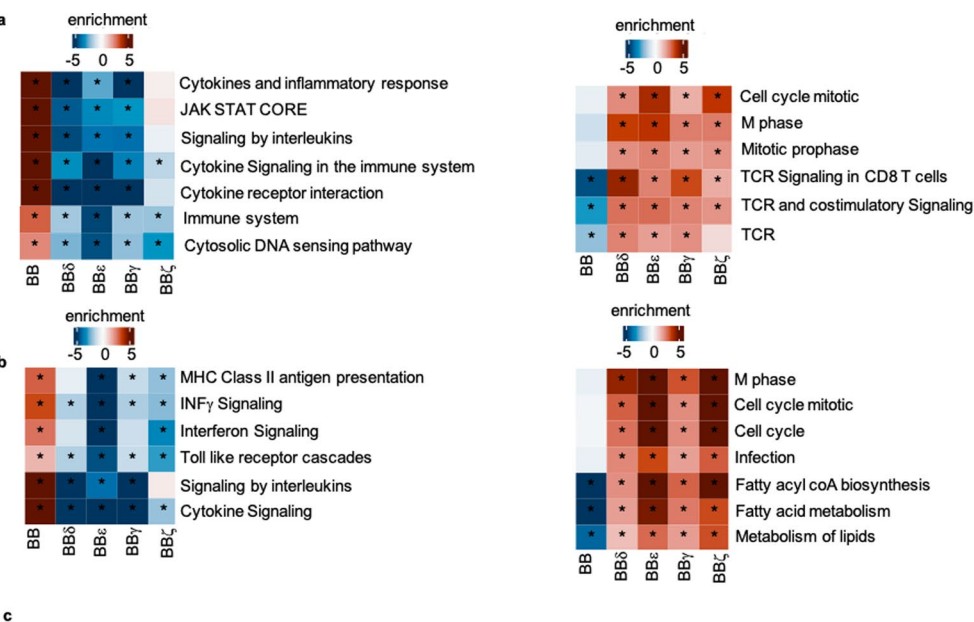

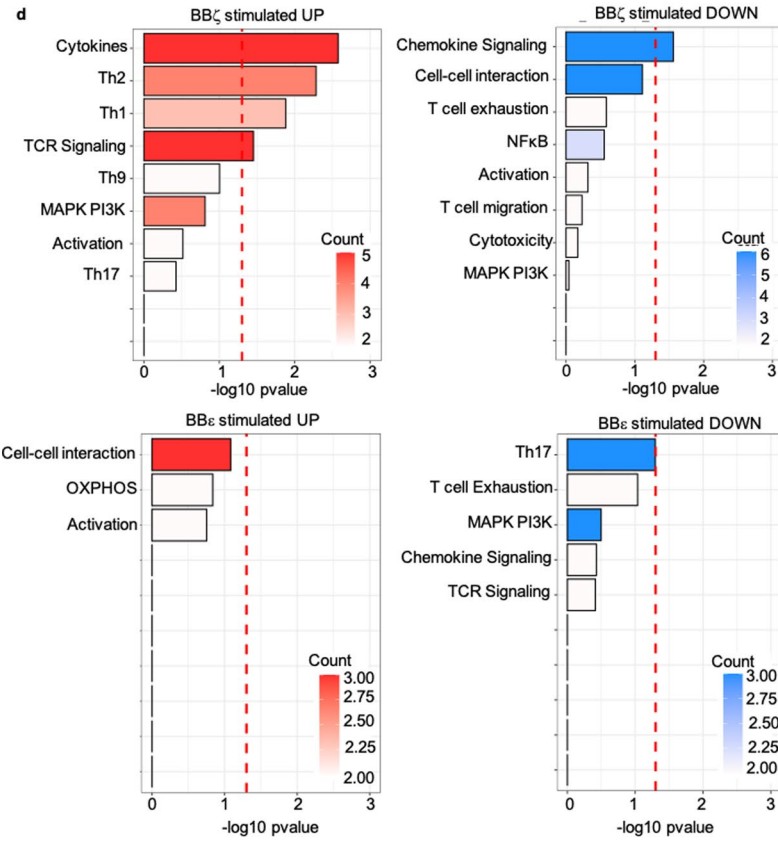

**Extended Data Fig. 4 | Gene expression profile induced by CARs upon antigen encounter. a, b,** Heatmaps of the pathway analysis, (**a**) Consensus and (**b**) Reactome. **c,** List of genes exclusively regulated by a single CAR construct compared to BB after 24 h stimulation with target cells. **d,** Pathway analysis (CAR T cells Nanostring) performed with the genes exclusively regulated by BBζ (up) or BBε (down) compared to BB upon 24 h stimulation with target cells. Two sample t-test, * indicates statistical significance, p < 0.05. Exact p values are provided in the source data file. n = 3 independent donors.

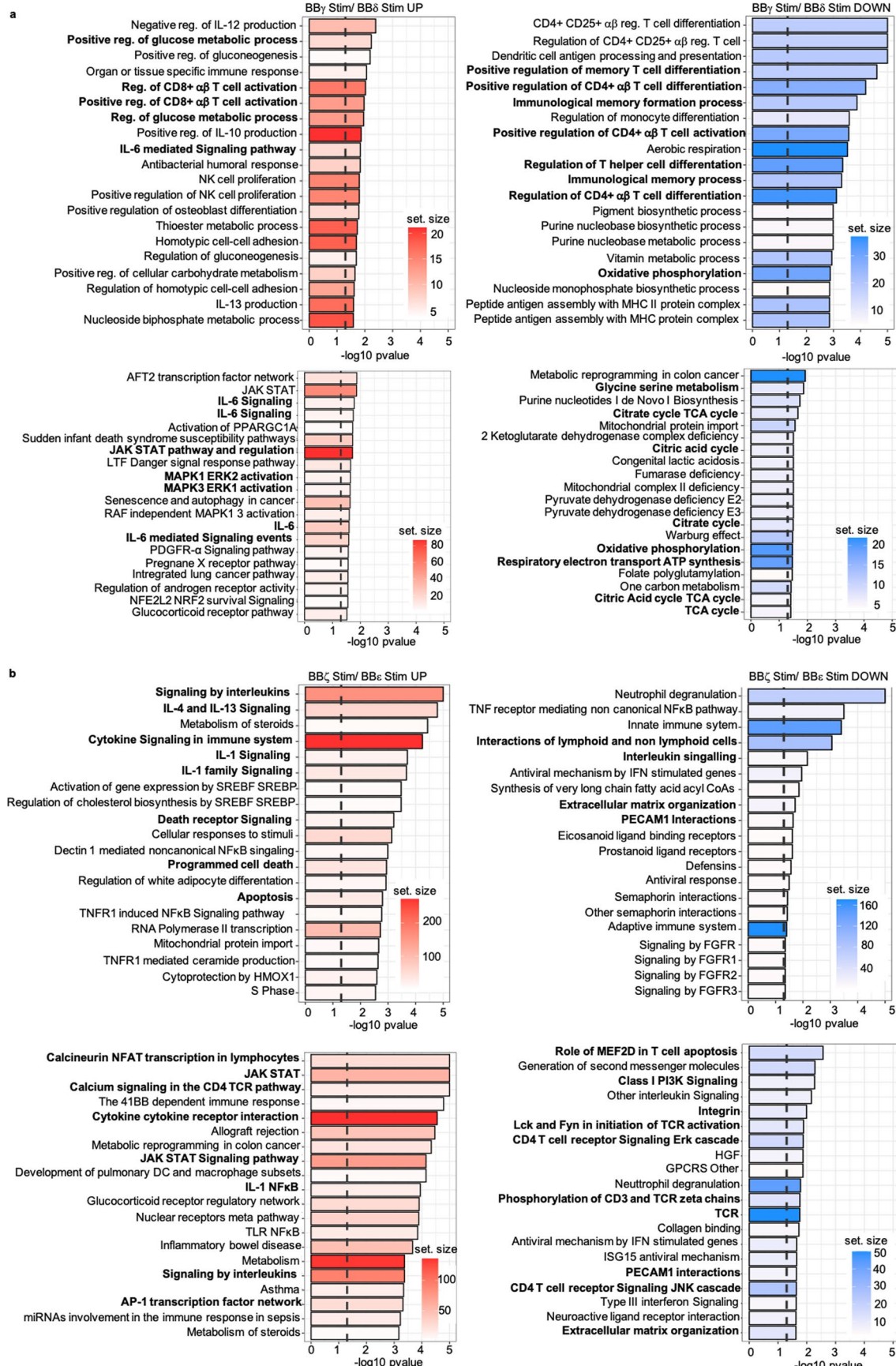

**Extended Data Fig. 5 | See next page for caption.**

**Extended Data Fig. 5 | Comparative gene expression profile induced by BBγ/BBδ and BBζ/BBε upon antigen stimulation. a**, Pathway analysis (GO and Consensus) directly comparing BBγ with BBδ (genes in the UP graph are significantly up-regulated in BBγ and genes in the DOWN graph are significantly up-regulated in BBδ). **b**, Pathway analysis (Reactome and Consensus) directly comparing BBζ with BBε (genes in the UP graph are significantly up-regulated in BBζ and genes in the DOWN graph are significantly up-regulated in BBε). Two-sample t-test, the dashed vertical line indicates statistical significance, p < 0.05. Relevant pathways are bolded. $n = 3$ independent donors.

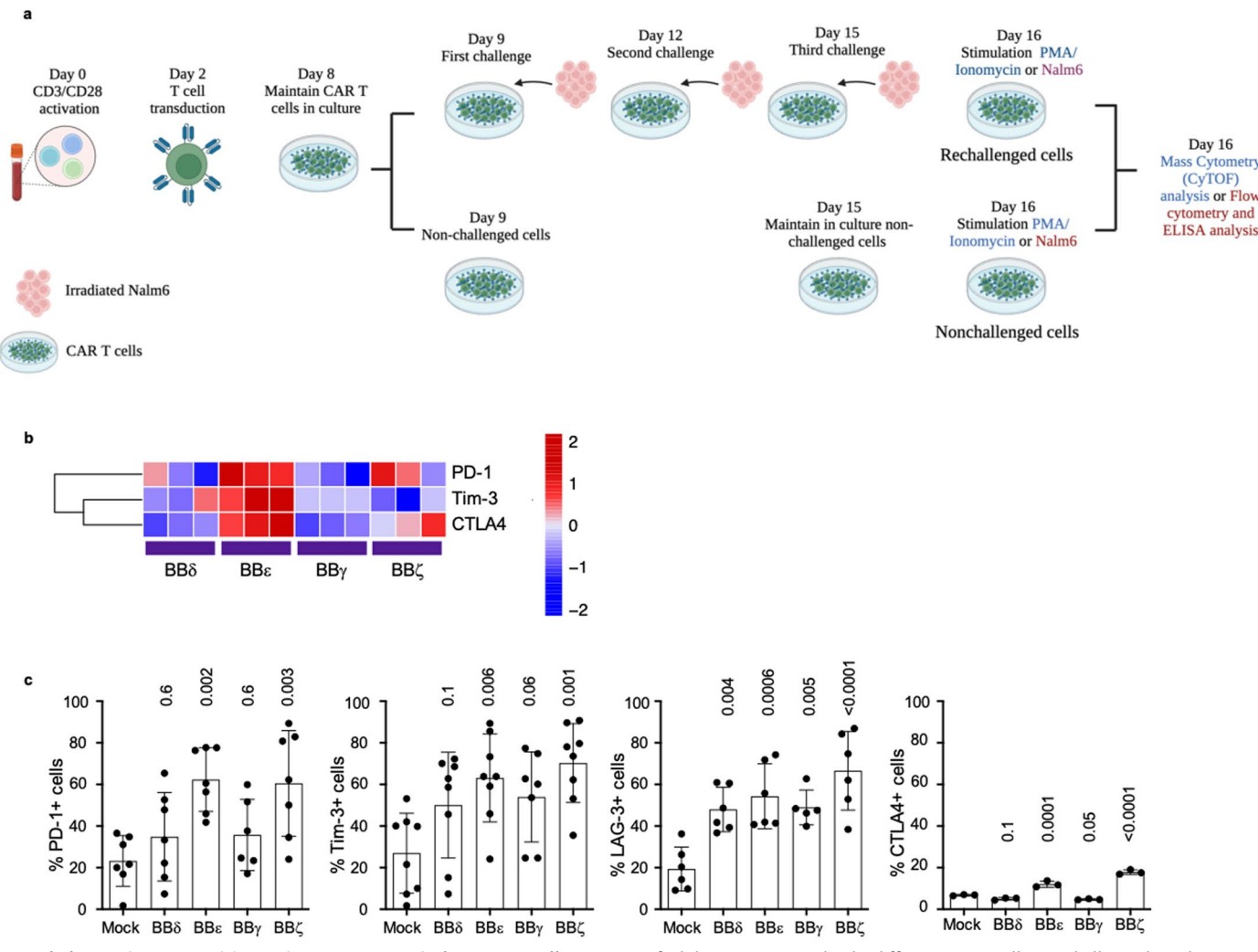

**Extended Data Fig. 6 | Repetitive antigen encounters induce CAR T cell dysregulation *in vitro*. a**, Schematic protocol to repetitively stimulate CAR T cells with irradiated Nalm6 (1:1 ratio) to generate nonchallenged and rechallenged CAR T cells. For CyTOF analysis, CAR T cells were sorted and stimulated with PMA/Ionomycin for 6 h to unravel the full activation potential. For flow cytometric analysis and ELISA, CAR T cells were incubated with Nalm6 for 24 h. **b**, Hierarchically clustered heatmap showing median marker expression of inhibitory receptors by the different CAR T cells in rechallenged conditions assayed by CyTOF analysis. Z-scores after column normalization are show $n = 3$. **c**, Up-regulation of inhibitory receptors by CAR T cells upon incubation with Nalm6 (1:1) for 24 h after the third challenge assayed by flow cytometry. Each dot represents an independent donor ($n = 7$ for PD-1, Tim-3 and LAG-3; $n = 3$ for CTLA4). Data are represented as mean ± s.d. One-way ANOVA followed by Dunnett's multiple comparisons test.

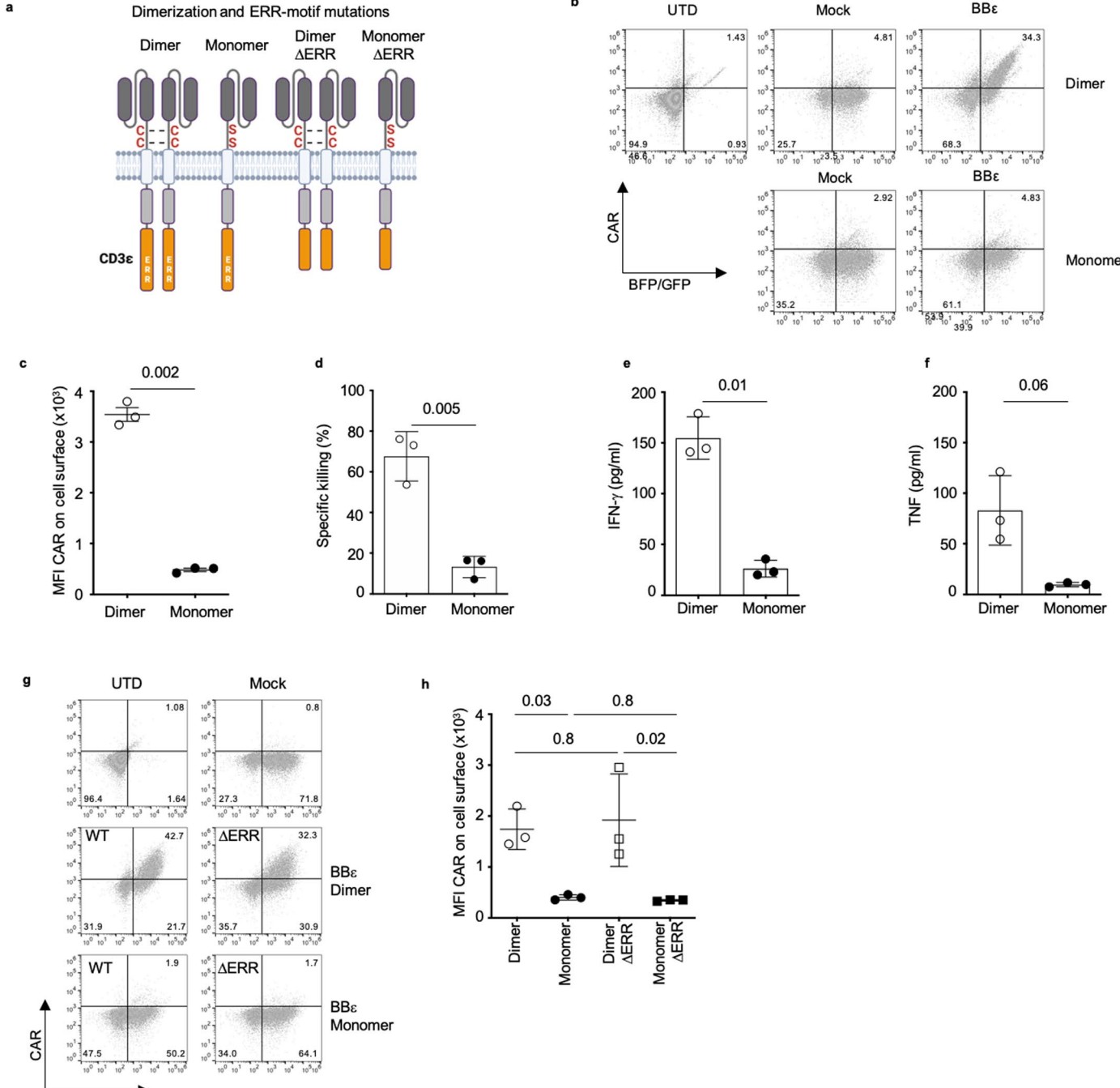

**Extended Data Fig. 7 | Dimerization is needed for BBε CAR surface expression. a**, Schematic representation of monomeric and dimeric BBε CARs and the mutations introduced. ERR, endoplasmic reticulum retention signal. **b,c**, Flow cytometric analysis (**b**) and quantification (**c**) of surface CAR expression for GFP$^+$ or BFP$^+$ cells ($n$ = 3). **d**, Specific killing of CD19$^+$ Nalm6 cells by CAR T cells (1:1 ratio) for 12 h. ($n$ = 3). **e**, **f**, Cytokine secretion, IFN-γ (**e**) and TNF (**f**), assayed by ELISA of CAR T cells upon incubation with Nalm6 (1:1 ratio) for 24 h ($n$ = 3). **g**, Flow cytometric analysis and **h**, quantification of surface CAR expression for BFP$^+$ or GFP$^+$ cells ($n$ = 3). Each dot represents an independent donor. Data are represented as mean ± s.d. Two-tailed paired t-test (**c**–**f**). One-way ANOVA followed by Holm-Sidak's multiple comparisons test (**h**).

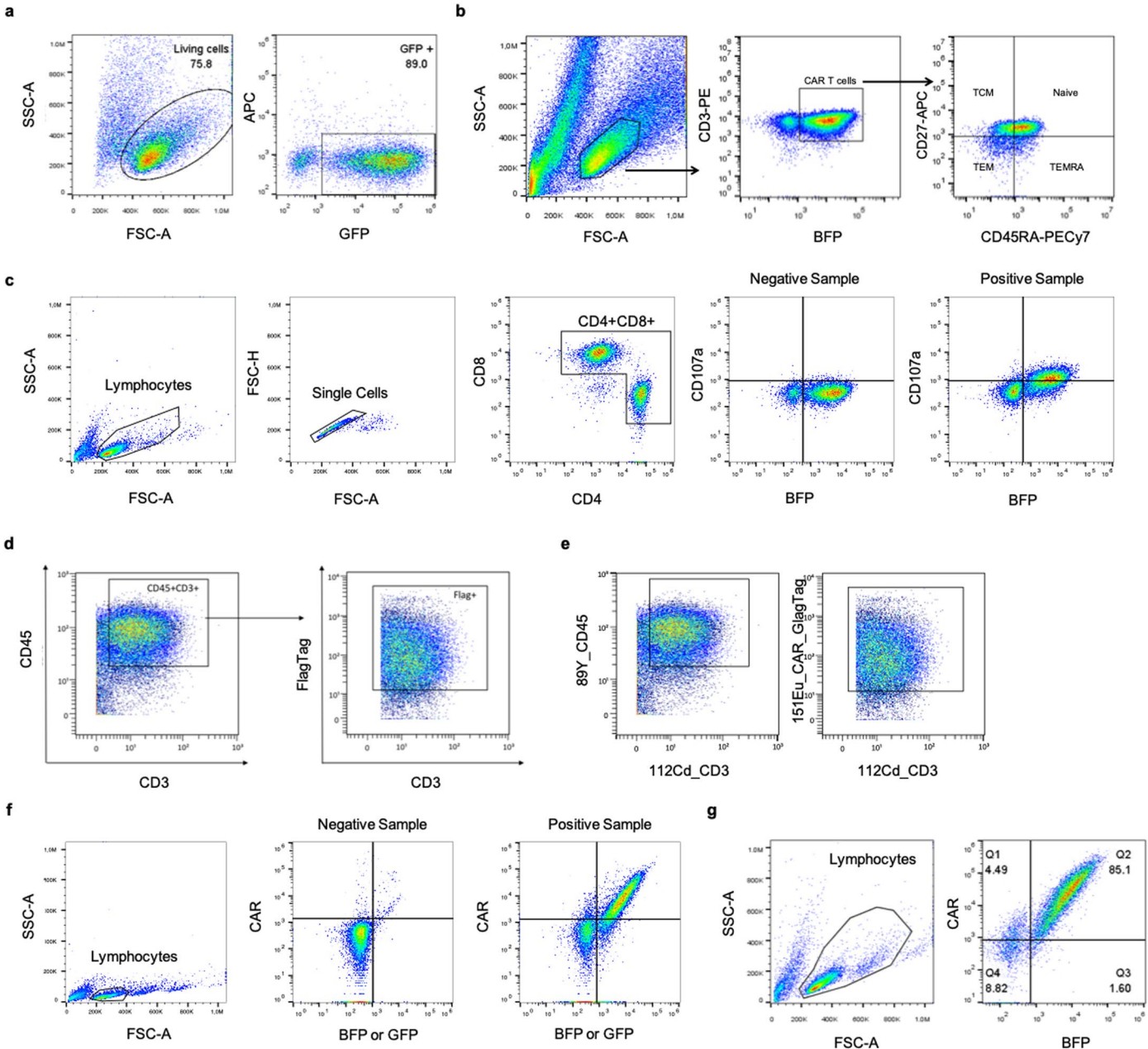

**Extended Data Fig. 8 | Gating strategy for flow cytometric analysis. a**, Gating strategy to sort for GFP⁺ cells as presented for Fig. 1b. **b** Gating strategy to gate on phenotypic T cell subsets as presented in Fig. 2b. **c** Gating Strategy to determine CD107a upregulation on CAR T cells upon Nalm6 co-culture as presented in Fig. 2g. **d,** Gating Strategy for Fig. 4a. **e**, Gating Strategy for Fig. 5a. **f**, Gating strategy to determine GFP⁺CAR⁺ or BFP⁺CAR⁺ T cell populations as presented for Fig. 6b, Extended Data Figs. 7 and g. **g**, Gating strategy to determine BFP⁺CAR⁺ T cell populations as presented for Fig. 7b.

# Reporting Summary

## Statistics

For all statistical analyses, confirm that the following items are present in the figure legend, table legend, main text, or Methods section.

| n/a | Confirmed | |
|---|---|---|
| ☐ | ☒ | The exact sample size (*n*) for each experimental group/condition, given as a discrete number and unit of measurement |
| ☐ | ☒ | A statement on whether measurements were taken from distinct samples or whether the same sample was measured repeatedly |
| ☐ | ☒ | The statistical test(s) used AND whether they are one- or two-sided<br>*Only common tests should be described solely by name; describe more complex techniques in the Methods section.* |
| ☒ | ☐ | A description of all covariates tested |
| ☒ | ☐ | A description of any assumptions or corrections, such as tests of normality and adjustment for multiple comparisons |
| ☐ | ☒ | A full description of the statistical parameters including central tendency (e.g. means) or other basic estimates (e.g. regression coefficient) AND variation (e.g. standard deviation) or associated estimates of uncertainty (e.g. confidence intervals) |
| ☐ | ☒ | For null hypothesis testing, the test statistic (e.g. *F*, *t*, *r*) with confidence intervals, effect sizes, degrees of freedom and *P* value noted<br>*Give P values as exact values whenever suitable.* |
| ☒ | ☐ | For Bayesian analysis, information on the choice of priors and Markov chain Monte Carlo settings |
| ☒ | ☐ | For hierarchical and complex designs, identification of the appropriate level for tests and full reporting of outcomes |
| ☐ | ☒ | Estimates of effect sizes (e.g. Cohen's *d*, Pearson's *r*), indicating how they were calculated |

*Our web collection on statistics for biologists contains articles on many of the points above.*

## Software and code

Policy information about availability of computer code

| Data collection | Cytokines were measured by ELISA with Thermo Scientific MultiScan.<br>For SLB experiments an iXon Ultra 897 EMCCD camera (Oxford Instruments) was used for data recording.<br>Bioluminescence-based cytotoxicity assay: Luminometer (Tecan infinity M200 Pro).<br>Flow cytometry: Gallios FACS, Beckman Coulter and Attune NxT FACS, Thermo Fisher Scientific.<br>Bioluminescence imaging: IVIS SpectrumCT System.<br>CyTOF (Helios) |
|---|---|

| Data analysis | Immunobotting: Quantification of the band intensities was performed with the Image Lab Software from BioRad after chemiluminescence detection in an Image Quant LAS 4000 Mini (GE Healthcare).<br>Flow cytometry analysis: Flowjo Software V10<br>Statistics: GraphPad Prism 9<br>SLB experiments were analysed with an inhouse custom-built Matlab software was used to track cells in each frame using a particle tracking algorithm.<br>Gene expression data was analysed nSolver 4.0 software. Downstream bioinformatic analysis was performed with R (4.2.1). Briefly, the differentially regulated genes were identified using the limma R package, with a paired design (donor-based). Adjusted p value (Benjamini Hochberg) below 0.05 was considered as significant. The Generally Applicable Gene-set Enrichment (GAGE) R package was used to identified regulated gene-sets among the whole MSigDB repository and the NanoString panel gene-sets.<br>Cytof: Normalizer (https://www.github.com/nolanlab/bead- normalization/releases), FlowJo (v10), Omiq (https://www.r-project.org) and R (https://www.r-project.org/). |
|---|---|

For manuscripts utilizing custom algorithms or software that are central to the research but not yet described in published literature, software must be made available to editors and reviewers. We strongly encourage code deposition in a community repository (e.g. GitHub). See the Nature Portfolio guidelines for submitting code & software for further information.

# Data

Policy information about availability of data

All manuscripts must include a data availability statement. This statement should provide the following information, where applicable:
- Accession codes, unique identifiers, or web links for publicly available datasets
- A description of any restrictions on data availability
- For clinical datasets or third party data, please ensure that the statement adheres to our policy

Transcriptome data have been deposited in the Gene Expression Omnibus under the accession number GSE243226. Source Data are provided with this paper. All other data that support the findings of this study are present in the article or are available from the corresponding author upon request.

# Human research participants

Policy information about studies involving human research participants and Sex and Gender in Research.

| Reporting on sex and gender | Use the terms sex (biological attribute) and gender (shaped by social and cultural circumstances) carefully in order to avoid confusing both terms. Indicate if findings apply to only one sex or gender; describe whether sex and gender were considered in study design whether sex and/or gender was determined based on self-reporting or assigned and methods used. Provide in the source data disaggregated sex and gender data where this information has been collected, and consent has been obtained for sharing of individual-level data; provide overall numbers in this Reporting Summary. Please state if this information has not been collected. Report sex- and gender-based analyses where performed, justify reasons for lack of sex- and gender-based analysis. |
|---|---|
| Population characteristics | Describe the covariate-relevant population characteristics of the human research participants (e.g. age, genotypic information, past and current diagnosis and treatment categories). If you filled out the behavioural & social sciences study design questions and have nothing to add here, write "See above." |
| Recruitment | Describe how participants were recruited. Outline any potential self-selection bias or other biases that may be present and how these are likely to impact results. |
| Ethics oversight | Identify the organization(s) that approved the study protocol. |

Note that full information on the approval of the study protocol must also be provided in the manuscript.

# Field-specific reporting

Please select the one below that is the best fit for your research. If you are not sure, read the appropriate sections before making your selection.

☒ Life sciences  ☐ Behavioural & social sciences  ☐ Ecological, evolutionary & environmental sciences

For a reference copy of the document with all sections, see nature.com/documents/nr-reporting-summary-flat.pdf

# Life sciences study design

All studies must disclose on these points even when the disclosure is negative.

| Sample size | For animal studies, the recommendations from Prof. Dr. Hauschke (Institute for Med. Biometry and Med. Informatics, University of Freiburg) were followed. These recommendations were enclosed in the animal protocol following the German animal protection law and with permission from the responsible local authorities. Briefly, The sample size was calculated using the following parameters: 1.06 effect size, 5% significance level, 80% power, and 1.06 standard deviation. |
|---|---|

| | For ex-vivo experiments, at least two experiments were performed to estimate the standard deviation and the effect size. These estimations were applied to the Lehr's formula to estimate the sample size needed. |
|---|---|
| Data exclusions | Animals which do not fill up the health-criteria described in the animal protocol that at the moment of the analysis were excluded. The criteria were therefore pre-established at the animal protocol. Failed Experiments due to technical issues were excluded. |
| Replication | Experiments were replicate at least twice and statistics were done to verify reproducibility. All attends of replications were sucessful. |
| Randomization | Allocation was ramdom |
| Blinding | Blinding was not done (impractical due to the difficulties to control all variables without implying mistakes) |

# Reporting for specific materials, systems and methods

We require information from authors about some types of materials, experimental systems and methods used in many studies. Here, indicate whether each material, system or method listed is relevant to your study. If you are not sure if a list item applies to your research, read the appropriate section before selecting a response.

## Materials & experimental systems

| n/a | Involved in the study |
|---|---|
| ☐ | ☒ Antibodies |
| ☐ | ☒ Eukaryotic cell lines |
| ☒ | ☐ Palaeontology and archaeology |
| ☐ | ☒ Animals and other organisms |
| ☒ | ☐ Clinical data |
| ☒ | ☐ Dual use research of concern |

## Methods

| n/a | Involved in the study |
|---|---|
| ☒ | ☐ ChIP-seq |
| ☐ | ☒ Flow cytometry |
| ☒ | ☐ MRI-based neuroimaging |

## Antibodies

| Antibodies used | Antibodies used for mass cytometry (CyTOF):<br>Channel Antigen Supplier Clone Cat#<br>89Y  CD45 fluidigm HI30 3089003B<br>104Pd β2m-Barcode Biolegend 2M2 316302<br>105Pd β2m-Barcode Biolegend 2M2 316302<br>106Cd β2m-Barcode Biolegend 2M2 316302<br>108Pd β2m-Barcode Biolegend 2M2 316302<br>110Pd β2m-Barcode Biolegend 2M2 316302<br>111 Cd CD4 Biolegend RPA-T4 300502<br>112 Cd CD3 Biolegend UCHT1 300402<br>113 Cd CD39 Biolegend A1 328202<br>114 Cd IRF4 Invitrogen 3E4 14-9858-82<br>115 In CD57 Invitrogen TB01 16-0577-85<br>116 Cd CD8 Biolegend RPA-T8 301002<br>139 La  MM-DOTA<br>140 etc Beads<br>141 Pr CD19 Biolegend HIB19 302202<br>142 Nd IFN-  Biolegend B27 506502<br>143 Nd HIF1  Abcam EP1215Y ab210073<br>144 Nd CTLA-4 BD BNI3 555850<br>145 Nd TNF-  Invitrogen MAb11 14-7349-85<br>146 Nd Ki-67 BD B56 556003<br>147 Sm CD45RA Biolegend HI100 304143<br>148 Nd ACC Cell Signaling Technology C83B10 3676BF<br>149 Sm GLS1 Cell Signaling Technology E4T9Q 49363BF<br>150 Nd CD127 Biolegend A019D5 351302<br>151 Eu Flag-Tag (anti-DYKDDDDK) BioLegend L5 637301<br>152 Sm IL-2 Invitrogen MQ1-17H12 14-7029-85<br>153 Eu CPT1  Cell Signaling Technology D3B3 12252BF<br>154 Sm XCL-1 R&D 109001 MAB6951<br>155 Gd CD27 Biolegend O323 302802<br>156 Gd TOMM20 Abcam ERP15581-54 ab232589<br>157 Gd FOXO1 Biolegend 2F8B08 658102<br>158 Gd PD-1 Biolegend EH12.2H7 329902<br>159 Tb CCR7 Biolegend G043H7 353255<br>160 Gd Tbet Biolegend 4B10 644802<br>161 Dy CD28 Biolegend CD28.2 302902<br>162 Dy Tim-3 BioLegend F38-2E2 345002<br>163 Dy TCF-1 Biolegend 7F11A10 615702 |
|---|---|

164 Dy CytC Biolegend 6H2.B4 612302
165 Ho Eomes Invitrogen WD1928 14-4877-82
166 Er Bodipy Invitrogen poly A5770
167 Er CD38 Biolegend HIT2 303502
168 Er TOX Milteny REA473 130-095-212
169 Tm TIGIT Invitrogen MBSA43 16-9500-82
170 Er LAG-3 Biolegend  11C3C65 369302
171 YB 2B4 Biolegend C1.7 329502
172 Yb CD71 Invitrogen OKT9 14-0719-82
173 Yb CCL3/4 R&D 93342 MAB2701
174 Yb IL-10 Biolegend JES3-9D7 501402
175 Lu GLUT1 Abcam EPR3915 ab252403
176 Yb CD36 Biolegend 5-271 336202
191/193 Iridium
194Pt Perforin abcam B-D48 ab47225
198 Pt β2m-Barcode
209 Bi CD16 fluidigm 3G8 3165001C
Antibodies used for Flow Cytometry:
Conjugate Antigen Supplier Clone Cat#
 Biotin Strep-Tag (NWSHPQFEK) Genscript 5A9F9 A01736
APC Flag-Tag (DYKDDDDK) BioLegend L5 637308
Biotin  mouse IgG  Invitrogen  31803
Biotin human PD-1 BioLegend EH12.2H7 329934
PECy7 human Tim-3 BioLegend F382E2 345014
Alexa 647 human LAG-3 BioLegend 11C3C65 369304
PE human CD25 BioLegend BC96 302606
PECy7 human CD137 Invitrogen 4B4-1 25-1379-42
APC human CD69 Invitrogen CH/4 MHCD6905
PE human CD8 Beckman Coulter B9.11 IM0452U
APC human CD8 Beckman Coulter B9.11 IM2469
PECy7 human CD4 eBioscience RPA-T4 25-0049-42
PE human CD107a Biolegend H4A3 328608
PECy7 human CD4 eBioscience 14D3 25-1529-42
PECy7 human TCF1/TCF7 Cell Signaling Technology C63D9 90511
PECy7 human CD45RA Biolegend HI100 304126
APC human CD27 BD Pharmingen MT271 561400
BV421 human CD3 BioLegend UCHT1 300434
Alexa 488 human CD3 BioLegend UCHT1 300415
APC Streptavidin BioLegend  405207
PECy7 Streptavidin Invitrogen  SA1012
eFluor 450  Streptavidin Invitrogen  48-4317-82

anti-human CD3 (UCHT-1, from J. Bluestone, UC San Francisco, USA) and anti-human CD28 (CD28.2, BioLegend) were used. Primary human T cells were grown with human IL-2 (Perprotech). For transduction, Proteamine Sulfate salt from herring (Sigma, Life Sciences) was used. For immunoprecipitation the THE™ NWSHPQFEK Tag Antibody (5A9F9, Genscript) or the biotin-coupled anti-mouse IgG (Fab´)2 biotin (AB_228311, Thermo Fisher) and the anti-human SH-PTP1 (D11 or C19, Santa Cruz Biotechnology) were used. Precision Count Beads (BioLegend) were employed to count the cells in the rechallenge and CyTOF experiments. For the rapid Expansion Protocol, anti-human CD3   (OKT-3) was utilized.

| Validation | All antibodies were purchased from manufactures that indicate validation in the data sheet. Antibodies for Western blot were validated using cells lacking the protein of interest and by protein size upon SDS-PAGE. Antibodies used in flow and mass cytometry were further validate using cell mixtures containing cells expressing or lacking the protein recognized by the antibody. |
|---|---|

# Eukaryotic cell lines

Policy information about cell lines and Sex and Gender in Research

| Cell line source(s) | HEK293T (CRL-1573) and Jurkat (TIB-152) were obtained from ATCC.  2B4 cells were obtained from Balbino Alarcon (CBM-SO, Spain). Nalm6 cells were obtained from TCR2 therapeutics. |
|---|---|
| Authentication | Cells were authenticated by expression of  surface markers by Flow cytometry. |
| Mycoplasma contamination | All cell lines were tested negative for Mycoplasm contamination. |
| Commonly misidentified lines (See ICLAC register) | No cell line listed by ICLAC was used. |

# Animals and other research organisms

Policy information about studies involving animals; ARRIVE guidelines recommended for reporting animal research, and Sex and Gender in Research

| Laboratory animals | Mice were used in this study. Rag2-γ- (Rag2tm1.1Flv Il2rgtm1.1Flv) were purchased from Jackson Laboratory, bred and kept at the |
|---|---|

| Laboratory animals | Center for Experimental Models and Transgenic Service, Freiburg, under specific pathogen-free conditions. Both adult (older than 8 weeks) females and males were used. |
| Wild animals | The study did not involved wild animals |
| Reporting on sex | The sex was not considered for the study design |
| Field-collected samples | The study did not involved field-collected samples |
| Ethics oversight | All animal protocols (G18/03) were performed according to the German animal protection law with permission from the Veterinär und Lebensmittelüberwachungsbehörde Freiburg. |

Note that full information on the approval of the study protocol must also be provided in the manuscript.

# Flow Cytometry

## Plots

Confirm that:

☒ The axis labels state the marker and fluorochrome used (e.g. CD4-FITC).

☒ The axis scales are clearly visible. Include numbers along axes only for bottom left plot of group (a 'group' is an analysis of identical markers).

☒ All plots are contour plots with outliers or pseudocolor plots.

☒ A numerical value for number of cells or percentage (with statistics) is provided.

## Methodology

| Sample preparation | For extracellular staining, cells were collected, washed once with FACS Buffer (PBS + 2% FBS), and stained for 15 minutes at 4°C in the darkness. Then, cells were washed twice with FACS buffer and acquisition was performed.  Intracellular staining has been performed using eBioscience FOXP3/Transcription Factor Staining Buffer Set (Invitrogen). Briefly, cells were washed with FACS buffer and subsequently resuspended in 1xFixation/Permeabilization solution and incubated for 30 minutes at 4°C. After fixation and permeabilization, cells were washed twice with 1xPermeabilization buffer and intracellular staining of TCF1 has been performed at RT for 30 minutes. After two washing steps with 1xPermeabilization buffer cells have been resuspended in FACS buffer. |
| Instrument | Gallios FACS, Beckman Coulter or in Attune® NxT Acoustic Focusing Cytometer, Invitrogen. |
| Software | FlowJo Software |
| Cell population abundance | Upon sorting, purifty <95% was obteined, |
| Gating strategy | Gating strategy is now depicted in Extended Data Fig. 8 |

☒ Tick this box to confirm that a figure exemplifying the gating strategy is provided in the Supplementary Information.

