## [Peer Review File · Nature Immunology]

Peer Review Information

Journal: Nature Immunology

Manuscript Title: Harnessing CD3 diversity to optimize CAR T cells

Corresponding author name(s): Professor Susana Minguet

Reviewer Comments & Decisions:

Decision Letter, initial version:
--

6th Jan 2023

Dear Susana,

Thank you for providing a point-by-point response to the referees' comments on your manuscript entitled, "Harnessing CD3 diversity to optimise CAR T cells". As noted previously, while they find your work of considerable potential interest, they have raised quite substantial concerns that must be addressed. In light of these comments, we cannot accept the current manuscript for publication, but would be very interested in considering a revised version that addresses these serious concerns.

We invite you to submit a substantially revised manuscript, however please bear in mind that we will be reluctant to approach the referees again in the absence of major revisions.

Specifically, the revision should include new experiments to address:

- (1) increase the donor number (referee #1 point 3) to address significance differences for the various CD3 BB constructs for naive T cell proportions
- (2) confirm CD3d-YpY/SHP-1 interactions identified by mass spectrometry - we noticed that the BBd-SHP1 immunoprecipitations performed in Fig. 7f,g were done using transfected Jurkat T cells; it would be preferable to use primary donor T cells.
- (3) Analyze SHP-1 binding to chimeric CD3d mutants that cannot dimerize (mutation of the extracellular CD8a dimerization motif, as well as the LL motif mentioned by referee #2)
- (4) perform degranulation assays to test whether the killing mediated by the CD3-based CARs correspond to CD107a up-regulation
- (5) perform validation qPCR assays to validate top hits identified by the CyTOF analysis

Please include the additional textual clarifications as indicated in your response letter, especially the distinctions of the current study studying the individual CD3 chains independently of the presence of

other CD3 chains.

Also, your response to referee #2 point 6 still does not provide an explanation (or speculate why) for the observed CD3d/SHP1 interaction only with the monophosphorylated form of CD3d ICD - please address in discussion.

Additionally, the Methods should be clarified to indicate the source of T cells used in the mass spectrometry experiments. Also, should include a Supplementary Figure depicting the ICD for all CD3 isoforms examined in this study.

When you revise your manuscript, please take into account all reviewer and editor comments, please highlight all changes in the manuscript text file in Microsoft Word format.

* If you have not done so already please begin to revise your manuscript so that it conforms to our Article format instructions at <http://www.nature.com/ni/authors/index.html>. Refer also to any guidelines provided in this letter.

The Reporting Summary can be found here:

When submitting the revised version of your manuscript, please pay close attention to our [href="https://www.nature.com/nature-portfolio/editorial-policies/image-integrity">Digital Image Integrity Guidelines. and to the following points below:](https://www.nature.com/nature-portfolio/editorial-policies/image-integrity)

-- that unprocessed scans are clearly labelled and match the gels and western blots presented in figures.

-- that control panels for gels and western blots are appropriately described as loading on sample processing controls

-- all images in the paper are checked for duplication of panels and for splicing of gel lanes.

[REDACTED]

If you wish to submit a suitably revised manuscript we would hope to receive it within 6 months. If you cannot send it within this time, please let us know. We will be happy to consider your revision so long as nothing similar has been accepted for publication at Nature Immunology or published elsewhere.

Nature Immunology is committed to improving transparency in authorship. As part of our efforts in this direction, we are now requesting that all authors identified as 'corresponding author' on published papers create and link their Open Researcher and Contributor Identifier (ORCID) with their account on the Manuscript Tracking System (MTS), prior to acceptance. ORCID helps the scientific community achieve unambiguous attribution of all scholarly contributions. You can create and link your ORCID from the home page of the MTS by clicking on 'Modify my Springer Nature account'. For more information please visit www.springernature.com/orcid.

Thank you for the opportunity to review your work.

Sincerely,

Laurie A. Dempsey, Ph.D.
Senior Editor
Nature Immunology
l.dempsey@us.nature.com
ORCID: 0000-0002-3304-796X

Referee expertise:

Referee #1: Tumor immunology

Referee #2: TCR signaling

Reviewers' Comments:

Reviewer #1:

Remarks to the Author:

This research article by Velasco Cardenas et al. investigates how individual CD3 chains of the TCR-CD3 complex can improve/affect CAR T cell performance. The authors generated and analyzed variants of the FDA-approved CAR Kymriah (4-1BB/ ζ), in which the CD3 ζ intercellular domain was systematically

exchanged for each of the CD3 δ / ϵ / γ intercellular domains (ICDs).

As an initial result, they showed that CD3 δ / ϵ / γ ICDs can generate functional CAR T cells by using a luciferase-based killing assay. All the generated CD3 ICDs had anti-tumor activity but CD3 δ had the best anti-tumor function. Next, they identified the role of CD3 ICDs in T cell activation outcome for the CAR T cells studied. BB ζ and BB ϵ CARs exhibited a higher signaling capacity whereas BB ζ and BB ϵ showed least response in terms of cytokine production and Ca²⁺ signaling. Then, the authors characterized the transcriptional programming of each modified CAR T cells upon antigen stimulation and data showed different transcription signatures for each CAR T cell type with BB δ ICD indicating memory T cell differentiation preference. A new repetitive-challenge protocol was formulated and UMAP analysis was performed to identify up/down-regulated genes, which showed BB δ cells maintain the best ability to kill tumor cells followed by BB γ cells. They then generated mutated versions of BB ϵ cells and studied their killing abilities. BB ϵ Δ PRS didn't result in reduction of killing ability suggesting that the PRS motif plays little role in T cell cytotoxicity. Finally, by SILAC labeling and mass spectrometry they show that C-terminally monophosphorylated BB δ interact with SHP-1 phosphatase which was a new interesting finding from this work.

Overall, the work indicates that incorporation BB δ into CAR design instead of BB ζ improves overall efficiency in T cell response and tumor killing and should be of interest to cancer immunologist and immunologists in general. Although this article studied in detail the contribution of each CD3 ICDs to CAR T cell efficiency other recent work has studied such contributions with focus on CD3 ϵ (Wu et al., Cell, 2020) which questions the novelty of this work. Nonetheless, the scientific work is thorough, clearly presented, technically and overall statistically sound. The work describe interesting findings useful for new CAR design and could potentially move the CAR-T field forward. The following major and minor concerns must be addressed.

Major concerns:

- 1) In the abstract, the authors claim "The contribution of the other chains of the TCR complex, namely CD3 δ , CD3 ϵ , and CD3 γ in a CAR format remains unknown" (Line 30). This is misleading as Wu et al., Cell, 2020 had studied adding CD3 ϵ to CAR setup which lead to better tumor control.
- 2) In Fig 1c, there are some differences in surface expressions of CAR (based on the MFI). The cells should have been sorted for equal surface expression.
- 3) In line 124, the authors claim BB ζ cells exhibited a lower proportion of naïve T cells (Fig.2b). However, from the figure the error bars are too high for all 4 cell types to claim this. This has to be addressed and statistics has to be provided. Fig.2f has no error bars.
- 4) In line 351, the authors say "killing potential of BB ϵ Δ PRS Δ ARK was similar to BB ϵ Δ ARK". However, from the Fig.6d, % killing for BB ϵ Δ ARK is lower than BB ϵ Δ PRS Δ ARK. This has to be clarified.
- 5) In fig. 6f, certain datasets are missing, namely BB ϵ Δ PRS Δ ARK, BB ϵ Δ BRS Δ ARK from Donor 1 and BB ϵ Δ ITAM, BB ϵ Δ PRS from Donor 2.
- 6) To confirm binding between CD3d-YpY and SHP-1, SPR or similar binding analysis should be performed and KD measured.

Minor changes:

- 1) Line 152 and 184 states that the cells were sorted. However, the markers used for sorting is not mentioned and should be included.
- 2) The font size in Extended Fig. 3c is very small.
- 3) References are missing in several places in the text – e.g, Lines 364, 406, 408.

Reviewer #2:

Remarks to the Author:

The authors generate a panel of CARs that relace the TCRz module used in the first generation of CARs with the other three ITAM sequences. Other permutations of this have been tested, but the theme that sets this work apart is the idea that attenuating CAR function may actually result in somewhat better tumour control. An important aspect of the constructs that is not investigated or even acknowledged is that the zeta-zeta dimer is a natural module or the TCR, whereas ϵ - ϵ , γ - γ and δ - δ are not natural dimeric modules of the TCR and thus may take on different characters as homodimers.

1. The in vivo data are key to the paper. A number of studies have demonstrated improvement on the BB-zeta second gen CAR in th NALM6 model. This study shows that a BB-delta shows an incremental improvement at a sub-optimal dose of cells. Is this effect boosted or lost at the normal dose. Perhaps the BB-delta CAR just has a different optimal dose compared to BB-zeta.
2. The expression of BB-delta and BB-gamma are lower than BB-zeta or BB-epsilon. Both BB-delta have LL motifs and the one in delta is known to have a role in quality control for partial TCR complexes. Is dimerization through CD8 sequences necessary for BB-delta to be expressed on the surface?
3. The equivalent cytotoxicity is surprising given the lack of Ca²⁺ signalling in the BB-delta and BB-gamma. Is the killing perforin and granzyme dependent and does it correspond to CD107A upregulation? The lack of predictive power of the SLB system may be due to lack of CD58 in the bilayer, which augments Ca²⁺ signalling.
4. The gene expression analysis is written in a confusing manner. In the same paragraph the authors state no tonic signalling based on PCA, but then suggest tonic signalling based on differential expression. Based on the Cytof the authors should have data to validate some of the gene expression hits. Can they call attention to data that would support the significance of any of these effects. There are also ways to validate some metabolic trends at the single cell level that could be undertaken to support the gene expression. The abstract suggest that these gene expression changes provide mechanistic insight, but seems more correlative.
5. The purpose of the deeper analysis of the BB-epsilon motif in Figure 6 is not clear. They seem to create a minimal epsilon construct with some feature of the BB-delta and BB-gamma, for example, with low Ca²⁺ signalling on bilayers. Can they test this in vivo to determine if it works similarly to BB-delta?
6. In Fig 7d, its surprising that the pull down with peptides corresponding to delta pY-pY shows attenuated SHP-1 binding and only the Y-pY binds SHP-1 strongly. I see why this would happen the CAR in cells where the ITAM is limiting and ZAP-70 can outcompete SHP-1, but can't the experiment

be done with enough beads to bind both ZAP-70 AND SHP-1?

7. SHP1 has two SH2 domains that can operate independently so might be recruited to dimeric hemi-ITAMs. Is the recruitment of SHP-1 to the BB-delta FY CAR dependent upon dimeric nature of the CAR?

The statistical analysis should be reviewed. In fig 5 b and c, for example, there are limited points (3) that appear non-normally distributed and that overall are still starred as significant. This doesn't seem possible with a non-parametric test and would certainly be marginal.

Author Rebuttal to Initial comments

See inserted PDF

NI-A35052A, "Harnessing CD3 diversity to optimise CAR T cells"**Point-by-point response****Reviewer #1**

This research article by Velasco Cardenas et al. investigates how individual CD3 chains of the TCR-CD3 complex can improve/affect CAR T cell performance. The authors generated and analysed variants of the FDA-approved CAR Kymriah (4-1BB/ ζ), in which the CD3 ζ intercellular domain was systematically exchanged for each of the CD3 $\delta/\epsilon/\gamma$ intercellular domains (ICDs).

As an initial result, they showed that CD3 $\delta/\epsilon/\gamma$ ICDs can generate functional CAR T cells by using a luciferase-based killing assay. All the generated CD3 ICDs had anti-tumour activity but CD3 δ had the best anti-tumour function. Next, they identified the role of CD3 ICDs in T cell activation outcome for the CAR T cells studied. BB ζ and BB ϵ CARs exhibited a higher signalling capacity whereas BB ζ and BB ϵ showed least response in terms of cytokine production and Ca²⁺ signalling. Then, the authors characterized the transcriptional programming of each modified CAR T cells upon antigen stimulation and data showed different transcription signatures for each CAR T cell type with BB δ ICD indicating memory T cell differentiation preference. A new repetitive-challenge protocol was formulated and UMAP analysis was performed to identify up/down-regulated genes, which showed BB δ cells maintain the best ability to kill tumour cells followed by BB γ cells. They then generated mutated versions of BB ϵ cells and studied their killing abilities. BB ϵ Δ PRS didn't result in reduction of killing ability suggesting that the PRS motif plays little role in T cell cytotoxicity. Finally, by SILAC labelling and mass spectrometry they show that C-terminally monophosphorylated BB δ interact with SHP-1 phosphatase which was a new interesting finding from this work.

Overall, the work indicates that incorporation BB δ into CAR design instead of BB ζ improves overall efficiency in T cell response and tumour killing and should be of interest to cancer immunologist and immunologists in general. Although this article studied in detail the contribution of each CD3 ICDs to CAR T cell efficiency other recent work has studied such contributions with focus on CD3 ϵ (Wu et al., Cell, 2020) which questions the novelty of this work. Nonetheless, the scientific work is thorough, clearly presented, technically and overall statistically sound. The work describes interesting findings useful for new CAR design and could potentially move the CAR-T field forward. The following major and minor concerns must be addressed.

Response: We thank the Reviewer for his/her thoughtful synthesis, for highlighting our major findings and for his/her positive comments towards our work. We are happy that he/she recognizes the importance of this work to move the CAR-T field forward.

Regarding the novelty of our study, the work of Wu et al., Cell, 2020 studied the contribution of the CD3 ϵ ICD when added to a ζ CAR (the CAR contained both ICDs, CD3 ϵ and ζ). In contrast to our work, Wu et al. did not study whether the CD3 ϵ ICD alone generates functional CARs. In all, we have for the first time studied the contribution of each CD3-ICD independently of the ζ -ICD and demonstrated that the presence of the ζ -ICD is not essential to generate functional CAR T cells¹. Thus, our approach and our findings are both novel. A more detailed answer is provided below (see answer to point 1).

Major concerns:

1) In the abstract, the authors claim “The contribution of the other chains of the TCR complex, namely CD3 δ , CD3 ϵ , and CD3 γ in a CAR format remains unknown” (Line 30). This is misleading as Wu et al., Cell, 2020 had studied adding CD3 ϵ to CAR setup which lead to better tumour control.

Response: This valuable feedback made us realize that this point needs to be more clearly explained, in order to set our manuscript apart from previous works and thus, to highlight the novelty of the present piece. The works from Wu et al. (Cell, 2020; as indicated by the Reviewer) and of Salter et al. (Science Signaling, 2021) as well as our previous work (Hartl, et al, Nature Immunology 2020) have studied the contribution of the CD3 ϵ chain to CAR T cells¹⁻³. However, in these three publications the CD3 ϵ ICD has been added to the framework of existing ζ -CAR constructs containing the ζ ICD with its three ITAMs. Hence, these CARs simultaneously contain the ζ and the CD3 ϵ ICDs. Thus, our study is the first one to design and characterize CAR constructs containing only the CD3 ϵ ICD. Moreover, we have created additional CARs containing only the CD3 γ or CD3 δ ICD. This novel approach has allowed us to exclusively study the contribution of each CD3 chain without the confounding effects introduced by the presence of the three ζ ITAMs.

To clarify this point, we have taken the following actions:

- We have made this point clearer in the Abstract, page 1, lines 29-33 as follows: “Current FDA-approved CAR T cells harbour the TCR-derived ζ chain as intracellular activation domain in addition to co-stimulatory domains. The functionality in a CAR format of the other chains of the TCR complex, namely CD3 δ , CD3 ϵ , and CD3 γ , instead of ζ remains unknown. Here, we have systematically engineered novel CD3 CARs, each containing only one of the CD3 intracellular domains.”
- We have made this point clearer in the Discussion, page 13, lines 480-487 as follows: “Additional reports have demonstrated that introducing the CD3 ϵ ICD into ζ -CARs is beneficial for preclinical CAR T cell therapy. These studies introduced a complete or partial CD3 ϵ ICD in combination with the ζ ICD, increasing the number of ITAMs but simultaneously introducing novel motifs that are lacking in the ζ ICD. In contrast, we have studied the contribution of each individual CD3-ICD independently of the ζ -ICD and demonstrated that the presence of the ζ -ICD is not essential to generate functional CAR T cells.”

2) In Fig 1c, there are some differences in surface expressions of CAR (based on the MFI). The cells should have been sorted for equal surface expression.

Response: We acknowledge this well-taken comment and have previously tried to sort the cells for equal CAR surface expression. However, the surface expression of each CAR construct is intrinsically linked to the construct itself (see below, mutations in the di-leucine and ER-retention motifs), and even sorted cells for a given level of CAR expression quickly return to their previous CAR expression level as the ones, such as those shown in Fig 1. Nevertheless, the differences in CAR expression were not significant among different healthy donors as shown in Fig. 1d. We have now clarified the fact that the cells shown in Figure 1 are indeed sorted cells in the corresponding figure legend (page 26, line 855).

3) In line 124, the authors claim BB ζ cells exhibited a lower proportion of naïve T cells (Fig.2b). However, from the figure the error bars are too high for all 4 cell types to claim

this. This has to be addressed and statistics has to be provided. Fig.2f has no error bars.

Response: As indicated by the Reviewer, the error bars shown in the old Fig. 2b are too high and indeed the differences are not significant when independent donors were analysed together. However, a lower proportion of naïve T cells in the BBζ CAR expressing cells was observed for each individual donor. We have revised this figure by analysing more donors. The new analysis (new Fig. 2b) was performed with 6 independent donors. The results show that expression of BBζ significantly reduced the percentage of naïve T cells when compared to Mock cells (Two-way ANOVA, Sidak's multiple comparisons test, $p= 0,0142$). Likewise, the proportion of non-naïve T cells was significantly increased for BBζ when compared to Mock cells (Two-way ANOVA, Sidak's multiple comparisons test, $p= 0,0093$). We have included the new figure and adapted the figure legend (page 26, lines 867-869).

New Fig. 2b: ... (b) Phenotype of CAR T cells after 48 hours of stimulation with Nalm6 (1:1 ratio) (n=6). TCM=T Central Memory, TEM=T Effector Memory, TEMRA=T Effector Memory cells re-expressing CD45RA ...

Regarding Figure 2f, it does not have error bars because this is a representative experiment out of 3 independent experiments. We have indicated this in the corresponding figure legend (page 26, line 875). The difficulty in providing error bars is manifold: the calcium responses vary to some degree between donors and between the well-being of CAR T cells in the stimulation cycle. The experimental settings are highly time-consuming, and it is not possible to do more than one donor at the time. Additionally, the SLBs have to be generated new each time, and some variability with regards to antigen densities (which are verified with each run) is introduced into the system. Therefore, showing one representative experiment out of 3 independent experiments has proven to be the most suitable approach⁴.

4) In line 351, the authors say “killing potential of BBεΔPRSDRK was similar to BBεΔRK”. However, from the Fig.6d, % killing for BBεΔRK is lower than BBεΔPRSDRK. This has to be clarified.

Response: We acknowledge the need to clarify this point. The old Fig. 6d shows a representative donor out of 2-5 donors tested. In this particular donor, the killing for BBεΔRK was lower than BBεΔPRSDRK. However, this result was not reproduced in all donors. Our analysis did not reveal any statistically significant difference between BBεΔRK and BBεΔPRSDRK ($p=0,0997$, one-way ANOVA) when the independent donors were pooled. We have now prepared a new figure and adapted the corresponding figure legend (page 28, lines 945-948).

New Fig. 7d (in place of the old Fig. 6d): ... Specific killing of CD19⁺ Nalm6 cells by primary human CAR T cells (E:T ratio 1:1 for 6-8 hrs). Each dot represents an independent donor (n=2-5). Only statistically significant comparisons are shown. Data are represented as mean ± SD and analysed by One-way ANOVA and Dunnett's multiple comparisons test, *p < 0.05, **p < 0.01, ****p < 0.0001.

5) In fig. 6f, certain datasets are missing, namely BBεΔPRSΔRK, BBεΔBRSΔRK from Donor 1 and BBεΔITAM, BBεΔPRS from Donor 2.

Response: Yes, the Reviewer is correct. In the generation of such a big panel of CAR T cells some transductions did not work efficiently, and thus for donor 1 the generation of cells expressing BBεΔPRSΔRK and BBεΔBRSΔRK failed, while for donor 2 the generation of cells expressing BBεΔITAM and BBεΔPRS failed. This is the reason why we show both donors for the sake of completeness.

6) To confirm binding between CD3d-YpY and SHP-1, SPR or similar binding analysis should be performed and KD measured.

Response: We acknowledge this comment and agree that it would be interesting to know the affinity of SHP-1 binding to CD3δ -YpY. However, we are confident that the CD3δ-SHP-1 interaction as such is shown with sufficient confidence. We provide manifold evidences; in pull-down assays, we were able to demonstrate the interaction using SHP-1 and CD3δ-derived peptides and detect this interaction by MS (now Fig. 8c) and by western blot (new Fig. 8d). Additionally, we confirmed this finding by immunoprecipitating BBδ CARs and detecting bound SHP-1 (now Fig. 8f and 8g). Concerning SPR measurements, we have previously tried to do so in collaboration with an expert in the field. Yet, the assay is extremely tricky, since SHP-1 dephosphorylates the CD3δ-YpY peptide and the binding can then not be detected since phosphorylation is required (new Fig. 8). Inhibiting SHP-1's catalytic activity with pervanadate did not help, since under these conditions SHP-1 got sticky preventing the SPR analysis. We have tried to repeat the assay at different temperatures - again without success.

Still, we have now validated the binding between CD3δ -YpY and SHP-1 using primary human CAR T cells expressing CAR constructs that can only be phosphorylated at the C-terminal tyrosine of the ITAM (BBδFY). Cells were then stimulated with pervanadate to ensure optimal phosphorylation. We subsequently performed a CAR immunoprecipitation to evaluate the binding of SHP-1 to each construct. SHP-1 preferentially bound to the CARs containing monophosphorylated BBδ as shown in new Fig. 8g.

New text additions are to be found on pages 13, lines 447-451 and page 16, lines 569-571 and page 21, lines 769-770 and 777-778 (methods) as well as page 29, lines 970 and 974 (Figure legend) Text additions are highlighted in blue.

New data are to be found in **new Figure 8g**.

Minor changes:

1) Line 152 and 184 states that the cells were sorted. However, the markers used for sorting is not mentioned and should be included.

Response: We apologize for not having mentioned this before; the cells were sorted by GFP expression. This information is now introduced on page 5, line 162 as follows: *“CAR T cells were sorted for GFP⁺ six days after transduction...”*

2) The font size in Extended Fig. 3c is very small.

Response: The font size is now increased.

3) References are missing in several places in the text – e.g., Lines 364, 406, 408.

Response: References are now included.

Reviewer #2

(Remarks to the Author)

The authors generate a panel of CARs that replace the TCR ζ module used in the first generation of CARs with the other three ITAM sequences. Other permutations of this have been tested, but the theme that sets this work apart is the idea that attenuating CAR function may actually result in somewhat better tumour control. An important aspect of the constructs that is not investigated or even acknowledged is that the zeta-zeta dimer is a natural module of the TCR, whereas epsilon-epsilon, gamma-gamma, and delta-delta and are not natural dimeric modules of the TCR and thus may take on different characters compared to the natural heterodimers, such as epsilon-gamma and epsilon-delta."

Response: We thank the Reviewer for his/her thoughtful synthesis and appreciation of the manuscript. We recognize that we had not acknowledged that the ζ - ζ homodimer is a natural module of the TCR and that, in contrast, the other CD3 chains are not naturally found as homodimeric modules. The CAR constructs used in our study are based on the FDA-approved CAR Kymriah (4-1BB/ ζ) and thus contain an extracellular linker and the transmembrane domain of human CD8 α (amino acids 138-206). This linker contains two cysteines that in the endogenous CD8 α molecule drive the formation of CD8 α -CD8 α homodimers⁵. Previous studies have addressed the role of the CD8 α dimerization in the context of ζ -based CARs to conclude that dimerization was only important to mediate successful activation of low-affinity CARs. In high-affinity CARs, a single CD3 ζ domain is indeed sufficient to promote efficient signaling⁶. The scFv used in our study (called FMC63) has a high affinity for CD19⁷ and therefore dimerization should not affect the signalling efficiency in the context of the ζ -based CAR. Still, we acknowledge that it would be important to study the role of dimerization in the context of our novel CD3-based CARs. To this end, we have now mutated the two extracellular cysteine residues in the CD8 α hinge region to serine residues in the novel CD3-based CARs. We have then compared these monomeric CAR versions to the non-mutated dimeric CARs with respect to their surface expression and functionality *in vitro* (New Fig. 6 and new Extended Data Fig. 7).

In addition, and motivated by question 2 (see below), we have also mutated the membrane proximal di-leucine motif (L153 and L154) in BB γ . In the context of the complete TCR complex, this di-leucine motif in the CD3 γ cytoplasmic tail is required, together with the phosphorylation of a closely located serine (S148), for TCR internalization and subsequent recycling in response to antigen-induced TCR signaling⁸. Further, this di-leucine motif, in the context of the TCR, is important for controlling T cell homeostasis and for responding to viral infections^{9,10}. The CD3 δ cytoplasmic tail also contains a membrane proximal di-leucine motif, however, it lacks the preceding serine at position -5 (new Extended Data Fig 1a and new Fig. 6a). We have not found studies mutating the di-leucine motif in CD3 δ . However, some previous studies using tailless chimeras of CD3 γ and CD3 δ suggested that amino acid sequences presented in their cytoplasmic tails control ligand-induced TCR downregulation¹¹.

Thus, we have generated the following new constructs: monomeric versions of CARs containing the cytoplasmic tails of CD3 δ , CD3 ϵ , CD3 γ and ζ by mutating the cysteine residues in the CD8 α hinge region, as well as dimeric and monomeric versions of the BB γ and BB δ CARs with the di-leucine motif mutated (named Δ LL in the manuscript). These 12 CAR constructs were expressed in primary human T cells from 5-8 independent healthy donors (new Fig. 6b,c and Extended Data Fig. 7).

Interestingly, the monomeric version of BB ϵ CAR was not detected on the surface of human primary T cells (new Extended Data Fig. 7a-c). In line with this observation,

BB ϵ CAR T cells failed to kill CD19⁺ Nalm6 cells as well as to secrete IFN- γ and TNF- α upon co-incubation with CD19⁺ Nalm6 cells (new Extended Data Fig. 7d, e, f). The cytoplasmic tail of CD3 ϵ has an endoplasmic reticulum (ER) retention motif containing the last five amino acids of CD3 ϵ (NQRRI) (new Extended Data Fig. 1a and 7a). Cell transfection studies and *in vivo* animal models have demonstrated that this retention motif in CD3 ϵ is dominant to control TCR surface expression^{12,13}. We have deleted the ER retention motif in the BB ϵ CAR in both the dimeric and in the monomeric constructs in an attempt to recover surface expression. However, deletion of the ER retention motif failed to recover surface expression for monomeric BB ϵ (new Extended Data Fig. 7g), suggesting the existence of additional retention signals in the cytoplasmic tail of CD3 ϵ that are only overridden upon the formation of dimers. Taken together, these data suggest that the un-natural ϵ - ϵ dimers are needed for the expression of the BB ϵ CARs on the cell surface. We hypothesize that dimeric formation serves to override retention signals located in the cytoplasmic tail of CD3 ϵ beyond the NQRRI sequence.

In contrast to the monomeric BB ϵ CAR, the monomeric versions of BB ζ , BB δ and BB γ CARs were detected well on the cell surface of primary human T cells (new Fig. 6b). Indeed, no statistically significant differences were observed for the expression of monomeric or dimeric BB ζ CARs. Conversely, the monomeric versions of BB δ and BB γ CARs were significantly better expressed on the surface than their dimeric counterparts (new Fig. 6c). Mutation of the di-leucine motif increased the cell surface expression of the dimeric and monomeric versions of BB δ and BB γ (new Fig. 6c). These data suggest, on the one hand, that the un-natural δ - δ and γ - γ dimers limit the number of CARs present on the cell surface. On the other hand, that the di-leucine motif in BB δ and BB γ reduces surface levels of the CARs as it does in TCRs. These results show that the di-leucine motif located in the CD3 δ cytoplasmic tail regulates receptor expression on the cell surface despite the absence of the serine in position -5. The di-leucine motif seems to be the key regulator for the expression of BB δ and BB γ , since once mutated, no significant differences were observed in the expression of the monomeric and dimeric CARs.

Regarding the functionality of the new constructs, tumour cell killing, expression of activation markers (CD25, CD69, and 4-1BB) and cytokine secretion (IFN- γ and TNF- α) upon co-incubation with CD19⁺ Nalm6 cells were analysed (new Fig. 6d-h).

The specific tumour cell killing did not differ between monomeric or dimeric BB ζ , BB δ or BB γ CARs, regardless of the di-leucine motif mutations in BB δ or BB γ (new Fig. 6d). Among dimeric CARs, the specific killing was significantly augmented when the di-leucine motif was mutated (new Fig. 6d), correlating with the highest increase in CAR surface expression (3-fold for BB δ , 2-fold for BB γ). Likewise, no differences were detected in degranulation between T cells expressing monomeric or dimeric BB ζ , BB δ nor BB γ (new Fig. 6e). Among the di-leucine motif mutants, monomeric BB δ and BB γ degranulated less than the dimeric CAR forms (new Fig. 6e). Among dimeric CARs, the specific killing was significantly increased when the di-leucine motif was mutated (new Fig. 6d). Still, in all of these comparisons the differences were statistically significant but extremely mild, questioning their biological significance since BB δ and BB γ are remarkably inefficient in inducing degranulation (new Fig. 2g,h and Fig. 6e).

We did not observe any differences between monomeric or dimeric BB ζ CARs, and only mild differences for BB δ or BB γ , concerning CD25, CD69 or 4-1BB upregulation upon contact with tumour cells (new Fig. 6f). Among dimeric BB δ or BB γ CARs, the percentage of cells expressing CD69 and 4-1BB (but not CD25) significantly increased when the di-leucine motif was mutated (new Fig. 6f). Lastly, monomeric BB ζ CARs showed significantly reduced production of IFN- γ and TNF- α upon activation, indicating that dimeric ζ forms are more efficient to transmit activation signals engaging cytokine secretion (new Fig. 6g,h). We did not observe statistically significant differences between the BB δ or BB γ CARs regarding IFN- γ and TNF- α production (new Fig. 6g,h).

Taking together, the natural dimeric ζ forms are more efficient to transmit activation signals engaging secretion of cytokines, while the monomeric counterparts are performing similarly well regarding degranulation and up-regulation of activation markers. These data suggest, that dimeric ζ forms are needed to optimally activate T cell effector functions. Regarding δ and γ , unnatural homodimeric forms increased the expression of BB δ and BB γ CARs on the cell surface, but did not remarkably impact T cell activation. Mutation of the di-leucine motif in BB γ , but also in BB δ , resulted in a notable increase in CAR expression, specific killing, expression of CD69 and 4-1BB as well as a tendency to increase cytokine secretion that did not reach statistical significance. These effects were best observed in the context of the dimeric forms. In summary, the unnatural homodimeric versions of ϵ are needed in the CAR context for surface expression. The unnatural dimeric forms of γ and δ did not impact T cell activation, but increased the expression of CARs on the cells surface that might be important in scenarios of limited antigen. Di-leucine-mutated dimeric forms of γ and δ are better expressed and more efficient in killing tumour cells *in vitro*, and therefore could be beneficial and are recommended for therapeutic approaches.

New text additions are to be found on pages 10, lines 362-372 and page 13, lines 447-451, page 13, lines 483-487, page 16, lines 569-575 (methods) and page 28, lines 929-942 (Figure legend). Text additions are highlighted in blue.

New data are to be found in **new Figure 6** and **new Extended Data Fig. 7**.

1. The *in vivo* data are key to the paper. A number of studies have demonstrated improvement on the BB-zeta second gen CAR in the NALM6 model. This study shows that a BB-delta shows an incremental improvement at a sub-optimal dose of cells. Is this effect boosted or lost at the normal dose? Perhaps the BB-delta CAR just has a different optimal dose compared to BB-zeta.

Response: We thank the Reviewer for this comment. As mentioned by him/her, many studies have assayed alternative approaches to improve the efficacy of the BB ζ CARs using different T cell doses. As a general conclusion, clear differences between different constructs at a sub-optimal dose were lost by increasing the T cell dose^{2,14,15}. Thus, we expect the same outcome when increasing the dose of CAR T cells in our study. For the sake of the 3R recommendations, we would prefer not to have to do these *in vivo* experiments.

Of note, in tumour patients, which usually have gone through multiple rounds of chemotherapy and/or radiotherapy, it is often difficult to obtain large numbers of CAR T cells. Thus, our approach to use sub-optimal doses in the pre-clinical mouse experiments mimics better the real clinical setting than the use of an optimal dose¹⁶⁻¹⁹

2. The expression of BB-delta and BB-gamma are lower than BB-zeta or BB-epsilon. Both BB-delta have LL motifs and the one in delta is known to have a role in quality control for partial TCR complexes. Is dimerization through CD8 sequences necessary for BB-delta to be expressed on the surface?

Response: We thank the Reviewer for his/her comment and also felt that a study of the role of dimerization and the LL motifs would be important. Thus, we have mutated the two extracellular cysteines in the CD8 α hinge to serines and assayed surface expression and function of our CD3-based CARs. In addition, we have mutated the di-leucine motif described in the cytoplasmic tail of CD3 γ , and the homologous motif in CD3 δ . We found that both, the LL motif and dimerization, indeed played a role. A detailed description can be found above (pages 9-11).

New text additions are to be found on pages 10, lines 362-372 and page 13, lines 447-451, page 13, lines 483-487, page 16, lines 569-575 (methods) and page 28, lines 929-942 (Figure legend). Text additions are highlighted in blue.

New data are to be found in **new Figure 6** and **new Extended Data Fig. 7**.

3. The equivalent cytotoxicity is surprising given the lack of Ca²⁺ signalling in the BB-delta and BB-gamma. Is the killing perforin and granzyme dependent and does it correspond to CD107A upregulation? The lack of predictive power of the SLB system may be due to lack of CD58 in the bilayer, which augments Ca²⁺ signalling.

Response: We thank the Reviewer for his/her comment. We have performed degranulation assays to test whether the killing mediated by the CD3-based CARs corresponds to CD107a up-regulation (new Fig. 2g,h). Indeed, and consistent with the argumentation of the Reviewer and the lack of detectable Ca²⁺ signalling, BBδ and BBy CAR T cells failed to efficiently degranulate upon co-incubation with CD19⁺ Nalm6 tumour cells for 4 hours (new Fig. 2g). We repeated the experiment by increasing the co-incubation time to consider the possibility that T cells expressing the BBδ or BBy CARs degranulate with slower kinetics. However, even upon 8 hours of co-incubation (same time that was used for our regular killing assays), BBδ and BBy CARs failed to efficiently degranulate (new Fig. 2h). Of note, the graphs in new Fig. 2 show CD4 and CD8 cells gated together, the same results were obtained by gating these two populations independently.

We next investigated the role of Fas and TRAILR1/2 in the killing of tumour cells by the four CAR T cells (BBζ, BBε, BBδ and BBy) by performing cytotoxicity experiments using blocking antibodies. We found that killing by none of the CAR T cells involves TRAILR1/2 interactions with its ligand DR5 (new Fig. 2i). In contrast, killing by all CAR T cells involves Fas-FasL interactions, however, to different extents depending on the CAR construct. Blocking Fas-FasL interactions reduced the killing by 18% for BBζ, 48% for BBε, 39% for BBδ, and 47% for BBy. Taken together these data suggest that the cytoplasmic tails of the TCR induced killing of target cells by different mechanisms, being the killing by ζ mainly induced by degranulation. Nevertheless, the killing mediated by CD3ε is a combination of degranulation and Fas-mediated killing. Lastly, the cytoplasmic tails of CD3δ and CD3γ failed to efficiently induce Ca²⁺ signalling and consequently degranulation. However, they achieve killing of target cells by using alternative mechanisms such as Fas-FasL interactions (shown in new Fig. 2). Alternatively, they might use other mechanisms, such as TRAIL interacting with DR4, that could not be tested due to the lack of blocking antibodies.

Taken together, we thank the Reviewer for encouraging us to perform these experiments, given that we have unravelled previously unknown links between the different cytoplasmic tails of the TCR and the killing mechanisms used by T cells. Thus, our data might provide new molecular insights of how TCR activation results in perforin- and Fas ligand-mediated killing working in synergy to achieve efficient elimination of target cells²⁰.

Regarding the supported lipid bilayer (SLB) approach and the calcium influx, it has been shown that ICAM-1 or CD58 are sufficient to enhance antigen sensitivity and discrimination for the TCR²¹ In fact, for the BBζ CAR T cells, ICAM-1 is enough to support calcium flux by as little as one CD19 molecule per μm² (Fig. 2f). All CARs used in our study have the same scFv and therefore the same affinity for CD19. It is possible that adding CD58 to the SLB will further increase the sensitivity of the system and will allow us to detect calcium influxes above background for BBy and BBδ. However, the new degranulation data (new Fig. 2g,h) provide further support for the idea that BBδ and BBy failed to efficiently induced Ca²⁺ signalling, and consequently, did not degranulate. In all, these data endorse our message that these two constructs have a

reduced potential to transmit downstream signals when compared to BB ϵ and BB ζ (Fig. 2, Fig. 3).

New text additions are to be found on pages 4-5, lines 150-159, page 17, lines 629-632 (methods) and page 26, lines 879-889 (Figure legend). Text additions are highlighted in blue.

New data are to be found in **new Fig. 2g,h,i**.

4. The gene expression analysis is written in a confusing manner. In the same paragraph the authors state no tonic signalling based on PCA, but then suggest tonic signalling based on differential expression. Based on the Cytof the authors should have data to validate some of the gene expression hits. Can they call attention to data that would support the significance of any of these effects? There are also ways to validate some metabolic trends at the single cell level that could be undertaken to support the gene expression. The abstract suggest that these gene expression changes provide mechanistic insight, but seems more correlative.

Response: We appreciate the Reviewer's comment; his/her valuable feedback made us realize that the description of the gene expression analysis needs to be more clearly explained to demonstrate the significance of our findings.

Regarding the tonic signalling; we have improved our wording to make our message clear that the tonic signalling of any of the CD3-ICD is not sufficient to change the global transcriptome of the CAR-T cells among different healthy donors to an extent that could be detected.

On page 5, lines 163-169 "*Principal component analysis (PCA) demonstrated that the constructs clustered by donor but not by cytoplasmic tail. This suggests that the expression of a given ICD by itself does not globally change the characteristic transcriptome signature beyond donor to donor variation (Extended Data Fig. 3a). Therefore, we performed a Differential Expressed Genes (DEG) analysis that revealed differences depending on the TCR-derived ICD indicative of differential tonic signalling of the CARs.*"

Regarding the CyTOF and the validation of gene expression hits: The CyTOF analysis and the transcriptome analysis were done at different stimulation conditions and different times upon CAR expression, and therefore, cross-validation between these two data sets is not possible. Still, we have validated some of our CyTOF hits using alternative methods, such as flow cytometry and ELISA for the secretion of cytokines. For instance, our CyTOF results suggested that repetitively challenged CAR T cells produced less pro-inflammatory cytokines (IFN- γ , TNF- α , IL-2) than non-challenged (Fig. 4b). Indeed, when each of the constructs was analysed separately, each of them showed a clear decrease in the production of those cytokines (Fig. 4d). We have validated these results using ELISA (new Fig. 4e), confirming functional dysfunction upon repetitive stimulations. In line with these results, the CyTOF results showed a higher expression of the inhibitory receptors PD-1 and CTLA4 in re-challenged cells expressing BB ζ or BB ϵ (new. Extended Data Fig. 6b). These observations have been validated by flow cytometric analysis (new. Extended Data Fig. 6c). In addition, the proportion of cells expressing TCF-1 was highest in BB δ and lowest in BB ζ CAR T cells as assayed by CyTOF (Fig. 5a,b). We have now validated these results by intracellular flow cytometric experiments (new Fig. 5c), further supporting that T cells expressing BB δ CAR best retain functionality, specific killing and maintenance of precursor populations expressing high levels of the transcription factor TCF-1. Thus, providing thus an explanation for the better lasting performance of BB δ CAR T cells in vivo (Fig. 1).

New text additions are to be found on pages 8, lines 292-300 and page 9, lines 315-316, page 20, line 746-747 (methods) as well as page 27, lines 907-909 and lines 916-917 (Figure legends). Text additions are highlighted in blue.

New data are to be found in **new Fig. 4e, new Fig. 5c and new Extended Data Fig. 6b,c.**

5. The purpose of the deeper analysis of the BB-epsilon motif in Figure 6 is not clear. They seem to create a minimal epsilon construct with some feature of the BB-delta and BB-gamma, for example, with low Ca²⁺ signalling on bilayers. Can they test this *in vivo* to determine if it works similarly to BB-delta?

Response: We appreciate this comment and have now described better the purpose of the deeper analysis of the CD3ε motifs in old Figure 6 (new Fig. 7 and new Extended Data Fig. 7).

On page 11, lines 387-391: *“We previously reported that combining CD3ε and ζ ICD into a BB-based CAR improved tumour therapy in a preclinical mouse model. However, the CD3ε ICD itself is sufficient to generate a functional CAR that outperformed the FDA-approved BBζ CAR *in vivo* (Fig. 1). The CD3ε ICD shows the highest number of known signalling motifs, however, their individual contributions to CAR signalling in the absence of the ζ ICD, has not yet been studied.”*

Indeed, the BBεΔBRSΔRK construct exhibits reduced calcium influx similar to the BBγ or the BBδ construct. However, in contrast to those, it also showed a strong reduction in cytotoxicity *in vitro*. To the best of our knowledge, there is not a single CAR construct that showed reduced killing *in vitro* but increased performance *in vivo* in the literature. Taken together and for the sake of the 3R recommendations, we would prefer not to perform the *in vivo* experiments suggested.

In the revised version, the characterization of the BBε CAR has been expanded by addressing the role of dimerization in the functioning of this CAR (new Extended Data Fig. 7). As described in detail above (pages 9-10), the monomeric version of the BBε CAR failed to be expressed on the surface of primary human T cells, despite the deletion of the ER retention motif located within CD3ε (new Extended Data Fig. 7). In all, these data suggest that the un-natural ε-ε dimers are needed for effective expression of the BBε CARs on the cell surface of primary human T cells. We hypothesized that dimeric formation serves to override ER retention signals located in the cytoplasmic tail of CD3ε beyond the one containing the C-terminal NQRR1 sequence (for a detailed answer please check page 5).

New text additions are to be found on page 11, lines 373-384 as well as 387-391, and page 16, lines 571-575 (methods) and pages 30-31, lines 1038-1049 (Figure legend). Text additions are highlighted in blue.

New data are to be found in **new Extended Data Fig. 7.**

6. In Fig 7d, it is surprising that the pull down with peptides corresponding to delta pY-pY shows attenuated SHP-1 binding and only the Y-pY binds SHP-1 strongly. I see why this would happen the CAR in cells where the ITAM is limiting and ZAP-70 can outcompete SHP-1, but can't the experiment be done with enough beads to bind both ZAP-70 AND SHP-1?

Response: We have used the beads and peptides in huge excess over ZAP-70, to prevent any competition between SHP-1 and ZAP-70 in binding to the peptides. We have now made this clear in the revised version of the manuscript (page 12, lines 441-442).

We provide here some calculations to support our statement:

ZAP-70: we used the lysate of 2×10^7 T cells. Since each T cell has about 1×10^6 ZAP-70 molecules, the lysate contains 2×10^{13} ZAP-70 molecules.

Peptide: the biotinylated CD3 δ -derived peptides (Fig. 7a) have a molecular weight of 5675 Da (g/mol). We used 2 micrograms of each peptide, corresponding to 0.01135 mol or 6.8×10^{21} molecules.

Thus, the peptides were in 3.4×10^8 excess. In words, this is about 340 million times more.

Thus, ZAP-70 did not outcompete SHP-1 in binding to the doubly phosphorylated ITAM tyrosines. Thus, it could be structural changes in the ITAM or a long-range effect of the phosphate group charge that prevented SHP-1 from binding.

New text additions are to be found on page 12, lines 441-442 and page 14, lines 519-521. Text additions are highlighted in blue.

7. SHP1 has two SH2 domains that can operate independently so might be recruited to dimeric hemi-ITAMs. Is the recruitment of SHP-1 to the BB-delta FY CAR dependent upon dimeric nature of the CAR?

Response: We thank the Reviewer for this idea. The experiments shown in the old Fig. 7c, d (now Fig. 8 in the revised version) are done with CD3 δ -derived peptides and showed recruitment of SHP-1 to the monophosphorylated peptide. Under our experimental conditions, the peptides were in proximity of unknown distance and geometry on the beads, but did not exclusively forming dimers, suggesting that the dimeric nature of the CARs might not be a crucial parameter for the recruitment of SHP-1.

In addition, we have assayed this point experimentally by performing CAR immunoprecipitations to evaluate the binding of SHP-1 using primary human CAR T cells expressing the dimeric or the monomeric version of the monophosphorylated BB δ construct (new Fig. 8g). No statistically significant difference was observed between the dimeric and the monomeric forms suggesting that the recruitment of SHP-1 to the BB δ FY CAR is independent of the dimeric nature of the CAR.

New text additions are to be found on pages 13, lines 447-451 and page 21, lines 769-770 and 776-777 (methods) as well as page 29, lines 970 and 974 (Figure legend) Text additions are highlighted in blue.

New data are to be found in **new Figure 8g**.

The statistical analysis should be reviewed. In fig 5 b and c, for example, there are limited points (3) that appear non-normally distributed and that overall are still starred as significant. This doesn't seem possible with a non-parametric test and would certainly be marginal.

Response: We appreciate the Reviewer's comment and checked the statistical analysis as suggested. Samples of the same donor were paired, to decrease the confounding effects of using different healthy donors. We have tested the data sets for Shapiro-Wilk normality test, which is recommended for biological samples with a sample size < 50 ²² The data set showed in Fig. 5b passed the normality test. We have therefore repeated the analysis and applied parametric tests, in particular, we applied paired One-way ANOVA analysis and a Tukey's multiple comparisons test. The new analysis supports the notion that the proportion of cells expressing TCF-1 was highest in BB δ and lowest in BB ζ CAR T cells as assayed by CyTOF (Fig. 5a,b). We have now validated these results by intracellular flow cytometric experiments (new Fig. 5c), and the results support our previous findings. Likewise, we have revised the statistical analysis of the cluster proportions shown in Fig. 5f. Again, this data set has passed the

Shapiro-Wilk normality test. The data are now analysed using paired One-way ANOVA analysis and a Tukey's multiple comparisons test.

Moreover, we have now introduced a new paragraph in the material and methods section describing the statistical analysis (pages 24, lines 820-825). Text additions are highlighted in blue.

References

1. Wu, W. *et al.* Multiple Signaling Roles of CD3 ϵ and Its Application in CAR-T Cell Therapy. *Cell* **182**, 855-871.e23; 10.1016/j.cell.2020.07.018 (2020).
2. Hartl, F. A. *et al.* Noncanonical binding of Lck to CD3 ϵ promotes TCR signaling and CAR function. *Nature immunology* **21**, 902–913; 10.1038/s41590-020-0732-3 (2020).
3. Salter, A. *et al.* Comparative analysis of TCR and CAR signaling informs CAR designs with superior antigen sensitivity and in vivo function. *Science Signalling* **24** (2021).
4. Gudipati, V. *et al.* Inefficient CAR-proximal signaling blunts antigen sensitivity.
5. Hennecke, S. & Cosson, P. Role of transmembrane domains in assembly and intracellular transport of the CD8 molecule. *Journal of Biological Chemistry* **268**, 26607–26612; 10.1016/S0021-9258(19)74355-5 (1993).
6. Salzer, B. *et al.* Engineering AvidCARs for combinatorial antigen recognition and reversible control of CAR function. *Nature communications* **11**, 4166; 10.1038/s41467-020-17970-3 (2020).
7. Nicholson, I. *et al.* Construction and characterisation of a functional CD19 specific single chain Fv fragment for immunotherapy of B lineage leukaemia and lymphoma. *Molecular Immunology* **34**, 1157–1165 (1997).
8. Dietrich, J., Hou, X., Wegener, A. M. & Geisler, C. CD3 gamma contains a phosphoserine-dependent di-leucine motif involved in down-regulation of the T cell receptor. *The EMBO journal* **13**, 2156–2166; 10.1002/j.1460-2075.1994.tb06492.x (1994).
9. Boding, L. *et al.* TCR down-regulation controls T cell homeostasis. *Journal of immunology (Baltimore, Md. : 1950)* **183**, 4994–5005; 10.4049/jimmunol.0901539 (2009).
10. Bonefeld, C. M. *et al.* TCR down-regulation controls virus-specific CD8+ T cell responses. *Journal of Immunology* **181**; 10.4049/jimmunol.181.11.7786 (2008).
11. Pan, Q., Gollapudi, A. S. & Dave, V. P. Biochemical evidence for the presence of a single CD3delta and CD3gamma chain in the surface T cell receptor/CD3 complex. *The Journal of biological chemistry* **279**, 51068–51074; 10.1074/jbc.M406145200 (2004).
12. Brodeur, J.-F., Li, S., Damlaj, O. & Dave, V. P. Expression of fully assembled TCR-CD3 complex on double positive thymocytes. Synergistic role for the PRS and ER retention motifs in the intra-cytoplasmic tail of CD3epsilon. *International immunology* **21**, 1317–1327; 10.1093/intimm/dxp098 (2009).
13. Delgado, P. & Alarcón, B. An orderly inactivation of intracellular retention signals controls surface expression of the T cell antigen receptor. *The Journal of experimental medicine* **201**, 555–566; 10.1084/jem.20041133 (2005).
14. Zhao, Z. *et al.* Structural Design of Engineered Costimulation Determines Tumor Rejection Kinetics and Persistence of CAR T Cells. *Cancer cell* **28**, 415–428; 10.1016/j.ccell.2015.09.004 (2015).
15. Baeuerle, P. A. *et al.* Synthetic TRuC receptors engaging the complete T cell receptor for potent anti-tumor response.
16. Robert C. Sterner & Rosalie M. Sterner. CAR-T cell therapy: current limitations and potential strategies; 10.1038/s41408-021-00459-7.
17. Arcangeli, S. *et al.* CAR T cell manufacturing from naive/stem memory T lymphocytes enhances antitumor responses while curtailing cytokine release syndrome. *Journal of Clinical Investigation* **132** (2022).

18. Maria Castella and Manel Juan. Point-Of-Care CAR T-Cell Production (ARI-0001) Using a Closed Semi-automatic Bioreactor: Experience From an Academic Phase I Clinical Trial. *Frontiers in Immunology* **11**; 10.3389/fimmu.2020.00482 (2020).
19. Mackall, C. L. *et al.* Distinctions Between CD8+ and CD4+ T-Cell Regenerative Pathways Result in Prolonged T-Cell Subset Imbalance After Intensive Chemotherapy. *Blood* **89**, 3700–3707; 10.1182/blood.V89.10.3700 (1997).
20. Hassin, D., Garber, O. G., Meiraz, A., Schiffenbauer, Y. S. & Berke, G. Cytotoxic T lymphocyte perforin and Fas ligand working in concert even when Fas ligand lytic action is still not detectable. *Immunology* **133**, 190–196; 10.1111/j.1365-2567.2011.03426.x (2011).
21. Pettmann, J. *et al.* The discriminatory power of the T cell receptor. *eLife* **10**; 10.7554/eLife.67092 (2021).
22. Ghasemi, A. & Zahedias, S. Normality Tests for Statistical Analysis: A Guide for Non-Statistician. *Int J Endocrinol Metab.* **10**, 486–489; 10.5812/ijem.3505 (2012).

Decision Letter, first revision:

17th Aug 2023

Dear Susana,

Thank you for submitting your revised manuscript "Harnessing CD3 diversity to optimise CAR T cells" (NI-A35052B). It has now been seen by the original referees and their comments are below. The reviewers find that the paper has improved in revision, and therefore we'll be happy in principle to publish it in Nature Immunology, pending minor revisions to satisfy the referees' final requests and to comply with our editorial and formatting guidelines.

We will now perform detailed checks on your paper and will send you a checklist detailing our editorial and formatting requirements in about a week. Please do not upload the final materials and make any revisions until you receive this additional information from us.

If you had not uploaded a Word file for the current version of the manuscript, we will need one before beginning the editing process; please email that to immunology@us.nature.com at your earliest convenience.

Thank you again for your interest in Nature Immunology. Please do not hesitate to contact me if you have any questions.

Kind regards,

Laurie

Laurie A. Dempsey, Ph.D.
Senior Editor
Nature Immunology
l.dempsey@us.nature.com
ORCID: 0000-0002-3304-796X

Reviewer #1 (Remarks to the Author):

The authors have addressed my concerns satisfactorily. However, in the 'Dimeric CARs improve overall functionality' section, the authors state that dimeric BBg and BBd CARs are more efficient to transmit signals at lower expression. But, the data in the new figure 6 does not support this. Based on the figures, CD25+ is higher for the dimer compared to monomer and vice-versa for CD69+ and 4-1BB+ for both BBg and BBd (Fig. 6f). IGNg and TNF- α production are comparable between monomers and dimers for both BBg and BBd (Fig. 6g,h). This needs to be clarified.

Reviewer #2 (Remarks to the Author):

The authors have addressed my concerns and provided additional experiments to address the importance of dimerisation for the normally heterodimeric CD3ed and CD3eg modules. They have clarified their writing in other cases to better explain why other experiments were done or how they were interpreted. I have no further concerns.

Author Rebuttal, first revision:

See inserted PDF

NI-A35052B, "Harnessing CD3 diversity to optimize CAR T cells"

Point-by-point response

Reviewer #1

Reviewer #1 (Remarks to the Author):

The authors have addressed my concerns satisfactorily. However, in the 'Dimeric CARs improve overall functionality' section, the authors state that dimeric BB γ and BB δ CARs are more efficient to transmit signals at lower expression. But, the data in the new figure 6 does not support this. Based on the figures, CD25⁺ is higher for the dimer compared to monomer and vice-versa for CD69⁺ and 4-1BB⁺ for both BB γ and BB δ (Fig. 6f). IGNg and TNF- α production are comparable between monomers and dimers for both BB γ and BB δ (Fig. 6g,h). This needs to be clarified.

Response: We are delighted that the Reviewer found our revised version satisfactory. We highly appreciate the insightful review process and her/his suggestions that help us to significantly improve our manuscript by providing new insights. Regarding the last concern in the section "Dimeric CARs improve overall functionality", we have now improved our wording to make our message clearer. Several reports have suggested a correlation between the level of expression of a given CAR and its functionality¹⁻⁵. The Monomeric BB γ and BB δ CARs are expressed 1.6 and 2.9 times more, respectively, than their dimeric counterparts. However, monomeric and dimeric CARs hold very similar levels of T cell activation (dimeric CARs showed an increase in CD25⁺ cells, but a decrease for CD69⁺ and 4-1BB⁺ cells while monomeric CARs vice-versa, both with comparable INF- γ and TNF- α production) is for us an indication that the dimeric forms transmit signals more efficiently.

The new wording can be found now on page 8 (lines 284-286): "*Dimeric BB γ and BB δ CARs are more efficient at transmitting signals than their monomeric counterparts, since they promote the similar level of activation despite lower expression (Fig. 6d-h)*".

Reviewer #2 (Remarks to the Author):

The authors have addressed my concerns and provided additional experiments to address the importance of dimerisation for the normally heterodimeric CD3 ϵ d and CD3 ϵ g modules. They have clarified their writing in other cases to better explain why other experiments were done or how they were interpreted. I have no further concerns.

Response: We are pleased to hear that the Reviewer is satisfied with our revised version and has no additional concerns. We value the thoughtful review process and the suggestions provided, as they have greatly enriched our manuscript by offering new perspectives and significant improvements.

References:

1. Rodriguez-Marquez, P. *et al.* CAR density influences antitumoral efficacy of BCMA CAR T cells and correlates with clinical outcome. *Sci. Adv.* **8**, eabo0514 (2022).
2. Eyquem, J. *et al.* Targeting a CAR to the TRAC locus with CRISPR/Cas9 enhances tumour rejection. *Nature* **543**, 113–117 (2017).
3. Tristán-Manzano, M. *et al.* Physiological lentiviral vectors for the generation of improved CAR-T cells. *Molecular Therapy - Oncolytics* **25**, 335–349 (2022).
4. Ho, J.-Y. *et al.* Promoter usage regulating the surface density of CAR molecules may modulate the kinetics of CAR-T cells in vivo. *Molecular Therapy - Methods & Clinical Development* **21**, 237–246 (2021).
5. Walker, A. J. *et al.* Tumor Antigen and Receptor Densities Regulate Efficacy of a Chimeric Antigen Receptor Targeting Anaplastic Lymphoma Kinase. *Molecular Therapy* **25**, 2189–2201 (2017).

Final Decision Letter:

Dear Susana,

I am delighted to accept your manuscript entitled "Harnessing CD3 diversity to optimize CAR T cells" for publication in an upcoming issue of Nature Immunology.

Over the next few weeks, your paper will be copyedited to ensure that it conforms to Nature Immunology style. Once your paper is typeset, you will receive an email with a link to choose the appropriate publishing options for your paper and our Author Services team will be in touch regarding any additional information that may be required.

Please note that *Nature Immunology* is a Transformative Journal (TJ). Authors may publish their research with us through the traditional subscription access route or make their paper immediately open access through payment of an article-processing charge (APC). Authors will not be required to make a final decision about access to their article until it has been accepted. [Find out more about Transformative Journals](https://www.springernature.com/gp/open-research/transformative-journals).

If you have any questions about our publishing options, costs, Open Access requirements, or our legal

forms, please contact ASJournals@springernature.com

Your paper will be published online soon after we receive your corrections and will appear in print in the next available issue. Content is published online weekly on Mondays and Thursdays, and the embargo is set at 16:00 London time (GMT)/11:00 am US Eastern time (EST) on the day of publication. Now is the time to inform your Public Relations or Press Office about your paper, as they might be interested in promoting its publication. This will allow them time to prepare an accurate and satisfactory press release. Include your manuscript tracking number (NI-A35052C) and the name of the journal, which they will need when they contact our office.

About one week before your paper is published online, we shall be distributing a press release to news organizations worldwide, which may very well include details of your work. We are happy for your institution or funding agency to prepare its own press release, but it must mention the embargo date and Nature Immunology. Our Press Office will contact you closer to the time of publication, but if you or your Press Office have any enquiries in the meantime, please contact press@nature.com.

Also, if you have any spectacular or outstanding figures or graphics associated with your manuscript - though not necessarily included with your submission - we'd be delighted to consider them as candidates for our cover. Simply send an electronic version (accompanied by a hard copy) to us with a possible cover caption enclosed.

Please note that we encourage the authors to self-archive their manuscript (the accepted version before copy editing) in their institutional repository, and in their funders' archives, six months after publication. Nature Portfolio recognizes the efforts of funding bodies to increase access of the research they fund, and strongly encourages authors to participate in such efforts. For information about our

editorial policy, including license agreement and author copyright, please visit www.nature.com/ni/about/ed_policies/index.html

Kind regards,

Laurie

Laurie A. Dempsey, Ph.D.
Senior Editor
Nature Immunology
l.dempsey@us.nature.com
ORCID: 0000-0002-3304-796X